# FORGET UNLEARNING: TOWARDS TRUE DATA-DELETION IN MACHINE LEARNING

## ABSTRACT

Unlearning has emerged as a technique to efficiently erase information of deleted records from learned models. We show, however, that the influence created by the original presence of a data point in the training set can still be detected after running certified unlearning algorithms (which can result in its reconstruction by an adversary). Thus, under realistic assumptions about the dynamics of model releases over time and in the presence of adaptive adversaries, we show that unlearning is not equivalent to data deletion and does not guarantee the "right to be forgotten." We then propose a more robust data-deletion guarantee and show that it is necessary to satisfy differential privacy to ensure true data deletion. Under our notion, we propose an accurate, computationally efficient, and secure data-deletion machine learning algorithm in the online setting based on noisy gradient descent algorithm.

## 1 INTRODUCTION

Many corporations today collect their customers' private information to train Machine Learning (ML) models that power a variety of services, encompassing recommendations, searches, targeted ads, and more. To prevent any unintended use of personal data, privacy policies, such as the General Data Protection Regulation (GDPR) and the California Consumer Privacy Act (CCPA), require that these corporations provide the "*right to be forgotten*" (RTBF) to their data subjects—if a user wishes to revoke access to their data, an organization must comply by erasing all information about the user without undue delay (which is typically a month). This includes ML models trained in standard ways as model inversion (Fredrikson et al., 2015) and membership inference attacks (Shokri et al., 2017; Carlini et al., 2019) demonstrate that individual training data can be exfiltrated from these models.

Periodic retraining of models after excluding deleted users can be costly. So, there is a growing interest in designing computationally cheap *Machine Unlearning* algorithms as an alternative to retraining for erasing the influence of deleted data from (and registering the influence of added data to) trained models. Since it is generally difficult to tell how a specific data point affects a model, Ginart et al. (2019) propose quantifying the worst-case information leakage from an unlearned model through an *unlearning guarantee* on the mechanism, defined as a *differential privacy* (DP) like $(\varepsilon, \delta)$-indistinguishability between its output and that of retraining on the updated database. With some minor variations in this definition, several mechanisms have been proposed and certified as unlearning algorithms in literature (Ginart et al., 2019; Izzo et al., 2021; Sekhari et al., 2021; Neel et al., 2021; Guo et al., 2019; Ullah et al., 2021).

However, *is indistinguishability to retraining a sufficient guarantee of data deletion?* We argue that it is not. In the real world, a user's decision to remove his information is often affected by what a deployed model reveals about him. The same revealed information may also affect other users' decisions. Such *adaptive* requests make the records in a database interdependent, causing a retrained model to contain influences of a record even if the record is no longer in the training set. We demonstrate on a certified unlearning mechanism that if an adversary is allowed to design an *adaptive requester* that interactively generates database edit requests as a function of published models, she can re-encode a target record in the curator's database before its deletion. We argue that under adaptive requests, measuring data-deletion via indistinguishability to retraining (as proposed by Gupta et al. (2021)) is fundamentally flawed because it does not capture the influence a record might have previously had on the rest of the database. Our example shows a clear violation of the

RTBF since even after retraining on the database with the original record removed, a model can reveal substantial information about the deleted record due to the possibility of re-encodings.

*Is an unlearning guarantee a sound and complete measure of data deletion when requests are non-adaptive?* Again, we argue that it is neither. A sound data-deletion guarantee must ensure the non-recovery of deleted records from an *infinite* number of model releases after deletion. However, approximate indistinguishability to retraining implies an inability to accurately recover deleted data from a *singular* unlearned model only, which we argue is not sufficient. We show that certain algorithms can satisfy an unlearning guarantee yet blatantly reveal the deleted data eventually over multiple releases. The vulnerability arises in algorithms that maintain partial computations in internal data structures for speeding up subsequent deletions. These internal states can retain information even after record deletion and influence multiple future releases, making the myopic unlearning guarantee unreliable in an online setting. Several proposed unlearning algorithms in literature (Ginart et al., 2019; Neel et al., 2021) are stateful (rely on internal states) and, therefore, cannot be trusted. Secondly, unlearning is an incomplete notion of data deletion as it excludes valid data-deletion mechanisms that do not imitate retraining. For instance, a (useless) mechanism that outputs a fixed untrained model on any request is a valid deletion algorithm. However, since its output is easily distinguishable from retraining, it fails to satisfy any meaningful unlearning guarantees.

This paper proposes a *sound definition of data deletion* that does not suffer from the abovementioned shortcomings. According to our notion, a data-deletion mechanism is reliable if **A)** it is stateless (i.e., it maintains no internal data structures), and **B)** generates models that are indistinguishable from some random variable that is independent of the deleted records. Statelessness thwarts the danger of sustained information leakage through internal data structures after deletion. Moreover, by defining *data deletion* as indistinguishability with any deleted-record independent random variable as oppsed to the output of retraining, we ensure reliability in presence of adaptive requests that create dependence between current and deleted records in the database.

In general, we show that data-deletion mechanisms must be *differentially private with respect to the remaining records* to be reliable when requests are adaptive. DP also protects against membership inference attacks that extract deleted records by looking at models before and after deletion (Chen et al., 2021). We emphasize that we are not advocating for doing data deletion through differentially-private mechanisms simply because it caps the information content of all records equally, deleted or otherwise. Instead, a data-deletion mechanisms should provide two differing information reattainment bounds; one for records currently in the database in the form of a differential privacy guarantee and the other for records previously deleted in the form of a data-deletion guarantee. We also provide a reduction theorem that if a mechanism is differentially private with respect to the remaining records and satisfies a data-deletion guarantee under non-adaptive edit requests, then it also satisfies a data-deletion guarantee under adaptive requests. Based on this reduction, we redefine the problem of data-deletion as designing a mechanism that **(1.)** satisfies a data-deletion guarantee against non-adaptive deletion requests, **(2.)** is differentially private for remaining records, and **(3.)** has the same utility guarantee as retraining under identical differential privacy constraints. We judge the usefulness of a data-deletion mechanism based on its computational savings over retraining.

For our refined problem formulation, we provide a data-deletion solution based on Noisy Gradient Descent (Noisy-GD), a popular differentially private learning algorithm (Bassily et al., 2014; Abadi et al., 2016; Chourasia et al., 2021). Our solution demonstrates a powerful synergy between data deletion and differential privacy as the same noise needed for the privacy of records in the database also rapidly erases information regarding records deleted from the database. We provide a data-deletion guarantee for Noisy-GD in terms of Rényi divergence (Rényi et al., 1961) bound (which implies $(\varepsilon, \delta)$-indistinguishability (Mironov, 2017)). For convex and smooth losses, we certify that under a $(q, \varepsilon_{\mathrm{dd}})$-Rényi data-deletion and $(q, \varepsilon_{\mathrm{dp}})$-Rényi DP constraint, our Noisy-GD based deletion mechanism for $d$-dimensional models over $n$-sized databases under adaptive edit requests that modify no more than $r$ records can maintain optimal excess empirical risk of the order $O\big(\frac{qd}{\varepsilon_{\mathrm{dp}}n^2}\big)$ while saving $\Omega(n\log(\min\{\frac{n}{r}, n\sqrt{\frac{\varepsilon_{\mathrm{dd}}}{qd}}\})$ computations in gradient complexity. Our utility guarantee matches the known lower bound on private empirical risk minimization under same privacy budget (Bassily et al., 2014). We also provide data-deletion guarantee in the non-convex setting under the assumption that loss function is bounded and smooth, and show a computational saving of $\Omega(dn\log\frac{n}{r})$ in gradient complexity while maintaining an excess risk of $\tilde{O}\big(\frac{qd}{\varepsilon_{\mathrm{dp}}n^2} + \frac{1}{n}\sqrt{\frac{q}{\varepsilon_{\mathrm{dp}}}}\big)$.

## 2 Model and Preliminaries

### 2.1 Indistinguishability Notions and Differential Privacy

We provide the basics of indistinguishability of random variables (with more details in Appendix D). Let $\Theta, \Theta'$ be two random variables in space $\mathcal{O}$ with probability densities $\nu, \nu'$ respectively.

**Definition 2.1** (($\varepsilon, \delta$)-indistinguishability (Dwork et al., 2014))**.** *We say $\Theta$ and $\Theta'$ are ($\varepsilon, \delta$)-indistinguishable and write $\Theta \overset{\varepsilon,\delta}{\approx} \Theta'$ if, for all $O \subset \mathcal{O}$,*

$$\mathbb{P}\left[\Theta \in O\right] \leq e^{\varepsilon} \mathbb{P}\left[\Theta' \in O\right] + \delta \quad and \quad \mathbb{P}\left[\Theta' \in O\right] \leq e^{\varepsilon} \mathbb{P}\left[\Theta \in O\right] + \delta. \tag{1}$$

**Definition 2.2** (Rényi divergence (Rényi et al., 1961))**.** *Rényi divergence of $\nu$ w.r.t. $\nu'$ of order $q > 1$ is defined as*

$$R_q\left(\nu\|\nu'\right) = \frac{1}{q-1} \log E_q\left(\nu\|\nu'\right), \quad where \quad E_q\left(\nu\|\nu'\right) = \underset{\theta \sim \nu'}{\mathbb{E}}\left[\left(\frac{\nu(\theta)}{\nu'(\theta)}\right)^q\right], \tag{2}$$

*when $\nu$ is absolutely continuous w.r.t. $\nu'$ (denoted as $\nu \ll \nu'$). If $\nu \not\ll \nu'$, we'll say $R_q\left(\nu\|\nu'\right) = \infty$.*

**Remark 1.** *Rényi divergence is assymetric (i.e. $R_q\left(\nu\|\nu'\right) \neq R_q\left(\nu'\|\nu\right)$) and implies indistinguishability only in one direction. Mironov (2017, Proposition 3) show that $R_q\left(\nu\|\nu'\right) \leq \varepsilon_0$ implies $\mathbb{P}\left[\Theta \in O\right] \leq e^{\varepsilon}\mathbb{P}\left[\Theta' \in O\right] + \delta$ with $\varepsilon = \varepsilon_0 + \frac{\log 1/\delta}{q-1}$ and any $0 < \delta < 1$.*

**Definition 2.3** (Differential Privacy (Dwork et al., 2014; Mironov, 2017))**.** *A randomized mechanism $\mathcal{M} : \mathcal{X}^n \to \mathcal{O}$ is ($\varepsilon, \delta$)-differentially private if $\mathcal{M}(\mathcal{D}) \overset{\varepsilon,\delta}{\approx} \mathcal{M}(\mathcal{D}')$ for all neighbouring databases $\mathcal{D}, \mathcal{D}' \in \mathcal{X}^n$. Similarly, $\mathcal{M}$ is ($q, \varepsilon$)-Rényi differentially private if $R_q\left(\mathcal{M}(\mathcal{D})\|\mathcal{M}(\mathcal{D}')\right) \leq \varepsilon$.*

### 2.2 Learning Framework: ERM

Let $\mathcal{D}$ be a database of $n$ ordered records taken from a data universe $\mathcal{X}$ and let $\mathcal{O}$ be the space of learnable parameters and any associated auxiliary metadata (what constitutes a metadata is clarified later). Let $\boldsymbol{\ell}(\theta; \mathbf{x}) : \mathcal{O} \times \mathcal{X} \to \mathbb{R}$ be a loss function of a parameter $\theta \in \mathcal{O}$ for a record $\mathbf{x} \in \mathcal{X}$. In this paper, we consider the problem of *empirical risk minimization* (ERM) of the average $\boldsymbol{\ell}(\theta; \mathbf{x})$ over records in the database $\mathcal{D}$ under $L2$ regularization $\mathbf{r}(\theta) = \frac{\lambda\|\theta\|_2^2}{2}$, i.e., the minimization objective is $\mathcal{L}_{\mathcal{D}}(\theta) = \frac{1}{n}\sum_{\mathbf{x}\in\mathcal{D}} \boldsymbol{\ell}(\theta; \mathbf{x}) + \mathbf{r}(\theta)$. The *excess empirical risk* of a model $\Theta$ for $\mathcal{D}$ by is defined as $\mathrm{err}(\Theta; \mathcal{D}) = \mathbb{E}\left[\mathcal{L}_{\mathcal{D}}(\Theta) - \mathcal{L}_{\mathcal{D}}(\theta_{\mathcal{D}}^*)\right]$, where $\theta_{\mathcal{D}}^* = \underset{\theta\in\mathcal{O}}{\arg\min} \mathcal{L}_{\mathcal{D}}(\theta)$ and expectation is taken over $\Theta$.

We build on a popular DP-ERM algorithm called Noisy-GD (Abadi et al., 2016), described in Algorithm 1 below, and provide Rényi DP guarantees on it in Appendix G.3.

---

**Algorithm 1** Noisy-GD: Noisy Gradient Descent

---

**Require:** Database $\mathcal{D} \in \mathcal{X}^n$, start model $\Theta_0 \in \mathcal{O}$, number of steps $K \in \mathbb{N}$.
1: **for** $k = 0, 1, \cdots, K-1$ **do**
2: $\quad \nabla\mathcal{L}_{\mathcal{D}}(\Theta_{\eta k}) = \frac{1}{n}\sum_{\mathbf{x}\in\mathcal{D}} \nabla\boldsymbol{\ell}(\Theta_{\eta k}; \mathbf{x}) + \nabla\mathbf{r}(\Theta_{\eta k})$
3: $\quad \Theta_{\eta(k+1)} = \Theta_{\eta k} - \eta\nabla\mathcal{L}_{\mathcal{D}}(\Theta_{\eta k}) + \sqrt{2\eta}\mathcal{N}\left(0, \sigma^2\mathbb{I}_d\right)$
4: Output $\Theta_{\eta K}$

---

### 2.3 Online Edit Requests and Machine Unlearning

Suppose that any database $\mathcal{D} \in \mathcal{X}^n$ can be modified by *edit requests* that replaces $r$ distinct records [1].

**Definition 2.4** (Edit request)**.** *A replacement operation $\langle\mathrm{ind}, \mathbf{y}\rangle \in [n] \times \mathcal{X}$ on a database $\mathcal{D} = (\mathbf{x}_1, \cdots, \mathbf{x}_n)$ performs the following modification:*

$$\mathcal{D} \circ \langle\mathrm{ind}, \mathbf{y}\rangle = (\mathbf{x}_1, \cdots, \mathbf{x}_{\mathrm{ind}-1}, \mathbf{y}, \mathbf{x}_{\mathrm{ind}+1}, \cdots, \mathbf{x}_n). \tag{3}$$

*Let $r \leq n$ and $\mathcal{U}^r = [n]_{\neq}^r \times \mathcal{X}^r$. An edit request $u = \{\langle\mathrm{ind}_1, \mathbf{y}_1\rangle, \cdots, \langle\mathrm{ind}_r, \mathbf{y}_r\rangle\} \in \mathcal{U}^r$ on $\mathcal{D}$ is defined as batch of $r$ replacement operations modifying distinct indices atomically, i.e.*

$$\mathcal{D} \circ u = \mathcal{D} \circ \langle\mathrm{ind}_1, \mathbf{y}_1\rangle \circ \cdots \circ \langle\mathrm{ind}_r, \mathbf{y}_r\rangle, \quad where \; \mathrm{ind}_i \neq \mathrm{ind}_j \; for \; all \; i \neq j. \tag{4}$$

---

[1] We consider replacement instead of separate addition/deletion to ensure that database size doesn't change.

Let $\mathcal{O}$ denote the space of model parameters plus any state variables or data-structures that may be leveraged for processing edit requests and $\Phi$ be the space of publishable outcomes such as sanitized models or predictions. For an initial database $\mathcal{D}_0 \in \mathcal{X}^n$ and an unbounded sequence of edit requests $(u_i)_{i \geq 1}$, the job of a *data curator* is to train a model $\hat{\Theta}_i \in \mathcal{O}$ with small empirical risk for each database $\mathcal{D}_i = \mathcal{D}_0 \circ u_1 \circ \cdots \circ u_i$ and use it to release a corresponding publishable outcome, $\phi_i \in \Phi$. For this task, the trivial approach of retraining from scratch by executing a *learning algorithm* on each $\mathcal{D}_i$ could be computationally expensive. This initiated the study of *machine unlearning* algorithms (Cao & Yang, 2015; Ginart et al., 2019; Guo et al., 2019; Izzo et al., 2021) that avoid the cost of retraining by instead fine-tuning an already trained model $\hat{\Theta}_{i-1}$ to cheaply erase (introduce) the influence of deleted (added) records in edit request $u_i$ to $\mathcal{D}_{i-1}$ for producing the next model $\hat{\Theta}_i$.

In this paper, we adopt the machine unlearning formulation of Gupta et al. (2021) and Neel et al. (2021) described as follows. The curator comprises of three mechanisms: a learning algorithm $\mathrm{A} : \mathcal{X}^n \to \mathcal{O}$, a data-deletion or unlearning algorithm $\bar{\mathrm{A}} : \mathcal{X}^n \times \mathcal{U}^r \times \mathcal{O} \to \mathcal{O}$, and a publish function $f_{\mathrm{pub}} : \mathcal{O} \to \Phi$. To generate the initial model with accompanying state metadata $\hat{\Theta}_0 \in \mathcal{O}$ on the initial database $\mathcal{D}_0 \in \mathcal{X}^n$, the curator executes the learning algorithm $\mathrm{A}(\mathcal{D}_0)$. Thereafter, to process an incoming edit request $u_i \in \mathcal{U}^r$ at any step $i \geq 1$, the curator executes data-deletion algorithm $\bar{\mathrm{A}}(\mathcal{D}_{i-1}, u_i, \hat{\Theta}_{i-1})$ that maps the current database $\mathcal{D}_{i-1}$, the edit request $u_i$, and the current model with metadata $\hat{\Theta}_{i-1}$ to the next model with metadata $\hat{\Theta}_i \in \mathcal{O}$. While all the generated models and corresponding metadata is kept secret, the curator uses the publish function to generate the publishable outcome $\phi_i = f_{\mathrm{pub}}(\hat{\Theta}_i)$ at every step $i \geq 0$.

Gupta et al. (2021) note that edit requests in real world could often be *adaptive*, i.e., a request $u_i$ may depend on (a subset of) the history of prior published outcomes $\phi_{<i} = (\phi_1, \cdots, \phi_i)$. For instance, a voter may decide to change his inclination after seeing pre-election results. They model such an intearctive environment through an *adaptive requester* defined as follows.

**Definition 2.5** (Update requester (Gupta et al., 2021)). *A $p$-adaptive $r$-requester is a mapping $\mathcal{Q} : \Phi^{\leq p} \times \mathcal{U}^{r*} \to \mathcal{U}^r$ that takes as input a maximum of $p$ of the published outcomes generated by the curator at arbitrary edit steps $s_1 < s_2 < \cdots < s_p$, and the entire history of previously generated edit requests to generate the next edit request. For a $p$-adaptive $r$-requester $\mathcal{Q}$, the edit request $u_i$ at any step $i \geq 1$ can be written as*

$$u_i = \mathcal{Q}(\phi_{s_1}, \phi_{s_2}, \cdots, \phi_{s_j}; u_1, u_2, \cdots, u_{i-1}), \tag{5}$$

*such that $s_j < i$. We refer to $0$-adaptive requesters as non-adaptive. And, by $\infty$-adaptive requesters, we mean requesters that have access to the entire history of interaction transcript $(\phi_{<i}; u_{<i})$.*

Since, unlike retraining, an unlearning algorithm fine-tunes a model containing the information of records to be deleted, we need statistical guarantees on the worst-case amount of information that might still remain in the unlearned model. Ginart et al. (2019) and Guo et al. (2019) propose quantifying *data-deletion* ability of an algorithm $\bar{\mathrm{A}}$ based on its $(\varepsilon, \delta)$-indistinguishability w.r.t. the fresh-retraining algorithm $\mathrm{A}$, calling it an *unlearning guarantee* (more details in Appendix E.1). In this paper, we mainly consider the extension of unlearning definitions by Neel et al. (2021) and Gupta et al. (2021) to the online setting of arbitrarily long and adaptive edit sequences [2].

**Definition 2.6** (($\varepsilon, \delta$)-unlearning (Neel et al., 2021; Gupta et al., 2021)). *We say that $\bar{\mathrm{A}}$ is an $(\varepsilon, \delta)$-unlearning algorithm for $\mathrm{A}$ under a publish function $f_{\mathrm{pub}}$, if for all initial databases $\mathcal{D}_0 \in \mathcal{X}^n$ and all non-adaptive $1$-requesters $\mathcal{Q}$, the following condition holds. For every edit step $i \geq 1$, and for all generated edit sequences $u_{\leq i} \overset{def}{=} (u_1, \cdots, u_i)$,*

$$f_{\mathrm{pub}}(\bar{\mathrm{A}}(\mathcal{D}_{i-1}, u_i, \hat{\Theta}_{i-1}))\big|_{u_{\leq i}} \overset{\varepsilon, \delta}{\approx} f_{\mathrm{pub}}(\mathrm{A}(\mathcal{D}_i)). \tag{6}$$

*If the same condition holds for all $\infty$-adaptive $1$-requesters $\mathcal{Q}$, we say that $\bar{\mathrm{A}}$ is an $(\varepsilon, \delta)$-adaptive-unlearning algorithm for $\mathrm{A}$.*

---

[2]Definition 2.6 is stronger than the adaptive unlearning definition of Gupta et al. (2021) since theirs require satisfying only one-sided indistinguishability with at-least $(1 - \gamma)$ probability over generated edit requests $u_{\leq i}$.

## 3 EXISTING UNLEARNING GUARANTEES ARE UNSOUND AND INCOMPLETE

Data deletion under the law of "right to be forgotten" (RTBF) is an obligation to *permanently erase* all information about an individual upon a verified request. In order to comply, a corporation's actions must not reveal any information identifiable or linkable to a deleted user in the future. In this section, we argue that unlearning guarantee in Definition 2.6 is *neither a sound nor a complete* measure of data-deletion from ML models that RTBF enforces.

**Threat model.** Suppose, for an arbitrary step $i \geq 1$, an adversary is interested in finding out the identity of a record in the database $\mathcal{D}_{i-1}$ that was deleted by the edit request $u_i$. Since RTBF is violated only when the curator reveals information *after the deletion request*, we assume that the adversary only has access to the post-deletion releases by the curator, i.e., she observes $\phi_i, \phi_{i+1}, \cdots$. Additionally, we also assume that the adversary knows how users might react to a published outcome. That is to say, our adversary *knows some dependence relationship* between random variables $\phi_0, \cdots, \phi_{i-1}$ and $u_1, \cdots, u_{i-1}$, but does not explicitly observe these random variables. For instance, an adversary might know that if the outcome $\phi_1$ predicts that "Donald Trump is winning the election," then some democratic users might delete their data while some new republican users might contribute their data to the curator. So, even though the adversary does not observe the actual outcome $\phi_1$ or the ensuing edit request $u_2$, and so on, she can still exploit knowledge about this dependence a-posteriori to infer the identity of a deleted record. To capture the *worst-case knowledge about the dependence* between unobserved outcomes and unobserved edit requests, we model our adversary to have the power to design an adaptive requester $\mathcal{Q}$ that interacts with the curator in the first $i - 1$ steps. However, the adversary *does not observe the interaction transcript* $(\phi_0, u_1, \phi_1, \cdots u_{i-1}, \phi_{i-1})$ of $\mathcal{Q}$.

**Unsoundness due to secret states.** The unlearning Definition 2.6 is a bound on information leakage about a deleted record through a *single released outcome*. However, our adversary can observe multiple (potentially infinite) releases after deletion. We argue that algorithms satisfying Definition 2.6 can lead to blatant non-privacy of a deleted record under our threat model, even for a weaker adversary that cannot design an adaptive requester (i.e., knows nothing about the dependence between unobserved outcomes and edit requests). The vulnerability arises as Definition 2.6 permits algorithms to maintain secret states while using a publishing function $f_{\text{pub}}$ for releases. These internal states may propagate encoded information about records even after their deletion from the database. So, every subsequent release by an unlearning algorithm can reveal new information about a record that was purportedly erased multiple edits earlier. We demonstrate in the following theorem that a certified unlearning algorithm can reveal a limited amount of information about a deleted record per release so as not to break the unlearning certification, yet eventually reveal everything about the record to an adversary that observes enough future releases.

**Theorem 1.** *For every $\varepsilon > 0$, there exists a pair $(A, \bar{A})$ of algorithms that satisfy $(\varepsilon, 0)$-unlearning under a publish function $f_{\text{pub}}$ such that for all non-adaptive 1-requesters $\mathcal{Q}$, their exists an adversary that can correctly infer the identity of a record deleted at any arbitrary edit step $i \geq 1$ by observing only the post-edit releases $\phi_{\geq i} = (\phi_i, \phi_{i+1}, \cdots)$.*

**Unsoundness due to adaptivity.** For an adversary that knows some dependence relationship between the unobserved outcomes and edit requests, a much more severe violation of RTBF may occur, even when an unlearning algorithm does not maintain secret internal states and perfectly imitates retraining (i.e., satisfy a $(0, 0)$-adaptive unlearning guarantee for identity publish function $f_{\text{pub}}(\theta) = \theta$). This vulnerability arises because the indistinguishability in Definition 2.6 protects the privacy of deleted records but not that of records currently present. A certified adaptive unlearning algorithm is allowed to reveal unbounded information about a target record before its deletion. This revealed information can have a major influence on the subsequent edit requests in the worst case, potentially causing the curator's database to have patterns specific to the identity of the target record even after its deletion. An adversary knowing the possible patterns and their causes (i.e., the dependence relationship between unobserved outcomes and requests) can therefore infer the target records's identity from post-deletion releases. We concretize this vulnerability in the following theorem.

**Theorem 2.** *There exists a pair $(A, \bar{A})$ of algorithms that satisfy $(0, 0)$-adaptive-unlearning for an identity publish function $f_{\text{pub}}(\theta) = \theta$ such that by designing a 1-adaptive 1-requester $\mathcal{Q}$, an adversary, even with no access to $\mathcal{Q}$'s interaction transcript, can infer the identity of a record deleted at any arbitrary edit step $i > 3$ with probability at-least $1 - (1/2)^{i-3}$ from the post-edit release $\phi_i$.*

**Incompleteness.** Another issue with unlearning guarantees is that valid data-deletion algorithms may fail to satisfy it. As per Definition 2.6, the publishable outcome generated by a data-deletion mechanism must imitate that of a retraining algorithm for satisfying an unlearning guarantee. However, imitating retraining is not necessary for data deletion. For instance, consider a (useless) mechanism $\bar{A}$ that outputs a fixed untrained model in $\mathcal{O}$ regardless of its inputs. This $\bar{A}$ would be an unacceptable unlearning algorithm for most re-training algorithms A under an identity publish function $f_{\text{pub}}(\theta) = \theta$. However, $f_{\text{pub}}(\bar{A}(\cdot))$ contains no information about the deleted records and should be acceptable under the data-deletion paradigm.

**Remark 2.** *Several prior works on machine unlearning propose data-deletion definitions similar to Definition 2.6 but without an explicit $f_{\text{pub}}$ and for the offline setting of a single-stage deletion to remove $r$ many records from a learned model (Ginart et al., 2019; Guo et al., 2019; Sekhari et al., 2021). In Appendix E.1, we show that these offline unlearning definitions are also unreliable when deletion requests are adaptive. In light of our demonstration of unsoundness of these data-deletion guarantees, we remark that several certified unlearning algorithms in literature should not be trusted to ensure RTBF.*

## 4   REDEFINING DATA-DELETION IN MACHINE LEARNING

In this section, we redefine *data deletion in Machine Learning* to address the problems with the notion of unlearning that we demonstrate in the preceding section. The first change we propose is to rule out the possibility of information leakage through internal data structures (as shown in Theorem 1) by requiring deletion mechanisms to be *stateless*. That is, the models produced by learning or the data-deletion algorithm are directly released without applying any publish function $f_{\text{pub}}$.

Secondly, the following definition of a *data-deletion guarantee* fixes the security blind spot of an adaptive unlearning guarantee. As demonstrated in Theorem 2, an adaptive requester can encode patterns specific to a target record in the database by making edit decisions in response to the observed outcomes. Thus, being indistinguishable from retraining on the edited database does not guarantee data deletion, as the target's information remains extractable even after the target record's deletion, potentially revealing its identity. In our definition, we account for an adaptive adversary's influence by measuring the indistinguishability of a data-deletion mechanism's output from some random variable that is independent of the deleted record.

**Definition 4.1** (($q, \varepsilon$)-data-deletion under $p$-adaptive $r$-requesters). *Let $q > 1$, $\varepsilon \geq 0$, and $p, r \in \mathbb{N}$. We say that an algorithm pair $(A, \bar{A})$ satisfies $(q, \varepsilon)$-data-deletion under $p$-adaptive $r$-requesters if the following condition holds for all $p$-adaptive $r$-requester $\mathcal{Q}$. For every step $i \geq 1$, there exists a randomized mapping $\pi_i^{\mathcal{Q}} : \mathcal{X}^n \to \mathcal{O}$ such that for all initial databases $\mathcal{D}_0 \in \mathcal{X}^n$,*

$$\mathrm{R}_q\left(\bar{A}(\mathcal{D}_{i-1}, u_i, \hat{\Theta}_{i-1}) \middle\| \pi_i^{\mathcal{Q}}(\mathcal{D}_0 \circ \langle \text{ind}, \mathbf{y} \rangle)\right) \leq \varepsilon, \quad \text{for all } u_i \in \mathcal{U}^r \text{ and all } \langle \text{ind}, \mathbf{y} \rangle \in u_i. \quad (7)$$

We argue that the above definition is a sound guarantee on data-deletion. Suppose that an adversary is interested in identifying a record at index 'ind' in $\mathcal{D}_0$ that is being replaced with record '$\mathbf{y}$' by one of the replacement operations in edit request $u_i \in \mathcal{U}^r$. The inequality (7) above implies that even with the power of designing an adaptive requester $\mathcal{Q}$, no adversary observing the unlearned model $\bar{A}(\mathcal{D}_{i-1}, u_i, \hat{\Theta}_{i-1})$ can be too confident that the observation was *not* from $\pi_i^{\mathcal{Q}}(\mathcal{D}_0 \circ \langle \text{ind}, \mathbf{y} \rangle)$, a distribution that contains no information about the target record $\mathcal{D}_0[\text{ind}]$ by construction. More formally, we provide the following soundness guarantee for Definition 4.1.

**Theorem 3** (Data-deletion Definition 4.1 is sound). *If the algorithm pair $(A, \bar{A})$ satisfies $(q, \varepsilon)$-data-deletion guarantee under all $p$-adaptive $r$-requesters, then even with the power of designing an $p$-adaptive $r$-requester $\mathcal{Q}$ that interacts with the curator before deletion of a target record at any step $i \geq 1$, any adversary observing only the post-deletion releases $(\hat{\Theta}_i, \hat{\Theta}_{i+1}, \cdots)$ has its membership inference advantage for inferring a deleted target bounded as*

$$Adv(MI) \leq \frac{q e^{\varepsilon(q-1)/q}}{q-1} [2(q-1)]^{1/q} - 1. \quad (8)$$

Note that the bound in (8) approaches 0 as $q \to \infty$ and $\varepsilon \to 0$, implying Definition 4.1 is sound.

**Remark 3.** *A non-adaptive requester $\mathcal{Q}$ is equivalent to fixing the request sequence $(u_i)_{i \geq 1}$ a-priori. Hence, given a non-adaptive $\mathcal{Q}$, the database $\mathcal{D}_i \circ \langle \mathrm{ind}, \mathbf{y} \rangle$ is a deterministic function of the database $\mathcal{D}_0 \circ \langle \mathrm{ind}, \mathbf{y} \rangle$ for any $i \geq 1$, thanks to the commutativity of '$\circ$'. Since $\langle \mathrm{ind}, \mathbf{y} \rangle \in u_i$, we remark that for a non-adaptive requester $\mathcal{Q}$, the random variable $\pi_i^{\mathcal{Q}}(\mathcal{D}_0 \circ \langle \mathrm{ind}, \mathbf{y} \rangle)$ in (7) can be the output $\pi(\mathcal{D}_i)$ of any randomized map $\pi : \mathcal{X}^n \to \mathcal{O}$, including the learning algorithm $\mathrm{A}$.*

**Connection with DP.** We highlight that our data-deletion guarantee on pair $(\mathrm{A}, \bar{\mathrm{A}})$ is an information bound on the records that were deleted, while the standard DP guarantee on $\mathrm{A}$ and $\bar{\mathrm{A}}$ is an information bound on the records currently present in the database. In the following theorem, we show that DP with respect to existing records is a *necessary* condition for $(\mathrm{A}, \bar{\mathrm{A}})$ to satisfy adaptive data-deletion.

**Theorem 4** (DP is necessary for adaptive data-deletion). *If learning algorithm $\mathrm{A} : \mathcal{X}^n \to \mathcal{O}$ is not $(0, \delta)$-DP with respect to the replacement of a single record and deletion algorithm $\bar{\mathrm{A}} : \mathcal{X}^n \times \mathcal{U} \times \mathcal{O} \to \mathcal{O}$ is not $(0, \delta)$-DP with respect to the replacement of a single record that is not being deleted, then the pair $(\mathrm{A}, \bar{\mathrm{A}})$ cannot satisfy $(q, \delta^4/2)$-data-deletion under 1-adaptive 1-requester for any $q > 1$.*

Additionally, if $\mathrm{A}$ and $\bar{\mathrm{A}}$ does satisfy DP with respect to existing records then a data-deletion bound under non-adaptive requesters reduces to a data-deletion bound under adaptive requesters.

**Theorem 5** (Non-adaptive data-deletion with Rényi DP implies adaptive data-deletion). *If an algorithm pair $(\mathrm{A}, \bar{\mathrm{A}})$ satisfies $(q, \varepsilon_{\mathrm{dd}})$-data-deletion under all non-adaptive $r$-requesters and $(q, \varepsilon_{\mathrm{dp}})$-Rényi DP, then it also satisfies $(q, \varepsilon_{\mathrm{dd}} + p\varepsilon_{\mathrm{dp}})$-data-deletion under all $p$-adaptive $r$-requesters.*

We provide additional discussion on this reduction theorem in Appendix F.1. By virtue of our reduction Theorem 5, we reformulate the data deletion problem in ML as follows.

**Problem Definition.** Let constants $q > 1$, $0 < \varepsilon_{\mathrm{dd}} \leq \varepsilon_{\mathrm{dp}}$, and $\alpha > 0$. The goal is to design a learning mechanism $\mathrm{A} : \mathcal{X}^n \to \mathcal{O}$ and a deletion mechanisms $\bar{\mathrm{A}} : \mathcal{X}^n \times \mathcal{U}^r \times \mathcal{O} \to \mathcal{O}$ such that

**(1.)** both $\mathrm{A}$ and $\bar{\mathrm{A}}$ satisfy $(q, \varepsilon_{\mathrm{dp}})$-Rényi DP with respect to records in the input database,

**(2.)** pair $(\mathrm{A}, \bar{\mathrm{A}})$ satisfies $(q, \varepsilon_{\mathrm{dd}})$-data-deletion guarantee for all non-adaptive $r$-requesters $\mathcal{Q}$,

**(3.)** and, all models $(\hat{\Theta}_i)_{i \geq 0}$ produced by $(\mathrm{A}, \bar{\mathrm{A}}, \mathcal{Q})$ on any $\mathcal{D}_0 \in \mathcal{X}^n$ have $\mathrm{err}(\hat{\Theta}_i; \mathcal{D}_i) \leq \alpha$.

We judge the benefit of data deletion based on the computation saving that $\bar{\mathrm{A}}$ offers over $\mathrm{A}$ per request. That is, if we want $\mathrm{err}(\bar{\Theta}_i; \mathcal{D}_i) \leq \alpha$ for all $i \geq 0$, where $\bar{\Theta}_i$ is trained from scratch by $\mathrm{A}(\mathcal{D}_i)$, how large is $\mathrm{Cost}(\mathrm{A})$ compared to $\mathrm{Cost}(\bar{\mathrm{A}})$. Note that $\alpha$ should ideally be the optimal excess risk attainable under the $(q, \varepsilon_{\mathrm{dp}})$-RDP constraint. We remark that one may use any reasonable measure of utility in the third constraint, such as population risk instead of excess empirical risk.

## 5 Data Deletion Using Noisy Gradient Descent

This section proposes a simple and effective data-deletion solution based on Noisy-GD Algorithm 1. Our proposed approach falls under the Descent-to-Delete framework proposed by Neel et al. (2021), wherein, after each deletion request $u_i$, we run Noisy-GD starting from the previous model $\hat{\Theta}_{i-1}$ and perform a small number of gradient descent steps over records in the modified database $\mathcal{D}_i = \mathcal{D}_{i-1} \circ u_i$; sufficient to ensure that the information about the deleted records is reduced within a desired bound in the subsequent model $\hat{\Theta}_i$. Formally, our data-deletion solution is defined as follows.

**Definition 5.1** (Data-deletion based on Noisy-GD). *Let space of model parameters be $\mathcal{O} = \mathbb{R}^d$. Our learning algorithm $\mathrm{A}_{Noisy\text{-}GD}(\mathcal{D}_0) = Noisy\text{-}GD(\mathcal{D}_0, \Theta_0, K_{\mathrm{A}})$ runs Noisy-GD algorithm for $K_{\mathrm{A}}$ iterations to generate the first learned model $\hat{\Theta}_0$ on the input database $\mathcal{D}_0 \in \mathcal{X}^n$, with $\Theta_0$ sampled from a weight initialization distribution $\rho$. Our data-deletion algorithm $\bar{\mathrm{A}}_{Noisy\text{-}GD}(\mathcal{D}_{i-1}, u_i, \hat{\Theta}_{i-1}) = Noisy\text{-}GD(\mathcal{D}_{i-1} \circ u_i, \hat{\Theta}_{i-1}, K_{\bar{\mathrm{A}}})$ processes an edit request $u_i \in \mathcal{U}^r$ by running $K_{\bar{\mathrm{A}}}$ iterations of Noisy-GD algorithm on the updated database $\mathcal{D}_i = \mathcal{D}_{i-1} \circ u_i$ to generate the updated model $\hat{\Theta}_i$ from the current model $\hat{\Theta}_{i-1} \in \mathbb{R}^d$.*

For this setup, our objective is to provide conditions under which the algorithm pair $(\mathrm{A}_{\text{Noisy-GD}}, \bar{\mathrm{A}}_{\text{Noisy-GD}})$ satisfies objectives **(1.)**, **(2.)**, and **(3.)** as stated in the problem definition above and analyze the computational savings of using $\bar{\mathrm{A}}_{\text{Noisy-GD}}$ over $\mathrm{A}_{\text{Noisy-GD}}$ in terms of the gradient complexity $n(K_{\mathrm{A}} - K_{\bar{\mathrm{A}}})$.

## 5.1 Data-Deletion and Accuracy Guarantees Under Convexity

We give the following guarantee for pair $(A_{\text{Noisy-GD}}, \bar{A}_{\text{Noisy-GD}})$ under convexity of loss $\ell(\theta; \mathbf{x})$.

**Theorem 6** (Accuracy, privacy, deletion, and computation tradeoffs). *Let constants $\lambda, \beta, L > 0$, $q > 1$, and $0 < \varepsilon_{\text{dd}} \leq \varepsilon_{\text{dp}}$. Define constant $\kappa = \frac{\lambda + \beta}{\lambda}$. Let the loss function $\ell(\theta; \mathbf{x})$ be twice differentiable, convex, $L$-Lipschitz, and $\beta$-smooth, the regularizer be $\mathbf{r}(\theta) = \frac{\lambda}{2} \|\theta\|_2^2$. If the learning rate be $\eta = \frac{1}{2(\lambda + \beta)}$, the gradient noise variance is $\sigma^2 = \frac{4qL^2}{\lambda \varepsilon_{\text{dp}} n^2}$, and the weight initialization distribution is $\rho = \mathcal{N}\left(0, \frac{\sigma^2}{\lambda(1 - \eta\lambda/2)\mathbb{I}_d}\right)$, then*

**(1.)** *both $A_{\text{Noisy-GD}}$ and $\bar{A}_{\text{Noisy-GD}}$ are $(q, \varepsilon_{\text{dp}})$-RDP for any $K_A, K_{\bar{A}} \geq 0$,*

**(2.)** *pair $(A_{\text{Noisy-GD}}, \bar{A}_{\text{Noisy-GD}})$ satisfies $(q, \varepsilon_{\text{dd}})$-data-deletion all non-adaptive $r$-requesters*

$$\text{if} \quad K_{\bar{A}} \geq 4\kappa \log \frac{\varepsilon_{\text{dp}}}{\varepsilon_{\text{dd}}}, \tag{9}$$

**(3.)** *and all models in $(\hat{\Theta}_i)_{i \geq 0}$ produced by $(A_{\text{Noisy-GD}}, \bar{A}_{\text{Noisy-GD}}, \mathcal{Q})$ on any $\mathcal{D}_0 \in \mathcal{X}^n$, where $\mathcal{Q}$ is any $r$-requester, have an excess empirical risk $\text{err}(\hat{\Theta}_i; \mathcal{D}_i) = O\left(\frac{qd}{\varepsilon_{\text{dp}} n^2}\right)$*

$$\text{if} \quad K_A \geq 4\kappa \log\left(\frac{\varepsilon_{\text{dp}} n^2}{4qd}\right), \quad \text{and} \quad K_{\bar{A}} \geq 4\kappa \log \max\left\{5\kappa, \frac{8\varepsilon_{\text{dp}} r^2}{qd}\right\}. \tag{10}$$

Proof of Theorem 6 can be found in Appendix G.4.

Our utility upper bound for data-deletion matches the theoretical lower bound of $\Omega(\min\{1, \frac{d}{\varepsilon^2 n^2}\})$ in Bassily et al. (2014) for the best attainable utility of $(\varepsilon, \delta)$-DP algorithms on Lipschitz, smooth, strongly-convex loss functions[3]. Thus, our data-deletion algorithm, $\bar{A}_{\text{Noisy-GD}}$, incurs no additional cost in utility but saves substantial computation costs. Our data-deletion algorithm offers a computation saving of $\Omega(n \log \min\{\frac{n}{r}, n\sqrt{\frac{\varepsilon_{\text{dd}}}{qd}}\})$ per request while guaranteeing privacy, deletion, and optimal utility and without unsafe internal data structures. This saving is better than all existing unlearning algorithms in literature that we know of, and we present a detailed comparison in Table 1.

Also, observe that for satisfying $(q, \varepsilon_{\text{dp}})$-RDP and $(q, \varepsilon_{\text{dd}})$-data-deletion for non-adaptive $r$-requesters, the number of iterations $K_{\bar{A}}$ needed is independent of the size, $r$, of the deletion batch, depending solely on the ratio $\frac{\varepsilon_{\text{dd}}}{\varepsilon_{\text{dp}}}$. However, the number of iterations required for ensuring optimal utility with differential privacy grows with $r$. As such, we highlight that when deletion batches are sufficiently large, i.e., $r \geq \sqrt{\frac{qd}{\varepsilon_{\text{dd}}}}$, ensuring optimal utility under $(q, \varepsilon_{\text{dp}})$-RDP results in $(q, \varepsilon_{\text{dd}})$-data-deletion for free, thus demonstrating synergy between privacy and data deletion.

## 5.2 Data-Deletion and Utility Guarantees Under Non-Convexity

We give the following guarantee for pair $(A_{\text{Noisy-GD}}, \bar{A}_{\text{Noisy-GD}})$ under non-convexity of loss $\ell(\theta; \mathbf{x})$.

**Theorem 7** (Accuracy, privacy, deletion, and computation tradeoffs). *Let constants $\lambda, \beta, L, \sigma^2, \eta > 0$, $q, B > 1$, and $0 < \varepsilon_{\text{dd}} \leq \varepsilon_{\text{dp}} < d$. Let the loss function $\ell(\theta; \mathbf{x})$ be $\frac{\sigma^2 \log(B)}{4}$-bounded, $L$-Lipschitz and $\beta$-smooth, the regularizer be $\mathbf{r}(\theta) = \frac{\lambda}{2} \|\theta\|_2^2$, and the weight initialization distribution be $\rho = \mathcal{N}\left(0, \frac{\sigma^2}{\lambda}\mathbb{I}_d\right)$. Then,*

**(1.)** *both $A_{\text{Noisy-GD}}$ and $\bar{A}_{\text{Noisy-GD}}$ are $(q, \varepsilon_{\text{dp}})$-RDP for any $\eta \geq 0$ and any $K_A, K_{\bar{A}} \geq 0$ if*

$$\sigma^2 \geq \frac{qL^2}{\varepsilon_{\text{dp}} n^2} \cdot \eta \max\{K_A, K_{\bar{A}}\}, \tag{11}$$

---

[3]Recall from Remark 1 that $(q, \varepsilon_{\text{dp}})$-RDP implies $(\varepsilon, \delta)$-DP for $q = 1 + \frac{2}{\varepsilon}\log(1/\delta)$ and $\varepsilon_{\text{dp}} = \varepsilon/2$. When $\varepsilon = \Theta(\log(1/\delta))$, one can evaluate that $\frac{q}{\varepsilon_{\text{dp}}} = \Theta(\frac{\log(1/\delta)}{\varepsilon^2})$.

Table 1: Comparison of the computation savings in gradient complexity per edit request along with requirement of secret states with prior unlearning algorithms. For a fair comparison, we require that each of them satisfy an $(\varepsilon, \delta)$-data-deletion guarantee (ignoring the statelessness requirement) and have the same excess empirical risk bound. Edit requests are non-adaptive and modify $r = 1$ record in $n$-sized databases. The models' dimension is $d$. We assume loss $\ell(\theta; \mathbf{x})$ to be convex, 1-Lipschitz, and $O(1)$-smooth, and $L2$ regularization constant to be $O(1)$.

| Unlearning Algorithm | Requires secret states? | Compute savings for $i$th edit |
|---|---|---|
| Noisy-m-A-SGD [Thm. 1, (Ullah et al., 2021)] | No | $\Omega\left(\sqrt{d}\left(1 - \frac{\sqrt{d}}{n}\right)\right)$ |
| Perturbed-GD [Thm. 9, (Neel et al., 2021)] | Yes | $\Omega\left(n \log\left(\frac{\varepsilon n}{\sqrt{d}}\right)\right)$ |
| Perturbed-GD [Thm. 28, (Neel et al., 2021)] | No | $\Omega\left(n \log\left(\frac{\varepsilon n}{\log^2(id)\sqrt{d}}\right)\right)$ |
| Noisy-GD [Thm. 6, Ours] | No | $\Omega\left(n \log \min\left\{n, \frac{\varepsilon n}{\sqrt{d}}\right\}\right)$ |

**(2.)** *pair* $(\mathrm{A}_{\textit{Noisy-GD}}, \bar{\mathrm{A}}_{\textit{Noisy-GD}})$ *satisfy* $(q, \varepsilon_{\mathrm{dd}})$-*data-deletion under all non-adaptive $r$-requesters for any $\sigma^2 > 0$, if learning rate is $\eta \leq \frac{\lambda \varepsilon_{\mathrm{dd}}}{64 d q B (\beta + \lambda)^2}$ and number of iterations satisfy*

$$K_{\mathrm{A}} \geq \frac{2B}{\lambda \eta} \log\left(\frac{q \log(B)}{\varepsilon_{\mathrm{dd}}}\right), \quad K_{\bar{\mathrm{A}}} \geq K_{\mathrm{A}} - \frac{2B}{\lambda \eta} \log\left(\frac{\log(B)}{2\left(\varepsilon_{\mathrm{dd}} + \frac{r}{n} \log(B)\right)}\right), \quad (12)$$

**(3.)** *and all models in sequence* $(\hat{\Theta}_i)_{i \geq 0}$ *output by* $(\mathrm{A}_{\textit{Noisy-GD}}, \bar{\mathrm{A}}_{\textit{Noisy-GD}}, \mathcal{Q})$ *on any* $\mathcal{D}_0 \in \mathcal{X}^n$, *where* $\mathcal{Q}$ *is an $r$-requester, satisfy* $\mathrm{err}(\hat{\Theta}_i; \mathcal{D}_i) = \tilde{O}\left(\frac{dq}{\varepsilon_{\mathrm{dp}} n^2} + \frac{1}{n}\sqrt{\frac{q \varepsilon_{\mathrm{dd}}}{\varepsilon_{\mathrm{dp}}}}\right)$ *when inequalities in* (12) *and* (11) *are equalities.*

Proof of Theorem 7 can be found in Appendix G.5. The Rényi DP result in **(1.)** is a restatement of Abadi et al. (2016, Theorem 1) (discussed further in Appendix G.3. We prove the deletion and utility results in **(2.)** and **(3.)** by building on recent rapid convergence results for Noisy-GD by Vempala & Wibisono (2019); Chewi et al. (2021). Specifically, we show bounds on Rényi divergence of generated models $\hat{\Theta}_i$ w.r.t. Gibbs distribution $\pi(\mathcal{D}_i) \propto \exp(-\mathcal{L}_{\mathcal{D}_i}/\sigma^2)$ for all $i \geq 0$.

Under non-convexity, all prior works on deletion have focused on empirical analysis for utility. As far as we know, we are the first to provide utility guarantees in this setting. Moreover, our non-convex utility bound exceeds the optimal privacy-preserving utility under convexity by only a factor of $\tilde{O}\left(\frac{1}{n}\sqrt{\frac{q\varepsilon_{\mathrm{dd}}}{\varepsilon_{\mathrm{dp}}}}\right)$, which becomes small for large databases or small deletion to privacy budget ratio.

Also, observe a strict computational benefit in using $\bar{\mathrm{A}}_{\text{Noisy-GD}}$ whenever the fraction of edited records in a single update request satisfies $\frac{r}{n} \leq \frac{1}{2} - \frac{\varepsilon_{\mathrm{dd}}}{\log B}$. In the deletion regime where $\varepsilon_{\mathrm{dd}} = \log(B)/4$, relying on $\bar{\mathrm{A}}_{\text{Noisy-GD}}$ rather than retraining with $\mathrm{A}_{\text{Noisy-GD}}$ is $\Omega(dn \log \frac{n}{r})$ cheaper.

**Remark 4.** *The assumption that* $\ell(\theta; \mathbf{x})$ *is L-Lipschitz in both Theorems 6 and 7 can be removed if gradients* $\nabla \ell(\Theta_{\eta k}; \mathbf{x})$ *computed in step 2 of Algorithm 1 are clipped to L (discussed in Appendix C).*

## 6 CONCLUSIONS

We showed that unlearning guarantees are an unsuitable data deletion measure in the online setting of adaptive edit requests and proposed a proper notion of data deletion in line with the "right to be forgotten." We also showed that differential privacy is necessary for data deletion when requests are adaptive and proved a reduction from secure deletion under non-adaptive requests to adaptive requests under DP. Our theoretical results on Noisy-GD based data-deletion, for both convex and non-convex losses, show a substantial computation saving over retraining with a reliable deletion guarantee and at no additional cost in utility.

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

# Appendix

## Table of Contents

# A    TABLE OF NOTATIONS

Table 2: Symbol reference

| Symbol | Meaning |
| --- | --- |
| $\mathcal{O}$ | Arbitrary model parameter space. |
| $\Phi$ | Space of publishable outcomes. |
| $d$ | Dimension of model parameters. |
| $\mathbb{R}^d$ | Space of $d$-dimensional model parameters. |
| $n$ | Database size. |
| $\mathcal{X}, \mathcal{X}^n$ | Data universe and Domain of all datasets of size $n$. |
| $\nu, \nu', \pi, \mu$ | Arbitrary distributions on $\mathcal{O}$ or on $\mathbb{R}^d$. |
| $\mathcal{Q}$ | An edit requester. |
| $r, p$ | Number of records replaced per edit and number of releases observable by a requester. |
| $\mathcal{U}, \mathcal{U}^r$ | Space of singular and batched replacement edits in $[n] \times \mathcal{X}$. |
| $u, u_i, U_i$ | Arbitrary edit request, $i^{th}$ edit request in $\mathcal{U}^r$ and its random variable. |
| $\mathcal{D}, \mathcal{D}_i$ | An example database and database after $i^{th}$ update. |
| $\mathbf{x}, \mathbf{y}$ | Singular data records from universe $\mathcal{X}$. |
| $\eta$ | Step size or learning rate in Noisy-GD. |
| $\sigma^2$ | Variance scaling used in weight initialization distribution or gradient noise. |
| $\ell(\theta; \mathbf{x})$ | Twice continuously differentiable loss function on models in $\mathbb{R}^d$. |
| $\mathbf{r}(\theta)$ | $L2$ regularizer $\lambda \|\theta\|_2^2 / 2$. |
| $\mathcal{L}(\theta), \mathcal{L}_\mathcal{D}(\theta)$ | Arbitrary optimization objective and an $\mathbf{r}(\theta)$ regularized objective on $\mathcal{D}$ over $\ell(\theta; \mathbf{x})$. |
| $\mathrm{err}(\Theta; \mathcal{D})$ | Excess empirical risk of random model $\Theta$ over objective $\mathcal{L}_\mathcal{D}$. |
| $\pi(\mathcal{D})$ | An mapping from $\mathcal{X}^n$ to distributions on $\mathbb{R}^d$; sometimes distributions are Gibbs. |
| $\Lambda_\mathcal{D}$ | Normalization constant of the Gibbs distribution $\pi(\mathcal{D})$. |
| $\pi_i^u$ | A distribution independent of record deleted by request $u$ on database $\mathcal{D}_{i-1}$. |
| $T_k$ | A map over $\mathbb{R}^d$. |
| $\rho$ | Weight initialization distribution for Noisy-GD. |
| $\mathbf{v}, \mathbf{v}'$ | Vector fields on $\mathbb{R}^d$. |
| $\theta_\mathcal{D}^*, \theta_{\mathcal{D}_i}^*$ | Risk minimizer for $\mathcal{L}_\mathcal{D}$ and $\mathcal{L}_{\mathcal{D}_i}$. |
| $q$ | Order of Rényi divergence. |
| $\varepsilon_{\mathrm{dp}}, \varepsilon_{\mathrm{dd}}$ | Differential privacy budget and data-deletion budget in $q$-Rényi divergence. |
| $\varepsilon, \delta$ | Parameters for DP-like indistinguishability. |
| $\mathrm{A}, \mathrm{A}_{\text{Noisy-GD}}$ | Learning algorithm and Noisy-GD based learning algorithm respectively. |
| $\bar{\mathrm{A}}, \bar{\mathrm{A}}_{\text{Noisy-GD}}$ | Data-deletion algorithm and Noisy-GD based data-deletion algorithm respectively. |
| $K_\mathrm{A}, K_{\bar{\mathrm{A}}}$ | Number of learning and data-deletion iterations in Noisy-GD. |
| $k, t$ | Index of a Noisy-GD iteration and continuous time variable for tracing diffusions. |
| $\Theta_{\eta k}, \Theta'_{\eta k}$ | Random variables representing the iteration $k$ parameter distribution in Noisy-GD. |
| $\Theta_t, \Theta'_t$ | Random variables representing the time $t$ distribution of tracing diffusion for Noisy-GD. |
| $\mu_t, \mu'_t$ | Probability density for $\Theta_t, \Theta'_t$. |
| $\mathbb{I}_d$ | $d$-dimensional identity matrix. |
| $\mathbf{Z}, \mathbf{Z}_k, \mathbf{Z}'_k$ | Random variables taken from $\mathcal{N}(0, \mathbb{I}_d)$. |
| $\mathrm{d}\mathbf{Z}_t, \mathrm{d}\mathbf{Z}'_t$ | Two independent Weiner process. |
| $\lambda, \beta, B, L$ | $L2$ regularizer constant and smoothness, boundedness, and Lipschitzness constants. |
| $\mathrm{Clip}_L(\cdot)$ | Operator that clips vectors in $\mathbb{R}^d$ to a magnitude of $L$. |
| $\mathrm{R}_q(\nu\|\nu')$ | Rényi divergence of distribution of $\nu$ w.r.t $\nu'$. |
| $\mathrm{E}_q(\nu\|\nu')$ | $q^{\text{th}}$ moment of likelihood ratio r.v. between $\nu$ and $\nu'$. |
| $\mathrm{I}(\nu\|\nu'), \mathrm{I}_q(\nu\|\nu')$ | Fisher and $q$-Rényi Information of distribution of $\nu$ w.r.t $\nu'$. |
| $\mathrm{W}_2(\nu, \nu')$ | Wasserstein distance between distribution $\nu$ and $\nu'$. |
| $\mathrm{KL}(\nu\|\nu')$ | Kullback Leibler divergence of distribution $\nu$ w.r.t. $\nu'$. |
| $P_t, \mathcal{G}, \mathcal{G}^*$ | Markov semigroup, its infintesimal generator, and its Fokker-Planck operator. |
| $\mathrm{Ent}_\pi(f^2)$ | Entropy of function $f^2$ under any arbitrary distribution $\pi$. |
| $\mathrm{H}(\cdot)$ | Differential entropy of a distribution. |
| $\mathrm{LS}(c)$ | Log-sobolev inequality with constant $c$. |
| $\mathrm{prox}_\mathcal{L}$ | Proxmial mapping for objective $\mathcal{L}$. |

## B  CALCULUS REFRESHER

Given a twice continuously differentiable function $\mathcal{L} : \mathcal{O} \to \mathbb{R}$, where $\mathcal{O}$ is a closed subset of $\mathbb{R}^d$, its gradient $\nabla\mathcal{L} : \mathcal{O} \to \mathbb{R}^d$ is the vector of partial derivatives

$$\nabla\mathcal{L}(\theta) = \left( \frac{\partial\mathcal{L}(\theta)}{\partial\theta_1}, \cdots, \frac{\partial\mathcal{L}(\theta)}{\partial\theta_2} \right). \tag{13}$$

Its Hessian $\nabla^2\mathcal{L} : \mathcal{O} \to \mathbb{R}^{d\times d}$ is the matrix of second partial derivatives

$$\nabla^2\mathcal{L}(\theta) = \left( \frac{\partial^2\mathcal{L}(\theta)}{\partial\theta_i\theta_j} \right)_{1\leq i,j\leq d}. \tag{14}$$

Its Laplacian $\Delta\mathcal{L} : \mathcal{O} \to \mathbb{R}$ is the trace of its Hessian $\nabla^2\mathcal{L}$, i.e.,

$$\Delta\mathcal{L}(\theta) = \mathsf{Tr}\left( \nabla^2\mathcal{L}(\theta) \right). \tag{15}$$

Given a differentiable vector field $\mathbf{v} = (\mathbf{v}_1, \cdots, \mathbf{v}_d) : \mathcal{O} \to \mathbb{R}^d$, its divergence $\mathrm{div}\,(\mathbf{v}) : \mathcal{O} \to \mathbb{R}$ is

$$\mathrm{div}\,(\mathbf{v})\,(\theta) = \sum_{i=1}^d \frac{\partial\mathbf{v}_i(\theta)}{\partial\theta_i}. \tag{16}$$

Some identities that we would rely on:

1. Divergence of gradient is the Laplacian, i.e.,

$$\mathrm{div}\,(\nabla\mathcal{L})\,(\theta) = \sum_{i=1}^d \frac{\partial^2\mathcal{L}(\theta)}{\partial\theta_i^2} = \Delta\mathcal{L}(\theta). \tag{17}$$

2. For any function $f : \mathcal{O} \to \mathbb{R}$ and a vector field $\mathbf{v} : \mathcal{O} \to \mathbb{R}^d$ with sufficiently fast decay at the border of $\mathcal{O}$,

$$\int_{\mathcal{O}} \langle\mathbf{v}(\theta), \nabla f(\theta)\mathrm{d}\theta\rangle = -\int_{\mathcal{O}} f(\theta)(\mathrm{div}\,(\mathbf{v}))(\theta)\mathrm{d}\theta. \tag{18}$$

3. For any two functions $f, g : \mathcal{O} \to \mathbb{R}$, out of which atleast for one the gradient decays sufficinetly fast at the border of $\mathcal{O}$, the following also holds.

$$\int_{\mathcal{O}} f(\theta)\Delta g(\theta)\mathrm{d}\theta = -\int_{\mathcal{O}} \langle\nabla f(\theta), \nabla g(\theta)\rangle\,\mathrm{d}\theta = \int_{\mathcal{O}} g(\theta)\Delta f(\theta)\mathrm{d}\theta. \tag{19}$$

4. Based on Young's inequality, for two vector fields $\mathbf{v}_1, \mathbf{v}_2 : \mathcal{O} \to \mathbb{R}^d$, and any $a, b \in \mathbb{R}$ such that $ab = 1$, the following inequality holds.

$$\langle\mathbf{v}_1, \mathbf{v}_2\rangle\,(\theta) \leq \frac{1}{2a}\|\mathbf{v}_1(\theta)\|_2^2 + \frac{1}{2b}\|\mathbf{v}_2(\theta)\|_2^2. \tag{20}$$

Wherever it is clear, we would drop $(\theta)$ for brevity. For example, we would represent $\mathrm{div}\,(\mathbf{v})\,(\theta)$ as only $\mathrm{div}\,(\mathbf{v})$.

## C  LOSS FUNCTION PROPERTIES

In this section, we provide the formal definition of various properties that we assume in the paper. Let $\ell(\theta; \mathbf{x}) : \mathbb{R}^d \times \mathcal{X} \to \mathbb{R}$ be a loss function on $\mathbb{R}^d$ for any record $\mathbf{x} \in \mathcal{X}$.

**Definition C.1** (Lipschitzness). *A function $\ell(\theta; \mathbf{x})$ is said to be $L$ Lipschitz continuous if for all $\theta, \theta' \in \mathbb{R}^d$ and any $\mathbf{x} \in \mathcal{X}$,*

$$|\ell(\theta; \mathbf{x}) - \ell(\theta'; \mathbf{x})| \leq L\|\theta - \theta'\|_2. \tag{21}$$

*If $\ell(\theta; \mathbf{x})$ is differentiable, then it is $L$-Lipschitz if and only if $\nabla\ell(\theta; \mathbf{x}) \leq L$ for all $\theta \in \mathbb{R}^d$.*

**Definition C.2** (Boundedness). *A function $\ell(\theta; \mathbf{x})$ is said to be $B$-bounded if for all $\mathbf{x} \in \mathcal{X}$, its output takes values in range $[-B, B]$.*

**Definition C.3** (Convexity). *A continuous differential function $\ell(\theta; \mathbf{x})$ is said to be convex if for all $\theta, \theta' \in \mathbb{R}^d$ and $\mathbf{x} \in \mathcal{X}$,*

$$\ell(\theta'; \mathbf{x}) \geq \ell(\theta; \mathbf{x}) + \langle \nabla \ell(\theta; \mathbf{x}), \theta' - \theta \rangle, \tag{22}$$

*and is said to be $\lambda$-strongly convex if*

$$\ell(\theta'; \mathbf{x}) \geq \ell(\theta; \mathbf{x}) + \langle \nabla \ell(\theta; \mathbf{x}), \theta' - \theta \rangle + \frac{\lambda}{2} \|\theta' - \theta\|_2^2. \tag{23}$$

**Theorem 8** ((Nesterov, 2003, Theorem 2.1.4)). *A twice continuously differentiable function $\ell(\theta; \mathbf{x})$ is convex if and only if for all $\theta \in \mathbb{R}^d$ and $\mathbf{x} \in \mathcal{X}$, its hessian matrix $\nabla^2 \ell(\theta; \mathbf{x})$ is positive semidefinite, i.e., $\nabla^2 \ell(\theta; \mathbf{x}) \succcurlyeq 0$ and is $\lambda$-strongly convex if its hessian matrix satisfies $\nabla^2 \ell(\theta; \mathbf{x}) \succcurlyeq \lambda \mathbb{I}_d$.*

**Definition C.4** (Smoothness). *A continuously differentiable function $\ell(\theta; \mathbf{x})$ is said to be $\beta$-Smooth if for all $\theta, \theta' \in \mathbb{R}^d$ and $\mathbf{x} \in \mathcal{X}$,*

$$\|\nabla \ell(\theta; \mathbf{x}) - \nabla \ell(\theta'; \mathbf{x})\|_2 \leq \beta \|\theta - \theta'\|_2. \tag{24}$$

**Theorem 9** ((Nesterov, 2003, Theorem 2.1.6)). *A twice continuously differentiable convex function $\ell(\theta; \mathbf{x})$ is $\beta$-smooth if and only if for all $\theta \in \mathbb{R}^d$ and $\mathbf{x} \in \mathcal{X}$,*

$$\nabla^2 \ell(\theta; \mathbf{x}) \preccurlyeq \beta \mathbb{I}_d. \tag{25}$$

### C.0.1 Effect of Gradient Clipping

First order optimization methods on a continuously differentiable loss function $\ell(\theta; \mathbf{x})$ over a database $\mathcal{D} \in \mathcal{X}^n$ with gradient clipping $\mathrm{Clip}_L(\mathbf{v}) = \mathbf{v} / \max\left(1, \frac{\|\mathbf{v}\|_2}{L}\right)$ is equivalent to optimizing

$$\mathcal{L}_\mathcal{D}(\theta) = \frac{1}{|\mathcal{D}|} \sum_{\mathbf{x} \in \mathcal{D}} \bar{\ell}(\theta; \mathbf{x}) + \mathbf{r}(\theta), \tag{26}$$

where $\bar{\ell}(\theta; \mathbf{x})$ is a surrogate loss function that satisfies $\nabla \bar{\ell}(\theta; \mathbf{x}) = \mathrm{Clip}_L(\nabla \ell(\theta; \mathbf{x}))$. This surrogate loss function inherits convexity, boundedness, and smoothness properties of $\ell(\theta; \mathbf{x})$, as shown below.

**Lemma 10** (Gradient clipping retains convexity). *If $\ell(\theta; \mathbf{x})$ is a twice continuously differentiable convex function for every $\mathbf{x} \in \mathbb{R}^d$, then surrogate loss $\bar{\ell}(\theta; \mathbf{x})$ resulting from gradient clipping is also convex for every $\mathbf{x} \in \mathbb{R}^d$.*

*Proof.* Note that the clip operation $\mathrm{Clip}_L(\mathbf{v})$ is a closed-form solution of the orthogonal projection onto a closed ball of radius $L$ and centered around origin, i.e.

$$\mathrm{Clip}_L(\mathbf{v}) = \arg\min_{\|\mathbf{v}'\|_2 \leq L} \|\mathbf{v} - \mathbf{v}'\|_2. \tag{27}$$

By properties of orthogonal projections on closed convex sets, for every $\mathbf{v}, \mathbf{v}' \in \mathbb{R}^d$,

$$\langle \mathbf{v}' - \mathrm{Clip}_L(\mathbf{v}), \mathbf{v} - \mathrm{Clip}_L(\mathbf{v}) \rangle \leq 0 \quad \text{if and only if } \|\mathbf{v}'\|_2 \leq L. \tag{28}$$

Therefore, for any $\theta \in \mathbb{R}^d$, and $\mathbf{x} \in \mathcal{X}$, we have

$$\langle \nabla \bar{\ell}(\theta + h\hat{\mathbf{v}}; \mathbf{x}) - \nabla \bar{\ell}(\theta; \mathbf{x}), \nabla \ell(\theta; \mathbf{x}) - \nabla \bar{\ell}(\theta; \mathbf{x}) \rangle \leq 0, \tag{29}$$

$$\langle \nabla \bar{\ell}(\theta; \mathbf{x}) - \nabla \bar{\ell}(\theta + h\hat{\mathbf{v}}; \mathbf{x}), \nabla \ell(\theta + h\hat{\mathbf{v}}; \mathbf{x}) - \nabla \bar{\ell}(\theta + h\hat{\mathbf{v}}; \mathbf{x}) \rangle \leq 0, \tag{30}$$

for all unit vectors $\hat{\mathbf{v}} \in \mathbb{R}^d$ and magnitude $h > 0$. For the directional derivative of vector field $\nabla \bar{\ell}(\theta; \mathbf{x})$ along $\hat{\mathbf{v}}$, defined as $\nabla_{\hat{\mathbf{v}}} \nabla \bar{\ell}(\theta; \mathbf{x}) = \lim_{h \to 0^+} \frac{\nabla \bar{\ell}(\theta + h\hat{\mathbf{v}}; \mathbf{x}) - \nabla \bar{\ell}(\theta; \mathbf{x})}{h}$, the above two inequalities imply

$$\langle \nabla_{\hat{\mathbf{v}}} \nabla \bar{\ell}(\theta; \mathbf{x}), \nabla \ell(\theta; \mathbf{x}) - \nabla \bar{\ell}(\theta; \mathbf{x}) \rangle = 0, \tag{31}$$

for all $\hat{\mathbf{v}}$. Therefore, when $\nabla \bar{\ell}(\theta; \mathbf{x}) \neq \nabla \ell(\theta; \mathbf{x})$, we must have $\nabla^2 \bar{\ell}(\theta; \mathbf{x}) = 0$. And, when $\nabla \bar{\ell}(\theta; \mathbf{x}) = \nabla \ell(\theta; \mathbf{x})$, gradients aren't clipped, which implies the rate of change of $\bar{\ell}(\theta; \mathbf{x})$ along any direction $\hat{\mathbf{v}}$ is

$$\nabla_{\hat{\mathbf{v}}} \cdot \nabla \bar{\ell}(\theta; \mathbf{x}) = \lim_{h \to 0^+} \left\langle \frac{\nabla \bar{\ell}(\theta + h\hat{\mathbf{v}}; \mathbf{x}) - \nabla \bar{\ell}(\theta; \mathbf{x})}{h}, \hat{\mathbf{v}} \right\rangle$$

$$= \begin{cases} \hat{\mathbf{v}}^\top \nabla^2 \ell(\theta; \mathbf{x}) \hat{\mathbf{v}} & \text{if } \exists h > 0 \text{ s.t. } \nabla \bar{\ell}(\theta + h\hat{\mathbf{v}}; \mathbf{x}) = \nabla \ell(\theta + h\hat{\mathbf{v}}; \mathbf{x}) \\ 0 & \text{otherwise} \end{cases} \geq 0.$$

$\square$

**Lemma 11** (Gradient clipping retains boundedness). *If $\ell(\theta; \mathbf{x})$ is a continuously differentiable and $B$-bounded function for every $\mathbf{x} \in \mathcal{X}$, then the surrogate loss $\bar{\ell}(\theta; \mathbf{x})$ resulting from gradient clipping is also $B$-bounded.*

*Proof.* Since $\ell(\theta; \mathbf{x})$ is continuously differentiable, its $B$-boundedness implies path integral of $\nabla \ell(\theta; \mathbf{x})$ along any curve between $\theta, \theta' \in \mathbb{R}^d$ is less than $2B$. Since $\text{Clip}_L(\cdot)$ operation clips the gradient magnitude, the path integral of $\nabla \bar{\ell}(\theta; \mathbf{x})$ is also less than $2B$. That is, the maximum and minimum values that $\bar{\ell}(\theta; \mathbf{x})$ takes differ no more than $2B$. By adjusting the constant of path integral, we can always ensure $\bar{\ell}(\theta; \mathbf{x})$ takes values in range $[-B, B]$ without affecting first order optimization algorithms. $\qquad\square$

**Lemma 12** (Gradient clipping retains smoothness). *If $\ell(\theta; \mathbf{x})$ is a continuously differentiable and $\beta$-smooth function for every $\mathbf{x} \in \mathbb{R}^d$, then surrogate loss $\bar{\ell}(\theta; \mathbf{x})$ resulting from gradient clipping is also $\beta$-smooth for every $\mathbf{x} \in \mathbb{R}^d$.*

*Proof.* Note that the gradient clipping operation is equivalent to an orthogonal projection operation into a ball of radius $L$, i.e. $\text{Clip}_L(\mathbf{v}) = \arg\min_{\mathbf{v}'}\{\|\mathbf{v}' - \mathbf{v}\|_2 : \mathbf{v} \in \mathbb{R}^d, \|\mathbf{v}'\|_2 \le L\}$. Since orthogonal projection onto a closed convex set is a 1-Lipschitz operation, for any $\theta, \theta' \in \mathbb{R}^d$,

$$\left\| \nabla \bar{\ell}(\theta; \mathbf{x}) - \nabla \bar{\ell}(\theta'; \mathbf{x}) \right\|_2 \le \left\| \nabla \ell(\theta; \mathbf{x}) - \nabla \ell(\theta'; \mathbf{x}) \right\|_2 \le \beta \left\| \theta - \theta' \right\|_2. \tag{32}$$

$\qquad\square$

Additionally, the surrogate loss $\bar{\ell}(\theta; \mathbf{x})$ is twice differentiable almost everywhere if $\ell(\theta; \mathbf{x})$ is smooth, which follows from the following Rademacher's Theorem.

**Theorem 13** (Rademacher's Theorem (Nekvinda & Zajíček, 1988)). *If $f : \mathbb{R}^n \to \mathbb{R}^n$ is Lipschitz continuous, then $f$ is differentiable almost everywhere in $\mathbb{R}^n$.*

All our results in Section 5 rely on the above four properties on losses and therefore apply with gradient clipping instead of the Lipschitzness assumption.

# D   DIVERGENCE MEASURES AND THEIR PROPERTIES

Let $\Theta, \Theta' \in \mathcal{O}$ be two random variables with probability measures $\nu, \nu'$ respectively. We abuse the notataions to denote respective probability densities with $\nu, \nu'$ as well. We say that $\nu$ is absolutely continuous with respect to $\nu'$ (denoted by $\nu \ll \nu'$) if for all measurable sets $O \subset \mathcal{O}$, $\nu(O) = 0$ whenever $\nu'(O) = 0$.

**Definition D.1** (($\varepsilon, \delta$)-indistinguishability (Dwork et al., 2014)). *We say $\nu$ and $\nu'$ are $(\varepsilon, \delta)$-indistinguishable if for all $O \subset \mathcal{O}$,*

$$\mathbb{P}_{\Theta \sim \nu}[\Theta \in O] \le e^\varepsilon \mathbb{P}_{\Theta' \sim \nu'}[\Theta' \in O] + \delta \quad \text{and} \quad \mathbb{P}_{\Theta' \sim \nu'}[\Theta' \in O] \le e^\varepsilon \mathbb{P}_{\Theta \sim \nu}[\Theta \in O] + \delta. \tag{33}$$

In this paper, we measure indistinguishability in terms of Rényi divergence.

**Definition D.2** (Rényi divergence (Rényi et al., 1961)). *Rényi divergence of $\nu$ w.r.t. $\nu'$ of order $q > 1$ is defined as*

$$\mathrm{R}_q(\nu\|\nu') = \frac{1}{q-1} \log \mathrm{E}_q(\nu\|\nu'), \quad \text{where} \quad \mathrm{E}_q(\nu\|\nu') = \mathbb{E}_{\theta \sim \nu'}\left[\left(\frac{\nu(\theta)}{\nu'(\theta)}\right)^q\right], \tag{34}$$

*when $\nu$ is absolutely continuous w.r.t. $\nu'$ (denoted as $\nu \ll \nu'$). If $\nu \not\ll \nu'$, we'll say $\mathrm{R}_q(\nu\|\nu') = \infty$. We abuse the notation $\mathrm{R}_q(\Theta\|\Theta')$ to denote divergence $\mathrm{R}_q(\nu\|\nu')$ between the measures of $\Theta, \Theta'$.*

A bound on Rényi divergence implies a one-directional $(\varepsilon, \delta)$-indistinguishability as described below.

**Theorem 14** (Conversion theorem of Rényi divergence (Mironov, 2017, Proposition 3)). *Let $q > 1$ and $\varepsilon > 0$. If distributions $\nu, \nu'$ satisfy $\mathrm{R}_q(\nu\|\nu') < \varepsilon_0$, then for any $O \subset \mathcal{O}$,*

$$\mathbb{P}_{\Theta \sim \nu}[\Theta \in O] \le e^\varepsilon \mathbb{P}_{\Theta' \sim \nu'}[\Theta' \in O] + \delta, \tag{35}$$

*for $\varepsilon = \varepsilon_0 + \frac{\log 1/\delta}{q-1}$ and any $0 < \delta < 1$.*

We use the following properties of Rényi divergence in some of our proofs.

**Theorem 15** (Mononicity of Rényi divergence (Mironov, 2017, Proposition 9)). *For $1 \leq q_0 < q$, and arbitrary probability measures $\nu$ and $\nu'$ over $\mathcal{O}$, $\mathrm{R}_{q_0}(\nu\|\nu') \leq \mathrm{R}_q(\nu\|\nu')$.*

**Theorem 16** (Rényi composition (Mironov, 2017, Proposition 1)). *If $\mathrm{A}_1, \cdots, \mathrm{A}_k$ are randomized algorithms satisfying, respectively, $(q, \varepsilon_1)$-RDP, $\cdots$, $(q, \varepsilon_k)$-RDP then their composed mechanism defined as $(\mathrm{A}_1(\mathcal{D}), \cdots, \mathrm{A}_k(\mathcal{D}))$ is $(q, \varepsilon_1 + \cdots + \varepsilon_k)$-RDP. Moreover, $i^{th}$ algorithm can be chosen on the basis of the outputs of algorithms $\mathrm{A}_1, \cdots, \mathrm{A}_{i-1}$.*

**Theorem 17** (Weak triangle inequality of Rényi divergence (Mironov, 2017, Proposition 12)). *For any distribution $\rho$ on $\mathcal{O}$, the Rényi divergence of $\nu$ w.r.t. $\nu'$ satisfies the following weak triangle inequality:*

$$\mathrm{R}_q(\nu\|\nu') \leq \mathrm{R}_q(\nu\|\rho) + \mathrm{R}_\infty(\rho\|\nu'). \tag{36}$$

Another popular notion of information divergence is the Kullback-Leibler divergence.

**Definition D.3** (Kullback-Leibler divergence (Kullback & Leibler, 1951)). *Kullback-Leibler (KL) divergence $\mathrm{KL}(\nu\|\nu')$ of $\nu$ w.r.t. $\nu'$ is defined as*

$$\mathrm{KL}(\nu\|\nu') = \mathop{\mathbb{E}}_{\theta \sim \nu}\left[\log \frac{\nu(\theta)}{\nu'(\theta)}\right]. \tag{37}$$

Rényi divergence generalizes Kullback-Leibler divergence (Van Erven & Harremos, 2014) as $\lim_{q \to 1} \mathrm{R}_q(\nu\|\nu') = \mathrm{KL}(\nu\|\nu')$.

Some other divergence notions that we rely on are the following.

**Definition D.4** (Wasserstein distance (Vaserstein, 1969)). *Wasserstein distance between $\nu$ and $\nu'$ is*

$$\mathrm{W}_2(\nu, \nu') = \inf_\Pi \mathop{\mathbb{E}}_{\Theta, \Theta' \sim \Pi}\left[\|\Theta - \Theta'\|_2^2\right]^{\frac{1}{2}}, \tag{38}$$

*where $\Pi$ is any joint distribution on $\mathcal{O} \times \mathcal{O}$ with $\nu$ and $\nu'$ as its marginal distributions.*

**Definition D.5** (Relative Fisher information (Otto & Villani, 2000)). *If $\nu \ll \nu'$ and $\frac{\nu}{\nu'}$ is differentiable, then relative Fisher information of $\nu$ with respect to $\nu'$ is defined as*

$$\mathrm{I}(\nu\|\nu') = \mathop{\mathbb{E}}_{\theta \sim \nu}\left[\left\|\nabla \log \frac{\nu(\theta)}{\nu'(\theta)}\right\|_2^2\right]. \tag{39}$$

**Definition D.6** (Relative Rényi information (Vempala & Wibisono, 2019)). *Let $q > 1$. If $\nu \ll \nu'$ and $\frac{\nu}{\nu'}$ is differentiable, then relative Rényi information of $\nu$ with respect to $\nu'$ is defined as*

$$\mathrm{I}_q(\nu\|\nu') = \frac{4}{q^2} \mathop{\mathbb{E}}_{\theta \sim \nu'}\left[\left\|\nabla \left(\frac{\nu(\theta)}{\nu'(\theta)}\right)^{q/2}\right\|_2^2\right] = \mathop{\mathbb{E}}_{\theta \sim \nu'}\left[\left(\frac{\nu(\theta)}{\nu'(\theta)}\right)^{q-2}\left\|\nabla \left(\frac{\nu(\theta)}{\nu'(\theta)}\right)\right\|_2^2\right]. \tag{40}$$

# E    PROOFS FOR SECTION 3

**Theorem 1.** *For every $\varepsilon > 0$, there exists a pair $(\mathrm{A}, \bar{\mathrm{A}})$ of algorithms that satisfy $(\varepsilon, 0)$-unlearning under a publish function $f_{\mathrm{pub}}$ such that for all non-adaptive 1-requesters $\mathcal{Q}$, their exists an adversary that can correctly infer the identity of a record deleted at any arbitrary edit step $i \geq 1$ by observing only the post-edit releases $\phi_{\geq i} = (\phi_i, \phi_{i+1}, \cdots)$.*

*Proof.* For a query $h : \mathcal{X} \to \{0, 1\}$, consider the task of learning the count over a database that is being edited online by a non-adaptive 1-requester $\mathcal{Q}$. Since $\mathcal{Q}$ is non-adaptive by assumption, it is equivalent to the entire edit sequence $\{u_i\}_{i \geq 1}$ being fixed before interaction. We design an algorithm pair $(\mathrm{A}, \bar{\mathrm{A}})$ for this task with secret model space being $\mathcal{O} = \mathbb{N}^3$ and published outcome space being $\Phi = \mathbb{R}$, with the publish function being $f_{\mathrm{pub}}(\langle a, b, c \rangle) = a + b/c + \mathrm{Lap}\left(\frac{1}{\varepsilon}\right)$ (with the convention that $b/c = 0$ if $b = c = 0$). At any step $i \geq 0$, our internal model $\hat{\Theta}_i = \langle \mathrm{cnt}_i, \mathrm{del}_i, i \rangle$ encodes the current count of $h$ on database $\mathcal{D}_i$, the count of $h$ on records previously deleted by $u_{\leq i}$, and the current step index $i$. Our learning algorithm initializes the secret model as $\hat{\Theta}_0 = \mathrm{A}(\mathcal{D}_0) = \langle \sum_{\mathbf{x} \in \mathcal{D}_0} h(\mathbf{x}), 0, 0 \rangle$,

and, for an edit request $u_i = \{\langle \text{ind}_i, \mathbf{y}_i \rangle\}$, our algorithm $\bar{A}$ updates the secret model $\hat{\Theta}_{i-1} \to \hat{\Theta}_i$ following the rule

$$\hat{\Theta}_i = \bar{A}(\mathcal{D}_{i-1}, u_i, \hat{\Theta}_{i-1}) = \langle \text{cnt}_i, \text{del}_i, i \rangle \text{ where } \begin{cases} \text{cnt}_i = \text{cnt}_{i-1} + h(\mathbf{y}_i) - h(\mathcal{D}_{i-1}[\text{ind}_i]), \\ \text{del}_i = \text{del}_{i-1} + h(\mathcal{D}_{i-1}[\text{ind}_i]). \end{cases}$$

Note that $\forall i \geq 1$, $\Delta_i \stackrel{\text{def}}{=} \text{del}_i / i \in [0, 1]$. Therefore, from properties of Laplace mechanism (Dwork et al., 2014), it is straightforward to see that for all $i \geq 1$,

$$f_{\text{pub}}(\bar{A}(\mathcal{D}_{i-1}, u_i, \hat{\Theta}_{i-1}))\big|u_{\leq i} = \sum_{\mathbf{x} \in \mathcal{D}_i} h(\mathbf{x}) + \Delta_i + \text{Lap}\left(\frac{1}{\varepsilon}\right)$$

$$\stackrel{\varepsilon, 0}{\approx} \sum_{\mathbf{x} \in \mathcal{D}_i} h(\mathbf{x}) + \text{Lap}\left(\frac{1}{\varepsilon}\right) = f_{\text{pub}}(A(\mathcal{D}_i)).$$

Hence, $\bar{A}$ is an $(\varepsilon, 0)$-unlearning algorithm for $A$ under $f_{\text{pub}}$.

To show that an adversary can still infer the identity of record deleted by update $u_i = (\text{ind}_i, \bullet)$, consider a database $\mathcal{D}'_{i-1}$ that differs from $\mathcal{D}_{i-1}$ only at index $\text{ind}_i$ such that $h(\mathcal{D}'_{i-1}[\text{ind}_i]) \neq h(\mathcal{D}_{i-1}[\text{ind}_i])$. Let random variable sequences $\phi_{\geq i}$ and $\phi'_{\geq i}$ denote the releases by $\bar{A}$ in the scenarios that the $(i-1)^{\text{th}}$ database was $\mathcal{D}_{i-1}$ and $\mathcal{D}'_{i-1}$ respectively. The divergence between these two random variable sequences reflect the capacity of any adversary to infer the record deleted by $u_i$. Since, we have identical databases after $u_i$, i.e. $\mathcal{D}_{j-1} \circ u_j = \mathcal{D}'_{j-1} \circ u_j$ for all $j \geq i$, note that both $\phi_j$ and $\phi'_j$ are independent Laplace distributions with a shift of exactly $\frac{1}{j}$ units. Therefore,

$$\max_{O \subset \Phi^*} \log \frac{\mathbb{P}[\phi_{\geq i} \in O]}{\mathbb{P}[\phi'_{\geq i} \in O]} = \sum_{j=i}^{\infty} \max_{O_j \subset \mathbb{R}} \log \frac{\mathbb{P}[\phi_j \in O_j]}{\mathbb{P}[\phi'_j \in O_j]} = \sum_{j=i}^{\infty} \log e^{\varepsilon/j} = \infty.$$

$\square$

**Theorem 2.** *There exists a pair $(A, \bar{A})$ of algorithms that satisfy $(0, 0)$-adaptive-unlearning for an identity publish function $f_{\text{pub}}(\theta) = \theta$ such that by designing a 1-adaptive 1-requester $\mathcal{Q}$, an adversary, even with no access to $\mathcal{Q}$'s interaction transcript, can infer the identity of a record deleted at any arbitrary edit step $i > 3$ with probability at-least $1 - (1/2)^{i-3}$ from the post-edit release $\phi_i$.*

*Proof.* Let data universe be $\mathcal{X}$, the internal state space $\mathcal{O}$, and publishable outcome space $\Phi$ be $\mathbb{R}$. Consider the task of releasing a sequence of medians using function $\text{med} : \mathbb{R}^* \to \mathbb{R}$ in the online setting when the initial database $\mathcal{D}_0 \in \mathcal{X}^n$ is being modified by some adaptive requester $\mathcal{Q}$. Given a database $\mathcal{D} \in \mathcal{X}^n$, our learning algorithm is defined as $A(\mathcal{D}) = \text{med}(\mathcal{D})$. For an arbitrary edit request $u \in \mathcal{U}^r$, our unlearning algorithm is defined as $\bar{A}(\mathcal{D}, u, \bullet) = \text{med}(\mathcal{D} \circ u)$. Let the publish function $f_{\text{pub}} : \mathcal{O} \to \Phi$ be an identity function, i.e. $f_{\text{pub}}(\theta) = \theta$.

For any initial database $\mathcal{D}_0 \in \mathcal{X}^n$ and an adaptive sequence $(u_i)_{i \geq 1}$ generated by any $\infty$-adaptive 1-requester $\mathcal{Q}$, note that

$$f_{\text{pub}}(\bar{A}(\mathcal{D}_{i-1}, u_i, \bullet)) = f_{\text{pub}}(A(\mathcal{D}_i)), \quad \text{for all } i \geq 1 \text{ and any } \bullet \in \mathcal{O}. \tag{41}$$

Therefore, $\bar{A}$ is a $(0, 0)$-adaptive unlearning algorithm for $A$ under $f_{\text{pub}}$.

Now suppose that $n$ is odd and $\mathcal{D}_0$ consists of unique entries. W.L.O.G assume that the median record $\text{med}(\mathcal{D}_0)$ is at index $\text{ind}^m$ and its owner will be deleting it at step $i$ by sending a non-adaptive edit request $u_i = \{\langle \text{ind}^m, \mathbf{y} \rangle\}$ such that $\mathbf{y} \neq \text{med}(\mathcal{D}_0)$. We design the following 1-adaptive 1-requester $\mathcal{Q}$ that sends edit requests in the first $i-1$ steps to ensure with high probability that the published outcome at step $i$ remains the deleted record, i.e., $\text{med}(\mathcal{D}_i) = \text{med}(\mathcal{D}_0)$:

$$\mathcal{Q}(\phi_0, u_1, u_2, \cdots, u_{j-1}) = \{\langle \text{ind}_j, \phi_0 \rangle\} \quad \forall 1 \leq j < i, \tag{42}$$

where $\text{ind}_j$ is randomly sampled from $[n] \setminus \{\text{ind}_1, \cdots, \text{ind}_{j-1}\}$ without replacement. Note that by the end of interaction, $\mathcal{Q}$ replaces at-least $i-2$ unique records in $\mathcal{D}_0$ with $\phi_0 = \text{med}(\mathcal{D}_0)$. If one of

those original records was larger than $\mathrm{med}(\mathcal{D}_0)$ and another was smaller than $\mathrm{med}(\mathcal{D}_0)$, then it is guaranteed that $\mathrm{med}(\mathcal{D}_i) = \mathrm{med}(\mathcal{D}_0)$. Therefore, $\mathbb{P}\left[\mathrm{med}(\mathcal{D}_i) = \mathrm{med}(\mathcal{D}_0)\right]$ is at-least

$$
\mathbb{P}\left[\exists \mathrm{ind}^l, \mathrm{ind}^u \in \{\mathrm{ind}_1, \cdots, \mathrm{ind}_{i-1}\} \text{ s.t. } \mathcal{D}_0[\mathrm{ind}^l] < \mathcal{D}_0[\mathrm{ind}^m] < \mathcal{D}_0[\mathrm{ind}^u]\right]
$$

$$
\geq 1 - 2 \times \binom{\lfloor n \rfloor / 2}{i-2} \bigg/ \binom{n}{i-2} \geq 1 - \left(\frac{1}{2}\right)^{i-3}.
$$

$\square$

### E.1 Unsoundness and Incompleteness of Offline Unlearning Definitions

In this subsection, we show that our criticisms on both soundness and completeness of unlearning notions in Section 3 also apply to the one-stage unlearning definitions of Guo et al. (2019); Sekhari et al. (2021); Ginart et al. (2019).

**Definition E.1** (($\varepsilon, \delta$)-certified removal (Guo et al., 2019)). *A removal mechanism $\bar{A}$ performs ($\varepsilon, \delta$)-certified removal for learning algorithm $A$ if for all databases $\mathcal{D} \subset \mathcal{X}$ and deletion subset $S \subset \mathcal{D}$,*

$$
\bar{A}(\mathcal{D}, S, A(\mathcal{D})) \overset{\varepsilon, \delta}{\approx} A(\mathcal{D} \setminus S). \tag{43}
$$

**Definition E.2** (($\varepsilon, \delta$)-unlearning (Sekhari et al., 2021)). *For all $\mathcal{D} \subset \mathcal{X}$ of size $n$ and deletion subset $S \subset \mathcal{D}$ such that $|S| \leq m$, a learning algorithm $A$ and an unlearning algorithm $\bar{A}$ is ($\varepsilon, \delta$)-unlearning if*

$$
\bar{A}(T(\mathcal{D}), S, A(\mathcal{D})) \overset{\varepsilon, \delta}{\approx} \bar{A}(T(\mathcal{D} \setminus S), \varnothing, A(\mathcal{D} \setminus S)), \tag{44}
$$

*where $\varnothing$ denotes the empty set and $T(\mathcal{D})$ denotes the data statistics available to $\bar{A}$ regarding $\mathcal{D}$.*

**Definition E.3** (Data Deletion Operation Ginart et al. (2019)). *Fix any dataset $\mathcal{D} \subset \mathcal{X}$ and learning algorithm $A$. Operation $\bar{A}$ is a deletion operation for $A$ if $\bar{A}(\mathcal{D}, S, \bar{A}(\mathcal{D})) \overset{0,0}{\approx} A(\mathcal{D} \setminus S)$ for any set $S \subset \mathcal{D}$ selected independently of $A(\mathcal{D})$.*

**Unsoundness.** Definitions E.1 and E.2 make no assumptions about dependence between the deletion request $S$ and the learned model $A(\mathcal{D})$. So, request $S$ can depend on $A(\mathcal{D})$. This dependence is common in the real world; for example, a user deletes her information if she doesn't like what model $A(\mathcal{D})$ reveals about her. We present the following construction along the same lines as our proof in Theorem 2 to show that Definitions E.1 and E.2, are unsound.

For the universe of records $\mathcal{X} = \{-2, -1, 1, 2\}$, consider the following learning and unlearning algorithms:

$$
A(\mathcal{D}) = \sum_{\mathbf{x} \in \mathcal{D}} \mathbf{x}, \quad \text{and} \quad \bar{A}(\mathcal{D}, S, A(\mathcal{D})) = \sum_{\mathbf{x} \in \mathcal{D} \setminus S} \mathbf{x}. \tag{45}
$$

Note that for any $\mathcal{D} \subset \mathcal{X}$ and any $S \subset \mathcal{D}$, the above algorithm pair $(A, \bar{A})$ satisfies Definitions E.1, E.2 and E.3 for $\varepsilon = \delta = 0$ and $T(\mathcal{D}) = \mathcal{D}$. Suppose the adversary is aware that the following dependence holds between the learned model $A(\mathcal{D})$ and deletion request $S$:

$$
S = \begin{cases} \{\mathbf{x} < 0 : \forall \mathbf{x} \in \mathcal{X}\} & \text{if } A(\mathcal{D}) < 0, \\ \{\mathbf{x} > 0 : \forall \mathbf{x} \in \mathcal{X}\} & \text{otherwise.} \end{cases} \tag{46}
$$

Consider two neighbouring databases $\mathcal{D}_{-1} = \{-2, -1, 2\}$ and $\mathcal{D}_1 = \{-2, 1, 2\}$. Knowing the above dependence, an adversary can determine whether $\mathcal{D} = \mathcal{D}_{-1}$ or $\mathcal{D} = \mathcal{D}_1$ by looking only at $\bar{A}(\mathcal{D}, S, A(\mathcal{D}))$. This is because if $\mathcal{D} = \mathcal{D}_{-1}$, then the observation after unlearning is 2, and if $\mathcal{D} = \mathcal{D}_1$, the observation after unlearning is $-2$. So, even though $(A, \bar{A})$ satisfies the guarantees of Guo et al. (2019) and Sekhari et al. (2021), it blatantly reveals the identity ($-1$ or $1$) of a deleted record to an adversary observing only the post-deletion release.

Note that Definition E.3 assumes that the requests $S$ is selected independently of the learned model $A(\mathcal{D})$. So, our construction does not apply, keeping the possibility that their definition is sound. We remark, however, that algorithms satisfying their definitions cannot be trusted in settings where we expect some dependence between deletion requests and the learned models.

**Incompleteness.** Definitions E.1, E.2 and E.3 are also incomplete. Consider an unlearning algorithm $\bar{A}$ that outputs a fixed output $\mathbf{x}_1 \in \mathcal{X}$ if the deletion request $S = \varnothing$ and outputs another fixed output $\mathbf{x}_2 \in \mathcal{X}$ if the deletion request $S \neq \varnothing$. It is easy to see that $\bar{A}$ is a valid deletion algorithm as its output does not depend on the input database $\mathcal{D}$ or the learned model $A(\mathcal{D})$. However, note that $\bar{A}$ does not satisfy the unlearning Definition E.2, for any learning algorithm A. And, for a learning algorithm $A(\mathcal{D}) = \sum_{\mathbf{x} \in \mathcal{D}} \mathbf{x}$, one can also verify that the pair $(A, \bar{A})$ does not satisfy Definitions E.1 and E.3 either.

# F  PROOFS FOR SECTION 4

**Theorem 3** (Data-deletion Definition 4.1 is sound). *If the algorithm pair* $(A, \bar{A})$ *satisfies* $(q, \varepsilon)$-*data-deletion guarantee under all* $p$-*adaptive* $r$-*requesters, then even with the power of designing an* $p$-*adaptive* $r$-*requester* $\mathcal{Q}$ *that interacts with the curator before deletion of a target record at any step* $i \geq 1$, *any adversary observing only the post-deletion releases* $(\hat{\Theta}_i, \hat{\Theta}_{i+1}, \cdots)$ *has its membership inference advantage for inferring a deleted target bounded as*

$$Adv(MI) \leq \frac{q e^{\varepsilon(q-1)/q}}{q-1} [2(q-1)]^{1/q} - 1. \tag{47}$$

*Proof.* For an arbitrary step $i \geq 1$, suppose one of the replacement operations in the edit request $u_i \in \mathcal{U}^r$ replaces a record at index 'ind' from the database $\mathcal{D}_{i-1}$ with '$\mathbf{y}$'. In the worst case, this record $\mathcal{D}_{i-1}[\text{ind}]$ might have been there from the start, i.e. $\mathcal{D}_0[\text{ind}] = \mathcal{D}_0[\text{ind}]$, and influenced all the decisions of the adaptive requester $\mathcal{Q}$ in the edit steps $1, \cdots, i-1$. To prove soundness, we need to show that if $(A, \bar{A})$ satisfies $(q, \varepsilon)$-data-deletion, then even in this worst-case scenario, no adaptive adversary can design a membership inference test $MI(\hat{\Theta}_i, \hat{\Theta}_{i+1}, \cdots) \in \{0, 1\}$ that can distinguish with high probability the null hypothesis $H_0 = \{\mathcal{D}_0[\text{ind}] = \mathbf{x}\}$ from the alternate hypothesis $H_1 = \{\mathcal{D}_0[\text{ind}] = \mathbf{x}'\}$ for any $\mathbf{x}, \mathbf{x}' \in \mathcal{X}$. That is, the advantage of any test MI, defined as

$$Adv(MI) \stackrel{\text{def}}{=} \mathbb{P}\left[MI(\hat{\Theta}_i, \hat{\Theta}_{i+1}, \cdots) = 1 | H_0\right] - \mathbb{P}\left[MI(\hat{\Theta}_i, \hat{\Theta}_{i+1}, \cdots) = 1 | H_1\right], \tag{48}$$

must be small. Since after processing edit request $u_i$, the databases $\mathcal{D}_i, \mathcal{D}_{i+1}, \cdots$ no longer contain the deleted record $\mathcal{D}_{i-1}[\text{ind}]$, the data-processing inequality implies that future models $\hat{\Theta}_{i+1}, \hat{\Theta}_{i+2}, \cdots$ cannot have more information about $\mathcal{D}_{i-1}[\text{ind}]$ that what is present in $\hat{\Theta}_i$. Therefore, any test $MI(\hat{\Theta}_i, \hat{\Theta}_{i+1}, \cdots)$ has a smaller advantage than the optimal test $MI^*(\hat{\Theta}_i) \in \{0, 1\}$ that only uses $\hat{\Theta}_i$.

Also, since $(A, \bar{A})$ satisfy $(q, \varepsilon)$-data-deletion for any $p$-adaptive $r$-requester $\mathcal{Q}$, we know from Definition 4.1 that there exists a mapping $\pi_i^{\mathcal{Q}}$ such that for all $\mathcal{D}_0 \in \mathcal{X}^n$, the model $\hat{\Theta}_i$ generated by the interaction between $(A, \bar{A}, \mathcal{Q})$ on $\mathcal{D}_0$ after $i$th edit satisfies $R_q\left(\hat{\Theta}_i \middle\| \pi_i^{\mathcal{Q}}(\mathcal{D}_0 \circ \langle \text{ind}, \mathbf{y}\rangle)\right) \leq \varepsilon$. As the database $\mathcal{D}_0 \circ \langle \text{ind}, \mathbf{y}\rangle$ is identical under both hypothesis $H_0$ and $H_1$, we have $R_q\left(\hat{\Theta}_i | H_b \middle\| \bar{\Theta}\right) \leq \varepsilon$ for $b \in \{0, 1\}$, where $\bar{\Theta} \sim \pi_i^{\mathcal{Q}}(\mathcal{D}_0 \circ \langle \text{ind}, \mathbf{y}\rangle)$. From Rényi divergence to $(\varepsilon, \delta)$-indistinguishability conversion described in Remark 1, we get

$$\mathbb{P}\left[MI^*(\hat{\Theta}_i) = 1 | H_0\right] \leq e^{\varepsilon'(\delta)} \mathbb{P}\left[MI^*(\bar{\Theta}) = 1\right] + \delta, \text{ and} \tag{49}$$

$$\mathbb{P}\left[MI^*(\hat{\Theta}_i) = 0 | H_1\right] \leq e^{\varepsilon'(\delta)} \mathbb{P}\left[MI^*(\bar{\Theta}) = 0\right] + \delta, \tag{50}$$

where $\varepsilon'(\delta) = \varepsilon + \frac{\log 1/\delta}{q-1}$ for any $0 < \delta < 1$. On adding the two inequalities, we get:

$$Adv(MI) \leq Adv(MI^*) = \mathbb{P}\left[MI^*(\hat{\Theta}_i) = 1 | H_0\right] - \mathbb{P}\left[MI^*(\hat{\Theta}_i) = 1 | H_1\right]$$

$$\leq \min_{\delta} e^{\varepsilon'(\delta)} - 1 + 2\delta$$

$$= \frac{q e^{\varepsilon(q-1)/q}}{q-1} [2(q-1)]^{1/q} - 1$$

$$\square$$

**Theorem 4** (DP is necessary for data-deletion). *If learning algorithm* $A : \mathcal{X}^n \to \mathcal{O}$ *is not* $(0, \delta)$-*DP with respect to the replacement of a single record and deletion algorithm* $\bar{A} : \mathcal{X}^n \times \mathcal{U} \times \mathcal{O} \to \mathcal{O}$ *is not* $(0, \delta)$-*DP with respect to the replacement of a single record that is not being deleted, then the pair* $(A, \bar{A})$ *cannot satisfy* $(q, \delta^4/2)$-*data-deletion under* 1-*adaptive* 1-*requester for any* $q > 1$.

*Proof.* If $A$ is not $(0, \delta)$-DP with respect to replacement of a single record, then there exists a pair of neighbouring databases $\mathcal{D}, \mathcal{D}'$ such that

$$\mathbf{TV}\left(A(\mathcal{D}); A(\mathcal{D}')\right) = \sup_{O \in \mathcal{O}} |\mathbb{P}\left[A(\mathcal{D}) \in O\right] - \mathbb{P}\left[A(\mathcal{D}') \in O\right]| > \delta. \tag{51}$$

Similarly, if $\bar{A}$ is not $(0, \delta)$-DP with respect to replacement of a single record that is not being deleted, then there exists a pair of databases $\bar{\mathcal{D}}, \bar{\mathcal{D}}'$ and an edit request $\bar{u} \in \mathcal{U}^1$ such that $\bar{\mathcal{D}} \circ \bar{u}$ and $\bar{\mathcal{D}}' \circ \bar{u}$ are neighbouring and for all $\theta \in \mathcal{O}$,

$$\mathbf{TV}\left(\bar{A}(\bar{\mathcal{D}}, \bar{u}, \theta); \bar{A}(\bar{\mathcal{D}}', \bar{u}, \theta)\right) = \sup_{O \in \mathcal{O}} |\mathbb{P}\left[\bar{A}(\bar{\mathcal{D}}, \bar{u}, \theta) \in O\right] - \mathbb{P}\left[\bar{A}(\bar{\mathcal{D}}', \bar{u}, \theta) \in O\right]| > \delta. \tag{52}$$

Since $\mathbf{TV}$ distance is bounded from below in both cases, there exists tests $\text{Test}, \overline{\text{Test}} : \mathcal{O} \to \{0, 1\}$ such that

$$\text{Adv}(\text{Test}; \mathcal{D}, \mathcal{D}') \stackrel{\text{def}}{=} \mathbb{P}\left[\text{Test}(A(\mathcal{D})) = 1\right] - \mathbb{P}\left[\text{Test}(A(\mathcal{D}')) = 1\right] > \delta, \tag{53}$$

and for all $\theta \in \mathcal{O}$,

$$\text{Adv}(\overline{\text{Test}}; \bar{\mathcal{D}}, \bar{\mathcal{D}}', \bar{u}) \stackrel{\text{def}}{=} \mathbb{P}\left[\overline{\text{Test}}(\bar{A}(\bar{\mathcal{D}}, \bar{u}, \theta)) = 1\right] - \mathbb{P}\left[\overline{\text{Test}}(\bar{A}(\bar{\mathcal{D}}', \bar{u}, \theta)) = 1\right] > \delta. \tag{54}$$

Assume W.L.O.G. that $\bar{u}$ replaces at index $n$ and the edited databases $\bar{\mathcal{D}} \circ u, \bar{\mathcal{D}}' \circ u$ still differs at index 1. Also assume that $\mathcal{D}, \mathcal{D}'$ differs at index $n$.

Recall from Definition 4.1 that satisfying $(q, \frac{\delta^4}{2})$-data-deletion under 1-adaptive 1-requesters requires existence of a map $\pi_n^{\mathcal{Q}} : \mathcal{X}^n \to \mathcal{O}$ for each $\mathcal{Q}$ such that for all $\mathcal{D}_0 \in \mathcal{X}^n$,

$$R_q\left(\bar{A}(\mathcal{D}_{n-1}, u_n, \hat{\Theta}_{n-1}) \middle\| \pi_n^{\mathcal{Q}}(\mathcal{D}_0 \circ u_n)\right) \le \frac{\delta^4}{2}, \tag{55}$$

To prove the theorem statement, we show that for a starting database $\mathcal{D}_0 \in \{\mathcal{D}, \mathcal{D}'\}$ and an edit request $u_n = \bar{u}$ that deletes the differing record in choices of $\mathcal{D}_0$ at edit step $n$, there exists a 1-adaptive 1-requester $\mathcal{Q}$ that sends adaptive edit requests $u_1, \cdots, u_{n-1}$ in the first $n - 1$ steps such that no map $\pi_n^{\mathcal{Q}}$ exists that satisfies (55) for both choices of $\mathcal{D}_0$.

Consider the following construction of 1-adaptive 1-requester $\mathcal{Q}$ that only observes the first model $\hat{\Theta}_0 = A(\mathcal{D}_0)$ and generates the edit requests $(u_1, \cdots, u_{n-1})$ as follows:

$$\mathcal{Q}(\hat{\Theta}_0; u_1, u_2, \cdots, u_{i-1}) = \begin{cases} \langle i, \bar{\mathcal{D}}[i] \rangle & \text{if Test}(\hat{\Theta}_0) = 1, \\ \langle i, \bar{\mathcal{D}}'[i] \rangle & \text{otherwise}. \end{cases} \tag{56}$$

This requester $\mathcal{Q}$ transforms any initial database $\mathcal{D}_0$ to $\mathcal{D}_{n-1} = \bar{\mathcal{D}}$ if the outcome $\text{Test}(\hat{\Theta}_0) = 1$, otherwise to $\mathcal{D}_{n-1} = \bar{\mathcal{D}}'$. Consider an adversary that does not observe the interaction transcript $(\hat{\Theta}_{<n}; u_{<n})$, but is interested in identifying whether $\mathcal{D}_0$ was $\mathcal{D}$ or $\mathcal{D}'$. The adversary gets to observe only the output $\hat{\Theta}_n = \bar{A}(\mathcal{D}_{n-1}, u_n, \hat{\Theta}_{n-1})$ generated after processing the edit request $u_n = \bar{u}$. On this observation, the adversary runs the membership inference test $\text{MI}(\hat{\Theta}_n) = \overline{\text{Test}}(\hat{\Theta}_n)$. The membership inference advantage of MI is

$$\text{Adv}(\text{MI}; \mathcal{D}, \mathcal{D}') \stackrel{\text{def}}{=} \mathbb{P}\left[\text{MI}(\hat{\Theta}_n) = 1 | \mathcal{D}_0 = \mathcal{D}\right] - \mathbb{P}\left[\text{MI}(\hat{\Theta}_n) = 1 | \mathcal{D}_0 = \mathcal{D}'\right]$$

$$= \sum_{b \in \{0,1\}} \mathbb{P}\left[\overline{\text{Test}}(\hat{\Theta}_n) = 1 | \text{Test}(\hat{\Theta}_0) = b\right] \times \mathbb{P}\left[\text{Test}(\hat{\Theta}_0) = b | \mathcal{D}_0 = \mathcal{D}\right]$$

$$- \sum_{b \in \{0,1\}} \mathbb{P}\left[\overline{\text{Test}}(\hat{\Theta}_n) = 1 | \text{Test}(\hat{\Theta}_0) = b\right] \times \mathbb{P}\left[\text{Test}(\hat{\Theta}_0) = b | \mathcal{D}_0 = \mathcal{D}'\right]$$

$$= \left(\mathbb{P}\left[\overline{\text{Test}}(\hat{\Theta}_n) = 1 | \mathcal{D}_{n-1} = \bar{\mathcal{D}}\right] - \mathbb{P}\left[\overline{\text{Test}}(\hat{\Theta}_n) = 1 | \mathcal{D}_{n-1} = \bar{\mathcal{D}}'\right]\right) \text{Adv}(\text{Test}; \mathcal{D}, \mathcal{D}')$$

$$= \text{Adv}(\overline{\text{Test}}; \bar{\mathcal{D}}, \bar{\mathcal{D}}', \bar{u}) \times \text{Adv}(\text{Test}; \mathcal{D}, \mathcal{D}') > \delta^2.$$

So, it must be true that the total variation distance between $\hat{\Theta}_n$ given $\mathcal{D}_0 = \mathcal{D}$ and $\mathcal{D}_0 = \mathcal{D}'$ is lower bounded as

$$\mathbf{TV}\left(\hat{\Theta}_n|_{\mathcal{D}_0=\mathcal{D}}; \hat{\Theta}_n|_{\mathcal{D}_0=\mathcal{D}'}\right) > \delta^2. \tag{57}$$

From triangle inequality of total variation distance, Pinsker's inequality, and monotonicity of Rényi divergence w.r.t. order $q$, note that for a random variable $\bar{\Theta} \in \mathcal{O}$ with any arbitrary distribution,

$$\begin{aligned}
\delta^2 &< \mathbf{TV}\left(\hat{\Theta}_n|_{\mathcal{D}_0=\mathcal{D}}; \hat{\Theta}_n|_{\mathcal{D}_0=\mathcal{D}'}\right) \\
&\leq \mathbf{TV}\left(\hat{\Theta}_n|_{\mathcal{D}_0=\mathcal{D}}; \bar{\Theta}\right) + \mathbf{TV}\left(\hat{\Theta}_n|_{\mathcal{D}_0=\mathcal{D}'}; \bar{\Theta}\right) \\
&\leq \sqrt{\frac{1}{2}\mathrm{KL}\left(\hat{\Theta}_n|_{\mathcal{D}_0=\mathcal{D}'}\left\|\bar{\Theta}\right.\right)} + \sqrt{\frac{1}{2}\mathrm{KL}\left(\hat{\Theta}_n|_{\mathcal{D}_0=\mathcal{D}'}\left\|\bar{\Theta}\right.\right))} \\
&\leq \sqrt{2\max\left\{\mathrm{R}_q\left(\hat{\Theta}_n|_{\mathcal{D}_0=\mathcal{D}'}\left\|\bar{\Theta}\right.\right), \mathrm{R}_q\left(\hat{\Theta}_n|_{\mathcal{D}_0=\mathcal{D}'}\left\|\bar{\Theta}\right.\right)\right\}}.
\end{aligned}$$

This implies that for all random variables $\bar{\Theta}$, and all $q > 1$,

$$\max\left\{\mathrm{R}_q\left(\bar{\mathrm{A}}(\mathcal{D}_{i-1}, u_n, \hat{\Theta}_n)|_{\mathcal{D}_0=\mathcal{D}'}\left\|\bar{\Theta}\right.\right), \mathrm{R}_q\left(\bar{\mathrm{A}}(\mathcal{D}_{i-1}, u_n, \hat{\Theta}_n)|_{\mathcal{D}_0=\mathcal{D}'}\left\|\bar{\Theta}\right.\right)\right\} > \frac{\delta^4}{2}. \tag{58}$$

But, to satisfy $(q, \frac{\delta^4}{2})$-data-deletion under 1-adaptive 1-requesters, there must exist a mapping $\pi_n^{\mathcal{Q}}$ for which (55) must hold for both choices of $\mathcal{D}_0 \in \{\mathcal{D}, \mathcal{D}'\}$. Since the random variable $\pi_n^{\mathcal{Q}}(\mathcal{D}_0 \circ u_n)$ is identically distributed in our construction for both choices of $\mathcal{D}_0 \in \{\mathcal{D}, \mathcal{D}'\}$, from (58) we have that no such map $\pi_n^{\mathcal{Q}}$ can exist. $\qquad\square$

**Theorem 5** (Reduction from Adaptive to Non-adaptive Data Deletion). *If an algorithm pair $(\mathrm{A}, \bar{\mathrm{A}})$ satisfy $(q, \varepsilon_{\mathrm{dd}})$-data-deletion under all non-adaptive $r$-requesters and $(q, \varepsilon_{\mathrm{dp}})$-Rényi differential privacy, then it also satisfies $(q, \varepsilon_{\mathrm{dd}} + p\varepsilon_{\mathrm{dp}})$-data-deletion for all $p$-adaptive $r$-requesters.*

*Proof.* To prove this theorem, we need to show that for any $p$-adaptive $r$-requester $\mathcal{Q}$, there exists a construction for a map $\pi_i^{\mathcal{Q}} : \mathcal{X}^n \to \mathcal{O}$ such that for all $\mathcal{D}_0 \in \mathcal{X}^n$, the sequence of model $(\hat{\Theta}_i)_{i\geq 0}$ generated by the interaction between $(\mathcal{Q}, \mathrm{A}, \bar{\mathrm{A}})$ on $\mathcal{D}_0$ satisfies the following inequaltiy for all $i \geq 1$:

$$\mathrm{R}_q\left(\bar{\mathrm{A}}(\mathcal{D}_{i-1}, u_i, \hat{\Theta}_{i-1})\left\|\pi_i^{\mathcal{Q}}(\mathcal{D}_0 \circ \langle\mathrm{ind}, \mathbf{y}\rangle)\right.\right) \leq \varepsilon_{\mathrm{dd}} + p\varepsilon_{\mathrm{dp}}, \quad \text{for all } u_i \in \mathcal{U}^r \text{ and } \langle\mathrm{ind}, \mathbf{y}\rangle \in u_i. \tag{59}$$

Fix a database $\mathcal{D}_0 \in \mathcal{X}^n$ and an edit request $u_i \in \mathcal{U}^r$. Let $\mathcal{D}_0' \in \mathcal{X}^n$ be a neighbouring database defined to be $\mathcal{D}_0' = \mathcal{D}_0 \circ \langle\mathrm{ind}, \mathbf{y}\rangle$ for an arbitrary replacement operation $\langle\mathrm{ind}, \mathbf{y}\rangle \in u_i$. Given any $p$-adaptive $r$-requester $\mathcal{Q}$, let $(\hat{\Theta}_i)_{i\geq 0}$ and $(U_i)_{i\geq 1}$ be the sequence of released model and edit request random variables generated on $\mathcal{Q}$'s interaction with $(\mathrm{A}, \bar{\mathrm{A}})$ with inital database as $\mathcal{D}_0$. Similarly, let $(\hat{\Theta}_i')_{i\geq 0}$ and $(U_i')_{i\geq 1}$ be the corresponding sequences generated due to the interaction among $(\mathcal{Q}, \mathrm{A}, \bar{\mathrm{A}})$ on $\mathcal{D}_0'$.

Since $(\mathrm{A}, \bar{\mathrm{A}})$ is assumed to satisfy $(q, \varepsilon_{\mathrm{dd}})$-data-deletion guarantee under non-adpative $r$-requesters, recall from Remark 3 that there exists a mapping $\pi : \mathcal{X}^n \to \mathcal{O}$ such that for any fixed edit sequence $u_{\leq i} \overset{\text{def}}{=} (u_1, u_2, \cdots, u_i)$,

$$\mathrm{R}_q\left(\hat{\Theta}_i|_{U_{\leq i}=u_{\leq i}}\left\|\pi(\mathcal{D}_0 \circ u_{\leq i})\right.\right) \leq \varepsilon_{\mathrm{dd}} \tag{60}$$

$$\implies \mathrm{R}_q\left(\bar{\mathrm{A}}(\mathcal{D}_0 \circ U_{<i}, u_i, \hat{\Theta}_i)|_{U_{<i}=u_{<i}}\left\|\pi(\mathcal{D}_0 \circ U_{<i}' \circ u_i)|_{U_{<i}=u_{<i}}\right.\right) \leq \varepsilon_{\mathrm{dd}}. \tag{61}$$

Note that since the replacement operation $\langle\mathrm{ind}, \mathbf{y}\rangle$ is part of the edit request $u_i$, we have $\mathcal{D}_0 \circ U_{<i}' \circ u_i = \mathcal{D}_0' \circ U_{<i}' \circ u_i$. Moreover, since the sequence $U_{<i}'$ of edit requests is generated by the interaction of $(\mathcal{Q}, \mathrm{A}, \bar{\mathrm{A}})$ on $\mathcal{D}_0' = \mathcal{D}_0 \circ \langle\mathrm{ind}, u\rangle$ and the $i$th edit request $u_i$ is fixed beforehand, we can define a valid construction of a map $\pi_i^{\mathcal{Q}} : \mathcal{X}^n \to \mathcal{O}$ as per Definition 4.1 as follows:

$$\pi_i^{\mathcal{Q}}(\mathcal{D}_0 \circ \langle\mathrm{ind}, \mathbf{y}\rangle) = \pi(\mathcal{D}_0' \circ U_{<i}' \circ u_i). \tag{62}$$

For brevity, let $\hat{\Theta}_u = \bar{A}(\mathcal{D}_0 \circ U_{<i}, u_i, \hat{\Theta}_{i-1})$, and $\hat{\Theta}'_u = \pi_i^{\mathcal{Q}}(\mathcal{D}_0 \circ \langle \text{ind}, \mathbf{y} \rangle)$. For this construction, we prove the requisite bound in (59) as follows.

$$R_q\left(\hat{\Theta}_u \middle\| \hat{\Theta}'_u\right) \le R_q\left((\hat{\Theta}_u, U_{<i}) \middle\| (\hat{\Theta}'_u, U'_{<i})\right)$$

(Data processing inequality (Van Erven & Harremos, 2014, Theorem 1))

$$= \frac{1}{q-1} \log \int_\theta \sum_{u_{<i}} \frac{J(\theta, u_{<i})^q}{J'(\theta, u_{<i})^{q-1}} \mathrm{d}\theta$$

($J$ & $J'$ are joint PDFs of $(\hat{\Theta}_u, U_{<i})$ & $(\hat{\Theta}'_u, U'_{<i})$)

$$= \frac{1}{q-1} \log \sum_{u_{<i}} \frac{\mathbb{P}\left[U_{<i} = u_{<i}\right]^q}{\mathbb{P}\left[U'_{<i} = u_{<i}\right]^{q-1}} \left\{ \int_\theta \frac{p_{\hat{\Theta}_u | U_{<i} = u_{<i}}(\theta)^q}{p_{\hat{\Theta}'_u | U'_{<i} = u_{<i}}(\theta)^{q-1}} \mathrm{d}\theta \right\}$$

$$\le \frac{1}{q-1} \log \sum_{u_{<i}} \frac{\mathbb{P}\left[U_{<i} = u_{<i}\right]^q}{\mathbb{P}\left[U'_{<i} = u_{<i}\right]^{q-1}} \exp((q-1)\varepsilon_{\text{dd}}) \qquad \text{(From (61))}$$

$$= \varepsilon_{\text{dd}} + R_q\left(U_{<i} \| U'_{<i}\right)$$

$$\le \varepsilon_{\text{dd}} + R_q\left(\left(\hat{\Theta}_{s^1}, \cdots, \hat{\Theta}_{s^p}\right) \middle\| \left(\hat{\Theta}'_{s^1}, \cdots, \hat{\Theta}'_{s^p}\right)\right)$$

(If $\mathcal{Q}$ sees outputs at steps $s^1, \cdots, s^p$)

$$\le \varepsilon_{\text{dd}} + p\varepsilon_{\text{dp}}. \qquad \text{(Via Rényi composition)}$$

$\square$

## F.1 Contrasting Our Reduction Theorem 5 with Gupta et al. (2021)'s Results

The recent work of Gupta et al. (2021), also studies adaptive data deletion and proves a reduction from adaptive unlearning guarantee to non-adaptive unlearning guarantee in Definition 2.6 under differential privacy. We remark that the reduction Theorem 3.1 by Gupta et al. (2021) relies on DP with regards to a change in the description of learning/unlearning algorithm's coins and not with regards to the standard replacement of records. In contrast, our Theorem 5 presents a reduction from adaptive to non-adaptive data-deletion guarantee under DP with respect to the standard replacement of records. We emphasize that these two Theorems are fundamentally different.

The adaptive unlearning definition of Gupta et al. (2021) is designed to ensure with a high probability that no adaptive requester $\mathcal{Q}$ can force the output distribution of the unlearning algorithm $\bar{A}(\mathcal{D}_{i-1}, u_i, \hat{\Theta}_{i-1})$ to diverge substantially from that of retraining algorithm $A(\mathcal{D}_i)$. Such an attack is possible in stateful unlearning algorithms that rely on persistent structures that are only randomized once during initialization, for example, the initial partitioning of start database $\mathcal{D}_0$ in Bourtoule et al. (2021)'s SISA algorithm. Gupta et al. (2021) show in their Theorem 5.1 that an adaptive update requester $\mathcal{Q}$ can interactively send deletion requests $u_1, \cdots, u_i$ to SISA so that the partitioning of remaining records in $\mathcal{D}_i = \mathcal{D}_0 \circ u_1 \cdots u_i$ follows a pattern that is unlikely to occur on repartitioning of $\mathcal{D}_i$ when executing $A(\mathcal{D}_i)$. As proved in their reduction in Theorem 3.1, a straightforward way to prevent this is by ensuring that the uncertainty regarding the persistent structures remains private for long periods of time to an adversary observing the unlearned model. Hence the proof of their reduction from adaptive unlearning guarantee to non-adaptive unlearning guarantee relies on DP with regards to the coins of the unlearning algorithm.

Our work shows that satisfying adaptive unlearning definition of Gupta et al. (2021) still does not guarantee data deletion. In Theorem 2, we demonstrate that there exists an algorithm pair $(A, \bar{A})$ satisfying (a strictly stronger version) of adaptive unlearning (Gupta et al., 2021, Definition 2.3), but still causes blatant non-privacy of deleted records in post-deletion release. The vulnerability we identify occurs because an adaptive requester can learn the identity of any target record before it is deleted and re-encode it back in the curator's database by sending edit requests. Because of this, an adversary (who knows how the adaptive requester works but does not have access to the requester's interaction transcript) can extract the identity of the target record from the model released after processing the deletion request. In our work, we argue that a reliable (and necessary) way to

prevent this attack is to make sure that no adaptive requester ever learns the identity of a target record from the $p$ pre-deletion model releases it has access to. Consequently, our reduction in Theorem 5 from adaptive to non-adaptive requests relies on differential privacy with respect to the replacement of records instead.

# G  ADDITIONAL PRELIMINARIES AND PROOFS FOR SECTION 5

## G.1  LANGEVIN DIFFUSION AND MARKOV SEMIGROUPS

Langevin diffusion process on $\mathbb{R}^d$ with noise variance $\sigma^2$ under the influence of a potential $\mathcal{L} : \mathbb{R}^d \to \mathbb{R}$ is characterized by the Stochastic Differential Equation (SDE)

$$d\Theta_t = -\nabla\mathcal{L}(\Theta_t)dt + \sqrt{2\sigma^2}d\mathbf{Z}_t, \tag{63}$$

where $d\mathbf{Z}_t = \mathbf{Z}_{t+dt} - \mathbf{Z}_t \sim \sqrt{dt}\mathcal{N}(0, \mathbb{I}_d)$ is the $d$-dimensional Weiner process.

We present some preliminary knowledge on the diffusion theory used in our analysis. Let $p_t(\theta_0, \theta_t)$ denote the probability density function describing the distribution of $\Theta_t$, on starting from $\Theta_0 = \theta_0$ at time $t = 0$. For SDE (63), the associated Markov Semigroup $\mathbf{P}$, is defined as a family of operators $(P_t)_{t\geq 0}$, such that an operator $P_t$ sends any real-valued bounded measurable function $f : \mathbb{R}^d \to \mathbb{R}$ to

$$P_t f(\theta_0) = \mathbb{E}\left[f(\Theta_t)|\Theta_0 = \theta_0\right] = \int f(\theta_t)p_t(\theta_0, \theta_t)d\theta_t. \tag{64}$$

The infentisimal generator $\mathcal{G} \stackrel{\text{def}}{=} \lim_{s\to 0} \frac{1}{s}\left[P_{t+s} - P_s\right]$ for this diffusion Semigroup is

$$\mathcal{G}f = \sigma^2\Delta f - \langle\nabla\mathcal{L}, \nabla f\rangle. \tag{65}$$

This generator $\mathcal{G}$, when applied on a function $f(\theta_t)$, gives the infentisimal change in the value of a function $f$ when $\theta_t$ undergoes diffusion as per (63) for $dt$ time. That is,

$$\partial_t P_t f(\theta_0) = \int \partial_t p_t(\theta_0, \theta_t)f(\theta_t)d\theta_t = \int p_t(\theta_0, \theta_t)\mathcal{G}f(\theta_t)d\theta_t. \tag{66}$$

The dual operator of $\mathcal{G}$ is the Fokker-Planck operator $\mathcal{G}^*$, which is defined as the adjoint of generator $\mathcal{G}$, in the sense that

$$\int f\mathcal{G}^*g d\theta = \int g\mathcal{G}f d\theta, \tag{67}$$

for all real-valued bounded measurable functions $f, g : \mathbb{R}^d \to \mathbb{R}$. Note from (66) that, this operator provides an alternative way to represent the rate of change of function $f$ at time $t$:

$$\partial_t P_t f(\theta_0) = \int f(\theta_t)\mathcal{G}^*p_t(\theta_0, \theta_t)d\theta_t. \tag{68}$$

To put it simply, Fokker-Planck operator gives the infentesimal change in the distribution of $\Theta_t$ with respect to time. For the Langevin diffusion SDE (63), the Fokker-Planck operator is the following:

$$\partial_t p_t(\theta) = \mathcal{G}^*p_t(\theta) = \text{div}\left(p_t(\theta)\nabla\mathcal{L}(\theta)\right) + \sigma^2\Delta p_t(\theta). \tag{69}$$

From this Fokker-Placnk equation, one can verify that the stationary or invariant distribution $\pi$ of Langevin diffusion, which is the solution of $\partial_t p_t = 0$, follows the Gibbs distribution

$$\pi(\theta) \propto e^{-\mathcal{L}(\theta)/\sigma^2}. \tag{70}$$

Since $\pi$ is the stationary distribution, note that for any measurable bounded function $f : \mathbb{R}^d \to \mathbb{R}$,

$$\mathbb{E}_\pi[\mathcal{G}f] = \int f\mathcal{G}^*\pi d\theta = 0. \tag{71}$$

## G.2 ISOPERIMETRIC INEQUALITIES AND THEIR PROPERTIES

Convergence properties of various diffusion semigroups have been extensively analyzed in literature under certain isoperimetric assumptions on the stationary distribution $\pi$ (Bakry et al., 2014). One such property of interest is the *logarithmic Sobolev (*LS*) inequality* (Gross, 1975), which we define next.

The carré du champ operator $\Gamma$ of a diffusion semigroup with invariant measure $\mu$ is defined using its infintesimal generator $\mathcal{G}$ as

$$\Gamma(f, g) = \frac{1}{2} \left[ \mathcal{G}(fg) - f\mathcal{G}g - g\mathcal{G}f \right], \tag{72}$$

for every $f, g \in \mathbb{L}^2(\mu)$. Carre du champ operator represent fundamental properties of a Markov semigroup that affect its convergence behaviour. One can verify that Langevin diffusion semigroup's carre du champ operator (on differentiable $f, g$) is

$$\Gamma(f, g) = \sigma^2 \langle \nabla f, \nabla g \rangle. \tag{73}$$

We use shorthand notation $\Gamma(f) = \Gamma(f, f) = \sigma^2 \|\nabla f\|_2$.

**Definition G.1** (Logarithmic Sobolev Inequality (see Bakry et al. (2014, p. 24))). *A distribution with probability density $\pi$ is said to satisfy a* logarithmic Sobolev inequality *(*LS$(c)$*) (with respect to $\Gamma$ in (73)) if for all functions $f \in \mathbb{L}^2(\mu)$ with continuous derivatives $\nabla f$,*

$$\mathrm{Ent}_\pi(f^2) \le \frac{1}{2c} \int \frac{\Gamma(f^2)}{f^2} \pi \mathrm{d}\theta = \frac{2\sigma^2}{c} \int \|\nabla f\|_2^2 \pi \mathrm{d}\theta, \tag{74}$$

*where entropy $\mathrm{Ent}_\pi$ is defined as*

$$\mathrm{Ent}_\pi(f^2) = \mathbb{E}_\pi \left[ f^2 \log f^2 \right] - \mathbb{E}_\pi \left[ f^2 \right] \log \mathbb{E}_\pi \left[ f^2 \right]. \tag{75}$$

Logarithmic Sobolev inequality is a very non-restrictive assumption and is satisfied by a large class of distributions. The following well-known result show that Gaussians satisfy LS inequality.

**Lemma 18** (LS inequality of Gaussian distributions (see Bakry et al. (2014, p. 258))). *Let $\rho$ be a Gaussian distribution on $\mathbb{R}^d$ with covariance $\sigma^2/\lambda$ (i.e., the Gibbs distribution (70) with $\mathcal{L}(\cdot)$ being the L2 regularizer $\mathbf{r}(\theta) = \frac{\lambda}{2} \|\theta\|_2^2$). Then $\rho$ satisfies* LS$(\lambda)$ *tightly (with respect to $\Gamma$ in (73)), i.e.*

$$\mathrm{Ent}_\rho(f^2) = \frac{2\sigma^2}{\lambda} \int \|\nabla f\|_2^2 \rho \mathrm{d}\theta. \tag{76}$$

*Additionally, if $\mu$ is a distribution on $\mathbb{R}^d$ that satisfy* LS$(c)$*, then the convolution $\mu \circledast \rho$, defined as the distribution of $\Theta + \mathbf{Z}$ where $\Theta \sim \mu$ and $\mathbf{Z} \sim \pi$, satisfies* LS *inequality with constant $\left( \frac{1}{c} + \frac{1}{\lambda} \right)^{-1}$.*

Bobkov (2007) show that like Gaussians, all strongly log concave distributions (or more generally, log-concave distributions with finite second order moments) satisfy LS inequality (e.g. Gibbs distribution $\pi$ with any strongly convex $\mathcal{L}$). LS inequality is also satisfied under non-log-concavity too. For example, LS inequality is stable under Lipschitz maps, although such maps can destroy log-concavity.

**Lemma 19** (LS inequality under Lipschitz maps (see Ledoux (2001))). *If $\pi$ is a distribution on $\mathbb{R}^d$ that satisfies* LS$(c)$*, then for any $L$-lipschitz map $T : \mathbb{R}^d \to \mathbb{R}^d$, the pushforward distribution $T_{\#\pi}$, representing the distribution of $T(\Theta)$ when $\Theta \sim \pi$, satisfies* LS$(c/L^2)$.

LS inequality is also stable under bounded perturbations to the distribution, as shown in the following lemma by Holley & Stroock (1986).

**Lemma 20** (LS inequality under bounded perturbations (see Holley & Stroock (1986))). *If $\pi$ is the probability density of a distribution that satisfies* LS$(c)$*, then any proability distribution with density $\pi'$ such that $\frac{1}{\sqrt{B}} \le \frac{\pi(\theta)}{\pi'(\theta)} \le \sqrt{B}$ everywhere in $\mathbb{R}^d$ for some constant $B > 1$ satisfies* LS$(c/B)$.

Logarithmic Sobolev inequality is of interest to us due to its equivalence to the following inequalities on Kullback-Leibler and Rényi divergence.

**Lemma 21** (LS inequality in terms of KL divergence (Vempala & Wibisono, 2019)). *The distribution $\pi$ satisfies $\mathrm{LS}(c)$ inequality (with respect to $\Gamma$ in (73)) if and only if for all distributions $\mu$ on $\mathbb{R}^d$ such that $\frac{\mu}{\pi} \in \mathbb{L}^2(\pi)$ with continuous derivatives $\nabla \frac{\mu}{\pi}$,*

$$\mathrm{KL}\left(\mu \| \pi\right) \leq \frac{\sigma^2}{2c} \mathrm{I}\left(\mu \| \pi\right). \tag{77}$$

*Proof.* Set $f^2$ in (74) to $\frac{\mu}{\pi}$ to obtain (77). Alternatively, set $\mu = \frac{f^2 \pi}{\mathbb{E}_\pi[f^2]}$ in (77) to obtain (74). $\qquad\square$

**Lemma 22** (Wasserstein distance bound under LS inequality (Otto & Villani, 2000, Theorem 1)). *If distribution $\pi$ satisfies $\mathrm{LS}(c)$ inequality (with respect to $\Gamma$ in (73)) then for all distributions $\mu$ on $\mathbb{R}^d$,*

$$\mathrm{W}_2\left(\mu, \pi\right)^2 \leq \frac{2\sigma^2}{c} \mathrm{KL}\left(\mu \| \pi\right). \tag{78}$$

**Lemma 23** (LS inequality in terms of Rényi Divergence (Vempala & Wibisono, 2019)). *The distribution $\pi$ satisfies $\mathrm{LS}(c)$ inequality (with respect to $\Gamma$ in (73)) if and only if for all distributions $\mu$ on $\mathbb{R}^d$ such that $\frac{\mu}{\pi} \in \mathbb{L}^2(\pi)$ with continuous derivatives $\nabla \frac{\mu}{\pi}$, and any $q > 1$,*

$$\mathrm{R}_q\left(\mu \| \pi\right) + q(q-1)\partial_q \mathrm{R}_q\left(\mu \| \pi\right) \leq \frac{q^2 \sigma^2}{2c} \frac{\mathrm{I}_q\left(\mu \| \pi\right)}{\mathrm{E}_q\left(\mu \| \pi\right)}. \tag{79}$$

*Proof.* For brevity, let the functions $R(q) = \mathrm{R}_q\left(\mu \| \pi\right)$, $E(q) = \mathrm{E}_q\left(\mu \| \pi\right)$, and $I(q) = \mathrm{I}_q\left(\mu \| \pi\right)$. Let function $f^2(\theta) = \left(\frac{\mu(\theta)}{\pi(\theta)}\right)^q$. Then,

$$\mathbb{E}_\pi\left[f^2\right] = \mathbb{E}_\pi\left[\left(\frac{\mu}{\pi}\right)^q\right] = E(q), \qquad \text{(From (34))}$$

and,

$$\mathbb{E}_\pi\left[f^2 \log f^2\right] = \mathbb{E}_\pi\left[\left(\frac{\mu}{\pi}\right)^q \log \left(\frac{\mu}{\pi}\right)^q\right] = q\partial_q \mathbb{E}_\pi\left[\int_q \left(\frac{\mu}{\pi}\right)^q \log\left(\frac{\mu}{\pi}\right) \mathrm{d}q\right] = q\partial_q \mathbb{E}_\pi\left[\left(\frac{\mu}{\pi}\right)^q\right] = q\partial_q E(q).$$
$$\text{(From Lebniz rule and (34))}$$

Moreover,

$$\mathbb{E}_\pi\left[\|\nabla f\|_2^2\right] = \mathbb{E}_\pi\left[\left\|\nabla \left(\frac{\mu}{\pi}\right)^{\frac{q}{2}}\right\|_2^2\right] = \frac{q^2}{4} I(q) \qquad \text{(From (40))}$$

On substituting (74) with the above equalities, we get:

$$\mathrm{Ent}_\pi(f^2) \leq \frac{2\sigma^2}{c} \mathbb{E}_\pi\left[\|\nabla f\|_2^2\right]$$

$$\iff q\partial_q E(q) - E(q)\log E(q) \leq \frac{q^2\sigma^2}{2c} I(q)$$

$$\iff q\partial_q \log E(q) - \log E(q) \leq \frac{q^2\sigma^2}{2c} \frac{I(q)}{E(q)}$$

$$\iff q\partial_q \left((q-1)R(q)\right) - (q-1)R(q) \leq \frac{q^2\sigma^2}{2c} \frac{I(q)}{E(q)} \qquad \text{(From (34))}$$

$$\iff R(q) + q(q-1)\partial_q R(q) \leq \frac{q^2\sigma^2}{2c} \frac{I(q)}{E(q)}$$

$\qquad\square$

## G.3 (RÉNYI) DIFFERENTIAL PRIVACY GUARANTEES ON NOISY-GD

In this section, we recap the differential privacy bounds in literature for Noisy-GD Algorithm 1.

**Theorem 24** (Rényi DP guarantee for Noisy-GD Algorithm 1). *If $\ell(\theta; \mathbf{x})$ is L-lipschitz, then Noisy-GD satisfies $(q, \varepsilon)$-RDP with $\varepsilon = \frac{qL^2}{\sigma^2 n^2} \cdot \eta K$.*

*Proof.* The $L_2$ sensitivity of gradient $\nabla \mathcal{L}_{\mathcal{D}}(\theta) \overset{\text{def}}{=} \frac{1}{n} \sum_{\mathbf{x} \in \mathcal{D}} \nabla \ell(\theta; \mathbf{x}) + \nabla \mathbf{r}(\theta)$ computed in step 2 of Algorithm 1 for neighboring databases in $\mathcal{X}^n$ that differ in a single record is $\frac{2L}{n}$ since $\ell(\theta; \mathbf{x})$ is $L$-Lipschitz.

Conditioned on observing the intermediate model $\Theta_{\eta k} = \theta_k$ at step $k$, the next model $\Theta_{\eta(k+1)}$ after the noisy gradient update is a Gaussian mechanism with noise variance $2\sigma^2/\eta$. So, for neighboring databases $\mathcal{D}, \mathcal{D}' \in \mathcal{X}^n$, we have from the Rényi DP bound of Gaussian mechanisms proposed by Mironov (2017, Proposition 7) that

$$\mathrm{R}_q \left( \Theta_{\eta(k+1)} \mid_{\Theta_{\eta k} = \theta_k} \middle\| \Theta'_{\eta(k+1)} \mid_{\Theta'_{\eta k} = \theta_k} \right) \leq \frac{\eta q L^2}{n^2 \sigma^2}, \tag{80}$$

where $(\Theta_{\eta k})_{0 \leq k \leq K}$ and $(\Theta'_{\eta k})_{0 \leq k \leq K}$ are intermediate parameters in Algorithm 1 when run on databases $\mathcal{D}$ and $\mathcal{D}'$ respectively. Finally, from Rényi composition Mironov (2017, Proposition 1), we have

$$\begin{aligned}
\mathrm{R}_q \left( \Theta_{\eta K} \middle\| \Theta'_{\eta K} \right) &\leq \mathrm{R}_q \left( (\Theta_0, \Theta_\eta, \cdots, \Theta_{\eta K}) \middle\| (\Theta'_0, \Theta'_\eta, \cdots, \Theta'_{\eta K}) \right) \\
&\leq \sum_{k=0}^{K-1} \mathrm{R}_q \left( \Theta_{\eta(k+1)} \mid_{\Theta_{\eta k} = \theta_k} \middle\| \Theta'_{\eta(k+1)} \mid_{\Theta'_{\eta k} = \theta_k} \right) \\
&\leq \frac{q L^2}{n^2 \sigma^2} \cdot \eta K.
\end{aligned}$$

$\square$

**Remark 5.** *Different papers discussing Noisy-GD variants adopt different notational conventions for the total noise added to the gradients. The noise variance in our Algorithm 1 is $2\eta\sigma^2$; but is $\frac{\eta^2 \sigma^2 L^2}{n^2}$ in the full-batch setting of DP-SGD by Abadi et al. (2016). To translate the bound in Theorem 24, one can simply rescale $\sigma$ across different conventions to have the same noise variance, i.e., $2\eta\sigma^2 = \frac{\eta^2 \hat{\sigma}^2 L^2}{n^2}$.*

Our Theorem 24 is somewhat identical to Abadi et al. (2016)'s $(\varepsilon, \delta)$-DP bound. To verify this, note from Rényi divergece to $(\varepsilon, \delta)$-indistinguishability conversion discussed in Remark 1 that $(1 + \frac{2}{\varepsilon} \log \frac{1}{\delta}, \frac{\varepsilon}{2})$-Rényi DP implies $(\varepsilon, \delta)$-DP. So, setting the bound in Theorem 24 to be smaller than $\frac{\varepsilon}{2}$ and substituting $q = 1 + \frac{2}{\varepsilon} \log \frac{1}{\delta}$, we get

$$\left( \frac{\varepsilon + 2 \log \frac{1}{\delta}}{\varepsilon} \right) \frac{L^2}{n^2 \sigma^2} \cdot \eta K \leq \frac{\varepsilon}{2} \iff \frac{\sqrt{K(\varepsilon + 2 \log \frac{1}{\delta})}}{\varepsilon} \leq \hat{\sigma}.$$

For $\varepsilon \leq 2 \log \frac{1}{\delta}$, we get the same noise bound as in Abadi et al. (2016, Theorem 1) for their (full-batch) DP-SGD algorithm.

Next, we recap the tighter Rényi DP guarantee of Chourasia et al. (2021) under stronger assumptions on the loss function.

**Theorem 25** (Rényi DP guarantee for Noisy-GD Algorithm 1 (Chourasia et al., 2021)). *If $\ell(\theta; \mathbf{x})$ is convex, $L$-Lipschitz, and $\beta$-smooth and $\mathbf{r}(\theta)$ is the L2 regularizer with constant $\lambda$, then Noisy-GD with learning rate $\eta < \frac{1}{\beta + \lambda}$ satisfies $(q, \varepsilon)$-RDP with $\varepsilon = \frac{4q L^2}{\lambda \sigma^2 n^2} \left( 1 - e^{-\lambda \eta K/2} \right)$.*

### G.4 PROOFS FOR SUBSECTION 5.1

In this appendix, we provide a proof of our Theorem 6 which applies to convex losses $\ell(\theta; \mathbf{x})$ under L2 regularizer $\mathbf{r}(\theta)$. Let $\mathcal{D}_0 \in \mathcal{X}^n$ be any arbitrary database, and $\mathcal{Q}$ be any non-adaptive $r$-requester.

Our first goal in this section is to prove $(q, \varepsilon_{\text{dd}})$-data-deletion guarantees on our proposed algorithm pair $(\mathrm{A}_{\text{Noisy-GD}}, \bar{\mathrm{A}}_{\text{Noisy-GD}})$ (in Definition 5.1) under $\mathcal{Q}$. That is, if $(\hat{\Theta}_i)_{i \geq 0}$ is the sequence of models produced by the interaction between $(\mathrm{A}_{\text{Noisy-GD}}, \bar{\mathrm{A}}_{\text{Noisy-GD}}, \mathcal{Q})$ on $\mathcal{D}_0$, we need to show that their exists a mapping $\pi_i^{\mathcal{Q}}$ such that for all $i \geq 1$ and any $u_i \in \mathcal{U}^r$,

$$\mathrm{R}_q \left( \bar{\mathrm{A}}(\mathcal{D}_{i-1}, u_i, \hat{\Theta}_{i-1}) \middle\| \pi_i^{\mathcal{Q}}(\mathcal{D}_0 \circ \langle \mathrm{ind}, \mathbf{y} \rangle) \right) \leq \varepsilon_{\text{dd}} \quad \text{for all } \langle \mathrm{ind}, \mathbf{y} \rangle \in u_i. \tag{81}$$

For an arbitrary replacement operation $\langle \text{ind}, \mathbf{y} \rangle$ in $u_i$, we define a map $\pi_i^{\mathcal{Q}}(\mathcal{D}_0 \circ \langle \text{ind}, \mathbf{y} \rangle) = \hat{\Theta}_i'$, where the model sequence $(\hat{\Theta}_i')_{i \geq 0}$ is produced by the interaction of between $(\text{A}_{\text{Noisy-GD}}, \bar{\text{A}}_{\text{Noisy-GD}}, \mathcal{Q})$ on $\mathcal{D}_0 \circ \langle \text{ind}, \mathbf{y} \rangle$. Since non-adaptive requester $\mathcal{Q}$ is equivalent to fixing the edit sequence $(u_i)_{i \geq 1}$ a-priori, note that showing the data-deletion guarantee reduces to proving the following DP-like bound

$$\text{R}_q \left( \bar{\text{A}}(\mathcal{D}_{i-1}, u_i, \hat{\Theta}_{i-1}) \middle\| \bar{\text{A}}(\mathcal{D}_{i-1}', u_i, \hat{\Theta}_{i-1}') \right) \leq \varepsilon_{\text{dd}}, \tag{82}$$

for for all $u_{\leq i}$ and for all neighbouring databases $\mathcal{D}_0, \mathcal{D}_0'$ s.t. $\mathcal{D}_0' = \mathcal{D}_0 \circ \langle \text{ind}, \mathbf{y} \rangle$ with $\langle \text{ind}, \mathbf{y} \rangle \in u_i$.

Note from our Definition 5.1 that the sequence of models $(\hat{\Theta}_0, \cdots, \hat{\Theta}_i)$ can be seen as being generated from a continuous run of Noisy-GD, where:

1. for iterations $0 \leq k < K_{\text{A}}$, the loss function is $\mathcal{L}_{\mathcal{D}_0}$,

2. for the iterations $K_{\text{A}} + (j-1)K_{\bar{\text{A}}} \leq k < K_{\text{A}} + jK_{\bar{\text{A}}}$ on any $1 \leq j \leq i-1$, the loss function is $\mathcal{L}_{\mathcal{D}_j}$, and

3. for the iterations $K_{\text{A}} + (i-1)K_{\bar{\text{A}}} \leq k < K_{\text{A}} + iK_{\bar{\text{A}}}$, the loss function is $\mathcal{L}_{\mathcal{D}_{i-1} \circ u_i}$.

Let $(\Theta_{\eta k})_{0 \leq k \leq K_{\text{A}} + iK_{\bar{\text{A}}}}$ be the sequence representing the intermediate parameters of this extended Noisy-GD run. Similarly, let $(\Theta_{\eta k}')_{k \geq 0}$ be the parameter sequence corresponding to the extended run on the neighbouring database $\mathcal{D}_0'$. Since $\langle \text{ind}, \mathbf{y} \rangle \in u_i$, note from the construction that $\mathcal{D}_{i-1}' \circ u_i = \mathcal{D}_{i-1} \circ u_i$, meaning that the loss functions while processing request $u_i$ is identical for the two processes, i.e. $\mathcal{L}_{\mathcal{D}_{i-1} \circ u_i} = \mathcal{L}_{\mathcal{D}_{i-1}' \circ u_i}$. For brevity, we refer to the database seen in iteration $k$ of the two respective extended runs as $\mathcal{D}(k)$ and $\mathcal{D}'(k)$ respectively. In short, these two discrete processes induced by Noisy-GD follow the following update rule for any $0 \leq k < K_{\text{A}} + iK_{\bar{\text{A}}}$:

$$\begin{cases} \Theta_{\eta(k+1)} = \Theta_{\eta k} - \eta \nabla \mathcal{L}_{\mathcal{D}(k)}(\Theta_{\eta k}) + \sqrt{2\eta \sigma^2} \mathbf{Z}_k \\ \Theta_{\eta(k+1)}' = \Theta_{\eta k}' - \eta \nabla \mathcal{L}_{\mathcal{D}'(k)}(\Theta_{\eta k}') + \sqrt{2\eta \sigma^2} \mathbf{Z}_k', \end{cases} \quad \text{where } \mathbf{Z}_k, \mathbf{Z}_k' \sim \mathcal{N}(0, \mathbb{I}_d), \tag{83}$$

and $\Theta_0$ and $\Theta_0'$ are sampled from same the weight initialization distribution $\rho$. To prove the bound in (82), we follow the approach proposed in Chourasia et al. (2021) of interpolating the two discrete stochastic process of Noisy-GD with two piecewise-continuous tracing diffusions $\Theta_t$ and $\Theta_t'$ in the duration $\eta k < t \leq \eta(k+1)$, defined as follows.

$$\begin{cases} \Theta_t = T_k(\Theta_{\eta k}) - \frac{(t-\eta k)}{2} \nabla \left( \mathcal{L}_{\mathcal{D}(k)}(\Theta_{\eta k}) - \mathcal{L}_{\mathcal{D}'(k)}(\Theta_{\eta k}) \right) + \sqrt{2\sigma^2}(\mathbf{Z}_t - \mathbf{Z}_{\eta k}), \\ \Theta_t' = T_k(\Theta_{\eta k}') + \frac{(t-\eta k)}{2} \nabla \left( \mathcal{L}_{\mathcal{D}(k)}(\Theta_{\eta k}') - \mathcal{L}_{\mathcal{D}'(k)}(\Theta_{\eta k}') \right) + \sqrt{2\sigma^2}(\mathbf{Z}_t' - \mathbf{Z}_{\eta k}'), \end{cases} \tag{84}$$

where $\mathbf{Z}_t, \mathbf{Z}_t'$ are two independent Weiner processes, and $T_k$ is a map on $\mathbb{R}^d$ defined as

$$T_k = \mathbb{I}_d - \frac{\eta}{2} \nabla \left( \mathcal{L}_{\mathcal{D}(k)} + \mathcal{L}_{\mathcal{D}'(k)} \right). \tag{85}$$

Note that equation (84) is identical to (83) when $t = \eta(k+1)$, and can be expressed by the following stochastic differential equations (SDEs):

$$\begin{cases} \text{d}\Theta_t = -\mathbf{g}_k(\Theta_{\eta k})\text{d}t + \sqrt{2\sigma^2}\text{d}\mathbf{Z}_t \\ \text{d}\Theta_t' = +\mathbf{g}_k(\Theta_{\eta k}')\text{d}t + \sqrt{2\sigma^2}\text{d}\mathbf{Z}_t', \end{cases} \quad \text{where } \mathbf{g}_k(\Theta) = \frac{1}{2n} \nabla \left[ \boldsymbol{\ell}(\Theta; \mathcal{D}(k)[\text{ind}]) - \boldsymbol{\ell}(\Theta; \mathcal{D}'(k)[\text{ind}]) \right],$$

$$\tag{86}$$

and initial condition $\lim_{t \to \eta k^+} \Theta_t = T_k(\Theta_{\eta k})$, $\lim_{t \to \eta k^+} \Theta_t' = T_k(\Theta_{\eta k}')$. These two SDEs can be equivalently described by the following pair of Fokker-Planck equations.

**Lemma 26** (Fokker-Planck equation for SDE (86)). *Fokker-Planck equation for SDE in* (86) *at time* $\eta k < t \leq \eta(k+1)$, *is*

$$\begin{cases} \partial_t \mu_t(\theta) &= \text{div} \left( \mu_t(\theta) \mathbb{E} \left[ \mathbf{g}_k(\Theta_{\eta k}) | \Theta_t = \theta \right] \right) + \sigma^2 \Delta \mu_t(\theta), \\ \partial_t \mu_t'(\theta) &= \text{div} \left( \mu_t'(\theta) \mathbb{E} \left[ -\mathbf{g}_k(\Theta_{\eta k}') \middle| \Theta_t' = \theta \right] \right) + \sigma^2 \Delta \mu_t'(\theta), \end{cases} \tag{87}$$

*where* $\mu_t$ *and* $\mu_t'$ *are the densities of* $\Theta_t$ *and* $\Theta_t'$ *respectively.*

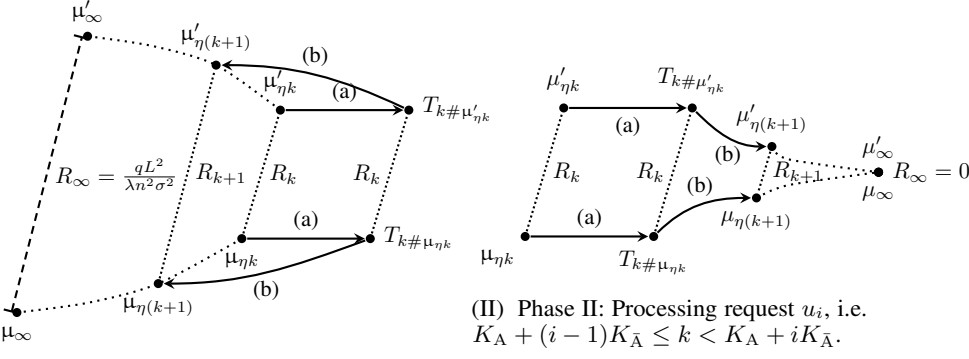

(I) Phase I: Processing requests $u_{<i}$, i.e. $0 \leq k < K_{\mathrm{A}} + (i-1)K_{\bar{\mathrm{A}}}$.

(II) Phase II: Processing request $u_i$, i.e. $K_{\mathrm{A}} + (i-1)K_{\bar{\mathrm{A}}} \leq k < K_{\mathrm{A}} + iK_{\bar{\mathrm{A}}}$.

Figure 1: Diagram illustrating the technical overview of Theorem 31. Here $\mu_{\eta k}$ and $\mu_{\eta k'}$ represent the $k$th iteration parameter distribution of $\Theta_{\eta k}$ and $\Theta'_{\eta k}$ respectively. We interpolate the two discrete processes in two steps: (a) an identical transformation $T_k$ (as defined in (85), and (b) a diffusion process. If divergence before descent step is $R_k = \mathrm{R}_q\left(\mu_{\eta k}\middle\|\mu'_{\eta k}\right)$, the stochastic mapping $T_k$ in (a) doesn't increase the divergence, while the diffusion (b) either increases it upto an asymptotic constant in phase I or decreases it exponentially to 0 in phase II.

*Proof.* Conditioned on observing parameter $\Theta_{\eta k} = \theta_{\eta k}$, the process $(\Theta_t)_{\eta k < t \leq \eta(k+1)}$ is a Langevin diffusion along a constant Vector field (i.e. on conditioning, we get a Langevin SDE (63) with $\nabla\mathcal{L}(\theta) = \mathbf{g}_k(\theta_{\eta k})$ for all $\theta \in \mathbb{R}^d$). Therefore as per (69), the conditional probability density $\mu_{t|\eta k}(\cdot|\theta_{\eta k})$ of $\Theta_t$ given $\Theta_{\eta k}$ follows the following Fokker-Planck equation:

$$\partial_t \mu_{t|\eta k}(\cdot|\theta_{\eta k}) = \mathrm{div}\left(\mu_{t|\eta k}(\cdot|\theta_{\eta k})\mathbf{g}_k(\theta_{\eta k})\right) + \sigma^2 \Delta \mu_{t|\eta k}(\cdot|\theta_{\eta k}) \tag{88}$$

Taking expectation over $\mu_{\eta k}$ which is the distribution of $\Theta_{\eta k}$,

$$\partial_t \mu_t(\cdot) = \int \mu_{\eta k}(\theta_{\eta k})\left\{\mathrm{div}\left(\mu_{t|\eta k}(\cdot|\theta_{\eta k})\mathbf{g}_k(\theta_{\eta k})\right) + \sigma^2 \Delta \mu_{t|\eta k}(\cdot|\theta_{\eta k})\right\} \mathrm{d}\theta_{\eta k}$$

$$= \mathrm{div}\left(\int \mathbf{g}_k(\theta_{\eta k})\mu_{t,\eta k}(\cdot,\theta_{\eta k})\mathrm{d}\theta_{\eta k}\right) + \sigma^2 \Delta \mu_t(\cdot)$$

$$= \mathrm{div}\left(\mu_t(\cdot)\left\{\int \mathbf{g}_k(\theta_{\eta k})\mu_{\eta k|t}(\theta_{\eta k}|\cdot)\mathrm{d}\theta_{\eta k}\right\}\right) + \sigma^2 \Delta \mu_t(\cdot)$$

$$= \mathrm{div}\left(\mu_t(\cdot)\mathbb{E}\left[\mathbf{g}_k(\Theta_{\eta k})|\Theta_t = \cdot\right]\right) + \sigma^2 \Delta \mu_t(\cdot).$$

where $\mu_{\eta k,|t}$ is the conditional density of $\Theta_{\eta k}$ given $\Theta_t$. Proof for second fokker-planck equation is similar. $\qquad\square$

We provide an overview of how we bound equation (82) in Figure 1. Basically, our analysis has two phases; in phase (I) we provide a bound on $\mathrm{R}_q\left(\hat{\Theta}_{i-1}\middle\|\hat{\Theta}'_{i-1}\right)$ that holds for any choice of number of iterations $K_{\mathrm{A}}$ and $K_{\bar{\mathrm{A}}}$, and in phase (II) we prove an exponential contraction in the divergence $\mathrm{R}_q\left(\bar{\mathrm{A}}(\mathcal{D}_{i-1}, u_i, \hat{\Theta}_{i-1})\middle\|\bar{\mathrm{A}}(\mathcal{D}'_{i-1}, u_i, \hat{\Theta}'_{i-1})\right)$ with number of iterations $K_{\bar{\mathrm{A}}}$.

We first introduce a few lemmas that will be used in both phases. The first set of following lemmas show that the transformation $\Theta_{\eta k}, \Theta'_{\eta k} \to T_k(\Theta_{\eta k}), T_k(\Theta_{\eta k})$ preserves the Rényi divergence. To prove this property, we show that $T_k$ is a differentiable bijective map in Lemma 28 and apply the following Lemma from Vempala & Wibisono (2019).

**Lemma 27** (Vempala & Wibisono (2019, Lemma 15)). *If $T : \mathbb{R}^d \to \mathbb{R}^d$ is a differentiable bijective map, then for any random variables $\Theta, \Theta' \in \mathbb{R}^d$, and for all $q > 0$,*

$$\mathrm{R}_q\left(T(\Theta)\|T(\Theta')\right) = \mathrm{R}_q\left(\Theta\|\Theta\right). \tag{89}$$

**Lemma 28.** *If $\ell(\theta; \mathbf{x})$ is a twice continuously differentiable, convex, and $\beta$-smooth loss function and regularizer is $\mathbf{r}(\theta) = \frac{\lambda}{2} \|\theta\|_2^2$, then the map $T_k$ defined in (85) is:*

1. *a differentiable bijection for any $\eta < \frac{1}{\lambda+\beta}$, and*

2. *$(1 - \eta\lambda)$-Lipschitz for any $\eta \leq \frac{2}{2\lambda+\beta}$.*

*Proof.* **Differentiable bijection.** To see that $T_k$ is injective, assume $T_k(\theta) = T_k(\theta')$ for some $\theta, \theta' \in \mathbb{R}^d$. Then, by $(\beta + \lambda)$-smoothness of $\mathcal{L} \stackrel{\text{def}}{=} (\mathcal{L}_{\mathcal{D}(k)} + \mathcal{L}_{\mathcal{D}'(k)})/2$,

$$
\begin{aligned}
\|\theta - \theta'\|_2 &= \|T_k(\theta) + \eta\nabla\mathcal{L}(\theta) - T_k(\theta') - \eta\nabla\mathcal{L}(\theta')\|_2 \\
&= \eta \|\nabla\mathcal{L}(\theta) - \nabla\mathcal{L}(\theta')\|_2 \\
&\leq \eta(\lambda + \beta) \|\theta - \theta'\|_2 .
\end{aligned}
$$

Since $\eta < 1/(\lambda + \beta)$, we must have $\|\theta - \theta'\|_2 = 0$. For showing $T_k$ is surjective, consider the proximal mapping

$$
\text{prox}_{\mathcal{L}}(\theta) = \arg\min_{\theta' \in \mathbb{R}^d} \frac{\|\theta' - \theta\|_2^2}{2} - \eta\mathcal{L}(\theta'). \tag{90}
$$

Note that $\text{prox}_{\mathcal{L}}(\cdot)$ is strongly convex for $\eta < \frac{1}{\lambda+\beta}$. Therefore, from KKT conditions, we have $\theta = \text{prox}_{\mathcal{L}}(\theta) - \eta\nabla\mathcal{L}(\text{prox}_{\mathcal{L}}(\theta)) = T_k(\text{prox}_{\mathcal{L}}(\theta))$. Differentiability of $T_k$ follows from the twice continuously differentiable assumption on $\ell(\theta; \mathbf{x})$.

**Lipschitzness.** Let $\mathcal{L} \stackrel{\text{def}}{=} (\mathcal{L}_{\mathcal{D}(k)} + \mathcal{L}_{\mathcal{D}'(k)})/2$. For any $\theta, \theta' \in \mathbb{R}^d$,

$$
\begin{aligned}
\|T_k(\theta) - T_k(\theta')\|_2^2 &= \|\theta - \eta\nabla\mathcal{L}(\theta) - \theta' + \eta\nabla\mathcal{L}(\theta')\|_2^2 \\
&= \|\theta - \theta'\|_2^2 + \eta^2 \|\nabla\mathcal{L}(\theta) - \nabla\mathcal{L}(\theta')\|_2^2 - 2\eta \langle\theta - \theta', \nabla\mathcal{L}(\theta) - \nabla\mathcal{L}(\theta')\rangle .
\end{aligned}
$$

We define a function $g(\theta) = \mathcal{L}(\theta) - \frac{\lambda}{2} \|\theta\|_2^2$, which is convex and $\beta$-smooth. By co-coercivity property of convex and $\beta$-smooth functions, we have

$$
\langle\theta - \theta', \nabla g(\theta) - \nabla g(\theta')\rangle \geq \frac{1}{\beta} \|\nabla g(\theta) - \nabla g(\theta')\|_2^2
$$

$$
\implies \langle\theta - \theta', \nabla\mathcal{L}(\theta) - \nabla\mathcal{L}(\theta')\rangle - \lambda \|\theta - \theta'\|_2^2 \geq \frac{1}{\beta}\bigg( \|\nabla\mathcal{L}(\theta) - \nabla\mathcal{L}(\theta')\|_2^2 + \lambda^2 \|\theta - \theta'\|_2^2
$$

$$
- 2\lambda \langle\theta - \theta', \nabla\mathcal{L}(\theta) - \nabla\mathcal{L}(\theta')\rangle \bigg)
$$

$$
\implies \langle\theta - \theta', \nabla\mathcal{L}(\theta) - \nabla\mathcal{L}(\theta')\rangle \geq \frac{1}{2\lambda + \beta} \|\nabla\mathcal{L}(\theta) - \nabla\mathcal{L}(\theta')\|_2^2 + \frac{\lambda(\lambda + \beta)}{2\lambda + \beta} \|\theta - \theta'\|_2^2 .
$$

Substituting this in the above inequality, and noting that $\eta \leq \frac{2}{2\lambda+\beta}$, we get

$$
\begin{aligned}
\|T_k(\theta) - T_k(\theta')\|_2^2 &\leq \left(1 - \frac{2\eta\lambda(\lambda + \beta)}{2\lambda + \beta}\right) \|\theta - \theta'\|_2^2 + \left(\eta^2 - \frac{2\eta}{\beta + 2\lambda}\right) \|\nabla\mathcal{L}(\theta) - \nabla\mathcal{L}(\theta')\|_2^2 \\
&\leq \left(1 - \frac{2\eta\lambda(\lambda + \beta)}{2\lambda + \beta}\right) \|\theta - \theta'\|_2^2 + \left(\eta^2\lambda^2 - \frac{2\eta\lambda^2}{\beta + 2\lambda}\right) \|\theta - \theta'\|_2^2 \\
&= (1 - \eta\lambda)^2 \|\theta - \theta'\|_2^2 .
\end{aligned}
$$

$\square$

The second set of lemmas presented below describe how $\mathrm{R}_q(\Theta_t \| \Theta_t)$ evolves with time in both phases I and II. Central to our analysis is the following lemma which bounds the rate of change of Rényi divergence for any pair of diffusion process characterized by their Fokker-Planck equations.

**Lemma 29** (Rate of change of Rényi divergence (Chourasia et al., 2021)). *Let $V_t, V_t' : \mathbb{R}^d \to \mathbb{R}^d$ be two time dependent vector field such that $\max_{\theta \in \mathbb{R}^d} \|V_t(\theta) - V_t'(\theta)\|_2 \leq L$ for all $\theta \in \mathbb{R}^d$ and $t \geq 0$. For a diffusion process $(\Theta_t)_{t \geq 0}$ and $(\Theta_t')_{t \geq 0}$ defined by the Fokker-Planck equations*

$$\begin{cases} \partial_t \mu_t(\theta) = \operatorname{div}(\mu_t(\theta) V_t(\theta)) + \sigma^2 \Delta \mu_t(\theta) & and \\ \partial_t \mu_t'(\theta) = \operatorname{div}(\mu_t'(\theta) V_t'(\theta)) + \sigma^2 \Delta \mu_t'(\theta), \end{cases} \tag{91}$$

*respectively, where $\mu_t$ and $\mu_t$ are the densities of $\Theta_t$ and $\Theta_t'$, the rate of change of Rényi divergence between the two at any $t \geq 0$ is upper bounded as*

$$\partial_t R_q(\mu_t \| \mu_t') \leq \frac{qL^2}{2\sigma^2} - \frac{q\sigma^2}{2} \frac{I_q(\mu_t \| \mu_t')}{E_q(\mu_t \| \mu_t')}. \tag{92}$$

We will apply the above lemma to the Fokker-Planck equation (87) of our pair of tracing diffusion SDE (84) and solve the resulting differential inequality to prove the bound in (82). To assist our proof, we rely on the following lemma showing that our two tracing diffusion satisfy the LS inequality described in Definition G.1, which enables the use the inequality (79) in Lemma 23.

**Lemma 30.** *If loss $\ell(\theta; \mathbf{x})$ is convex and $\beta$-smooth, regularizer is $\mathbf{r}(\theta) = \frac{\lambda}{2} \|\theta\|_2^2$, and learning rate $\eta \leq \frac{2}{2\lambda + \beta}$, then the tracing diffusion $(\Theta_t)_{0 \leq t \leq \eta(K_A + iK_{\bar{A}})}$ and $(\Theta_t')_{0 \leq t \leq \eta(K_A + iK_{\bar{A}})}$ defined in (84) with $\Theta_0, \Theta_0' \sim \rho = \mathcal{N}\left(0, \frac{\sigma^2}{\lambda(1 - \eta\lambda/2)} \mathbb{I}_d\right)$ satisfy LS inequality with constant $\lambda(1 - \eta\lambda/2)$.*

*Proof.* For any iteration $0 \leq k < K_A + iK_{\bar{A}}$ in the extended run of Noisy-GD, and any $0 \leq s \leq \eta$, let's define two functions $\mathcal{L}_s, \mathcal{L}_s' : \mathbb{R}^d \to \mathbb{R}$ as follows:

$$\mathcal{L}_s = \frac{1 + s/\eta}{2} \mathcal{L}_{\mathcal{D}(k)} + \frac{1 - s/\eta}{2} \mathcal{L}_{\mathcal{D}'(k)}, \quad \text{and} \quad \mathcal{L}_s' = \frac{1 - s/\eta}{2} \mathcal{L}_{\mathcal{D}(k)} + \frac{1 + s/\eta}{2} \mathcal{L}_{\mathcal{D}'(k)}. \tag{93}$$

Since $\mathbf{r}(\cdot)$ is the $L2(\lambda)$ regularizer and $\ell(\theta; \mathbf{x})$ is convex and $\beta$-smoothness, both $\mathcal{L}_s$ and $\mathcal{L}_s'$ are $\lambda$-strongly convex and $(\lambda + \beta)$-smooth for all $0 \leq s \leq \eta$ and any $k$. We define maps $T_s, T_s' : \mathbb{R}^d \to \mathbb{R}^d$ as

$$T_s(\theta) = \theta - \eta \nabla \mathcal{L}_s(\theta), \quad \text{and} \quad T_s'(\theta) = \theta - \nabla \mathcal{L}_s'(\theta). \tag{94}$$

From a similar argument as in Lemma 28, both $T_s$ and $T_s'$ are $(1 - \eta\lambda)$-Lipschitz for learning rate $\eta \leq \frac{2}{2\lambda + \beta}$.

Note that the densities of $\Theta_t$ and $\Theta_t'$ of the tracing diffusion for $t = \eta k + s$ can be respectively expressed as

$$\mu_t = T_{s\#}(\mu_{\eta k}) \circledast \mathcal{N}(0, 2s\sigma^2 \mathbb{I}_d), \quad \text{and} \quad \mu_t' = T_{s\#}'(\mu_{\eta k}') \circledast \mathcal{N}(0, 2s\sigma^2 \mathbb{I}_d), \tag{95}$$

where $\mu_{\eta k}$ and $\mu_{\eta k}'$ represent the distributions of $\Theta_{\eta k}$ and $\Theta_{\eta k}'$. We prove the lemma via induction.

**Base step:** Since $\Theta_0, \Theta_0'$ are both Gaussian distributed with variance $\frac{\sigma^2}{\lambda(1 - \eta\lambda/2)}$, from Lemma 18 they satisfy LS inequality with constant $\lambda(1 - \eta\lambda/2)$.

**Induction step:** Suppose $\mu_{\eta k}$ and $\mu_{\eta k}'$ satisfy LS inequality with constant $\lambda(1 - \eta\lambda/2)$. Since equation (95) shows that $\mu_t, \mu_t'$ are both gaussian convolution on a pushforward distribution of $\mu_{\eta k}, \mu_{\eta k}'$ respectively over a Lipschitz function, from Lemma 18 and Lemma 19, both $\mu_t, \mu_t'$ satisfy LS inequality with constant

$$\left(\frac{(1 - \eta\lambda)^2}{\lambda(1 - \eta\lambda/2)} + 2s\right)^{-1} \geq \lambda(1 - \eta\lambda/2) \times \underbrace{[(1 - \eta\lambda)^2 + \lambda\eta(2 - \eta\lambda)]^{-1}}_{=1}, \tag{96}$$

for all $\eta k \leq t \leq \eta(k + 1)$. $\qquad \square$

We are now ready to prove the data-deletion bound in (82).

**Theorem 31** (Data-Deletion guarantee on $(A_{\text{Noisy-GD}}, \bar{A}_{\text{Noisy-GD}})$ under convexity). *Let the weight initialization distribution be $\rho = \mathcal{N}\left(0, \frac{\sigma^2}{\lambda(1 - \eta\lambda/2)}\right)$, the loss function $\ell(\theta; \mathbf{x})$ be convex, $\beta$-smooth,*

and $L$-Lipschitz, the regularizer be $\mathbf{r}(\theta) = \frac{\lambda}{2}\|\theta\|_2^2$, and learning rate be $\eta < \frac{1}{\lambda+\beta}$. Then Algorithm pair $(\mathrm{A}, \bar{\mathrm{A}})$ satisfies a $(q, \varepsilon_{\mathrm{dd}})$-data-deletion guarantee under all non-adaptive $r$-requesters for any noise variance $\sigma^2 > 0$ and $K_{\mathrm{A}} \geq 0$ if

$$K_{\bar{\mathrm{A}}} \geq \frac{2}{\eta\lambda} \log\left(\frac{4qL^2}{\lambda\varepsilon_{\mathrm{dd}}\sigma^2 n^2}\right). \tag{97}$$

*Proof.* Following the preceding discussion, to prove this theorem, it suffices to show that the inequality (82) holds under the stated conditions. Consider the Fokker-Planck equation described in Lemma 26 for the pair of tracing diffusions SDEs in (86): at any time $t$ in duration $\eta k < t \leq \eta(k+1)$ for any iteration $0 \leq k < K_{\mathrm{A}} + iK_{\bar{\mathrm{A}}}$,

$$\begin{cases} \partial_t \mu_t(\theta) &= \mathrm{div}\left(\mu_t(\theta)\mathbb{E}\left[\mathbf{g}_k(\Theta_{\eta k})|\Theta_t = \theta\right]\right) + \sigma^2\Delta\mu_t(\theta), \\ \partial_t \mu_t'(\theta) &= \mathrm{div}\left(\mu_t'(\theta)\mathbb{E}\left[-\mathbf{g}_k(\Theta_{\eta k}')\Big|\Theta_t' = \theta\right]\right) + \sigma^2\Delta\mu_t'(\theta), \end{cases} \tag{98}$$

where $\mu_t$ and $\mu_t'$ are the distribution of $\Theta_t$ and $\Theta_t'$. Since $\ell(\theta; \mathbf{x})$ is $L$-Lipschitz and for any $K_{\mathrm{A}} + (i-1)K_{\bar{\mathrm{A}}} \leq k < K_{\mathrm{A}} + iK_{\bar{\mathrm{A}}}$ we have $\mathcal{D}(k)[\mathrm{ind}] = \mathcal{D}'(k)[\mathrm{ind}]$, note from the definition of $\mathbf{g}_k(\theta)$ in (86) that

$$\left\|\mathbb{E}\left[\mathbf{g}_k(\Theta_{\eta k})|\Theta_t = \theta\right] - \mathbb{E}\left[-\mathbf{g}_k(\Theta_{\eta k}')\big|\Theta_t' = \theta\right]\right\|_2 \leq \begin{cases} \frac{2L}{n} & \text{if } k < K_{\mathrm{A}} + (i-1)K_{\bar{\mathrm{A}}} \\ 0 & \text{otherwise} \end{cases}. \tag{99}$$

Therefore, applying Lemma 29 to the above pair of Fokker-Planck equations gives that for any $t$ in duration $\eta k < t \leq \eta(k+1)$,

$$\partial_t \mathrm{R}_q\left(\mu_t\|\mu_t'\right) \leq \frac{2qL^2}{\sigma^2 n^2}\mathbb{1}\left\{t \leq \eta(K_{\mathrm{A}} + (i-1)K_{\bar{\mathrm{A}}})\right\} - \frac{q\sigma^2}{2}\frac{\mathrm{I}_q\left(\mu_t\|\mu_t'\right)}{\mathrm{E}_q\left(\mu_t\|\mu_t'\right)}. \tag{100}$$

Equation (100) suggests a phase change in the dynamics at iteration $k = K_{\mathrm{A}} + (i-1)K_{\bar{\mathrm{A}}}$. In phase I, the divergence bound increases with time due to the effect of the differing record in database pairs $(\mathcal{D}_j, \mathcal{D}_j')_{0 \leq j \leq i-1}$. In phase II however, the update request $u_i$ makes $\mathcal{D}_{i-1} \circ u_i = \mathcal{D}_{i-1}' \circ u_i$, and so doing gradient descent rapidly shrinks the divergence bound. This phase change is illustrated in the Figure 1.

For brevity, we denote $R(q, t) = \mathrm{R}_q\left(\mu_t\|\mu_t'\right)$. Since $\eta < \frac{1}{\lambda+\beta} < \frac{2}{2\lambda+\beta}$, from Lemma 30, the distribution $\mu_t'$ satisfies LS inequality with constant $\lambda(1 - \lambda\eta/2)$. So, we can apply Lemma 23 to simplify the above partial differential inequality as follows.

$$\partial_t R(q, t) + \lambda(1 - \lambda\eta/2)\left(\frac{R(q, t)}{q} + (q-1)\partial_q R(q, t)\right) \leq \frac{2qL^2}{\sigma^2 n^2}\mathbb{1}\left\{t \leq \eta(K_{\mathrm{A}} + (i-1)K_{\bar{\mathrm{A}}})\right\}. \tag{101}$$

For brevity, let constant $c_1 = \lambda(1 - \lambda\eta/2)$ and constant $c_2 = \frac{2L^2}{\sigma^2 n^2}$. We define $u(q, t) = \frac{R(q, t)}{q}$. Then,

$$\partial_t R(q, t) + c_1\left(\frac{R(q, t)}{q} + (q-1)\partial_q R(q, t)\right) \leq c_2 q \times \mathbb{1}\left\{t \leq \eta(K_{\mathrm{A}} + (i-1)K_{\bar{\mathrm{A}}})\right\}$$

$$\implies \partial_t u(q, t) + c_1 u(q, t) + c_1(q-1)\partial_q u(q, t) \leq c_2 \times \mathbb{1}\left\{t \leq \eta(K_{\mathrm{A}} + (i-1)K_{\bar{\mathrm{A}}})\right\}.$$

For some constant $\bar{q} > 1$, let $q(s) = (\bar{q} - 1)\exp\left[c_1\left\{s - \eta(K_{\mathrm{A}} + iK_{\bar{\mathrm{A}}})\right\}\right] + 1$ and $t(s) = s$. Note that $\frac{dq(s)}{ds} = c_1(q(s) - 1)$ and $\frac{dt(s)}{ds} = 1$. Therefore, for any $\eta k < s \leq \eta(k+1)$, the differential inequality followed along the path $u(s) = u(q(s), t(s))$ is

$$\frac{du(s)}{ds} + c_1 u(s) \leq c_2 \times \mathbb{1}\left\{t \leq \eta(K_{\mathrm{A}} + (i-1)K_{\bar{\mathrm{A}}})\right\} \tag{102}$$

$$\implies \frac{d}{ds}\{e^{c_1 s}u(s)\} \leq c_2 \times \mathbb{1}\left\{t \leq \eta(K_{\mathrm{A}} + (i-1)K_{\bar{\mathrm{A}}})\right\}. \tag{103}$$

Since the map $T_k(\cdot)$ in (85) is a differentiable bijection for $\eta < \frac{1}{\lambda+\beta}$ as per Lemma 28, note that Lemma 27 implies that $\lim_{s \to \eta k+} u(s) = u(\eta k)$. Therefore, we can directly integrate in the duration

$0 \le t \le \eta(K_A + iK_{\bar{A}})$ to get

$$[e^{c_1 s}u(s)]_0^{\eta(K_A + iK_{\bar{A}})} \le \int_0^{\eta(K_A + (i-1)K_{\bar{A}})} c_2 e^{c_1 s}\mathrm{d}s$$

$$\implies e^{c_1\eta(K_A + iK_{\bar{A}})}u(\eta(K_p + iK_u)) - u(0) \le \frac{c_2}{c_1}[e^{c_1\eta(K_A + (i-1)K_{\bar{A}})} - 1]$$

$$\implies u(\eta(K_A + iK_{\bar{A}})) \le \frac{c_2}{c_1}e^{-c_1\eta K_{\bar{A}}}. \qquad \text{(Since } u(0) = R(q(0),0)/q(0) = 0.)$$

Noting that $q(0) \ge 1$, on reverting the substitution, we get

$$R_{\bar{q}}\left(\mu_{\eta(K_A + iK_{\bar{A}})}\Big\|\mu'_{\eta(K_A + iK_{\bar{A}})}\right) \le \frac{2\bar{q}L^2}{\lambda\sigma^2 n^2(1 - \eta\lambda/2)}\exp\left(-\eta\lambda K_{\bar{A}}(1 - \eta\lambda/2)\right)$$

$$\le \frac{4\bar{q}L^2}{\lambda\sigma^2 n^2}\exp\left(-\frac{\eta\lambda K_u}{2}\right) \qquad \text{(Since } \eta < \tfrac{1}{\lambda+\beta})$$

Recall from our construction that $\mu_{\eta(K_A + iK_{\bar{A}})}$ and $\mu'_{\eta(K_A + iK_{\bar{A}})}$ are the distributions of $\bar{A}(\mathcal{D}_{i-1}, u_i, \hat{\Theta}_{i-1})$ and $\bar{A}(\mathcal{D}'_{i-1}, u_i, \hat{\Theta}'_{i-1})$ respectively. Therefore, choosing $K_{\bar{A}}$ as specified in the theorem statement concludes the proof. $\qquad\square$

Our next goal in this section is to provide utility guarantees for the algorithm pair $(A_{\text{Noisy-GD}}, \bar{A}_{\text{Noisy-GD}})$ in form of excess empirical risk bounds. For that, we introduce some additional auxiliary results first. The following Lemma 32 shows that excess empirical risks does not increase too much on replacing $r$ records in a database, and Lemma 33 provides a convergence guarantee on the excess empirical risk of Noisy-GD algorithm under convexity.

**Lemma 32.** *Suppose the loss function $\ell(\theta; \mathbf{x})$ is convex, $L$-Lipschitz, and $\beta$-smooth, and the regularizer is $\mathbf{r}(\theta) = \frac{\lambda}{2}\|\theta\|_2^2$. Then, the excess empirical risk of any randomly distributed parameter $\Theta$ for any database $\mathcal{D} \in \mathcal{X}^n$ after applying any edit request $u \in \mathcal{U}^r$ that modifies no more than $r$ records is bounded as*

$$\mathrm{err}(\Theta; \mathcal{D} \circ u) \le \left(1 + \frac{\beta}{\lambda}\right)\left[2\,\mathrm{err}(\Theta; \mathcal{D}) + \frac{16r^2 L^2}{\lambda n^2}\right]. \tag{104}$$

*Proof.* Let $\theta_{\mathcal{D}}^*$ and $\theta_{\mathcal{D}\circ u}^*$ be the minimizers of objectives $\mathcal{L}_{\mathcal{D}}(\cdot)$ and $\mathcal{L}_{\mathcal{D}\circ u}(\cdot)$ as defined in (**??**). From $\lambda$-strong convexity of the $\mathcal{L}_{\mathcal{D}}$,

$$\mathcal{L}_{\mathcal{D}}(\theta_{\mathcal{D}\circ u}^*) - \mathcal{L}_{\mathcal{D}}(\theta_{\mathcal{D}}^*) \ge \frac{\lambda}{2}\|\theta_{\mathcal{D}\circ u}^* - \theta_{\mathcal{D}}^*\|_2^2. \tag{105}$$

From optimality of $\theta_{\mathcal{D}\circ u}^*$ and $L$-Lipschitzness of $\ell(\theta; \mathbf{x})$, we have

$$\mathcal{L}_{\mathcal{D}}(\theta_{\mathcal{D}\circ u}^*) = \mathcal{L}_{\mathcal{D}\circ u}(\theta_{\mathcal{D}\circ u}^*) + \frac{1}{n}\left(\sum_{\mathbf{x}\in\mathcal{D}}\ell(\theta_{\mathcal{D}\circ u}^*; \mathbf{x}) - \sum_{\mathbf{x}\in\mathcal{D}\circ u}\ell(\theta_{\mathcal{D}\circ u}^*; \mathbf{x})\right)$$

$$\le \mathcal{L}_{\mathcal{D}\circ u}(\theta_{\mathcal{D}}^*) + \frac{1}{n}\left(\sum_{\mathbf{x}\in\mathcal{D}}\ell(\theta_{\mathcal{D}\circ u}^*; \mathbf{x}) - \sum_{\mathbf{x}\in\mathcal{D}\circ u}\ell(\theta_{\mathcal{D}\circ u}^*; \mathbf{x})\right)$$

$$= \mathcal{L}_{\mathcal{D}}(\theta_{\mathcal{D}}^*) + \frac{1}{n}\sum_{\mathbf{x}\in\mathcal{D}}(\ell(\theta_{\mathcal{D}\circ u}^*; \mathbf{x}) - \ell(\theta_{\mathcal{D}}^*; \mathbf{x})) + \frac{1}{n}\sum_{\mathbf{x}\in\mathcal{D}\circ u}(\ell(\theta_{\mathcal{D}}^*; \mathbf{x}) - \ell(\theta_{\mathcal{D}\circ u}^*; \mathbf{x}))$$

$$\le \mathcal{L}_{\mathcal{D}}(\theta_{\mathcal{D}}^*) + \frac{2rL}{n}\|\theta_{\mathcal{D}\circ u}^* - \theta_{\mathcal{D}}^*\|_2.$$

Combining the two inequalities give

$$\|\theta_{\mathcal{D}\circ u}^* - \theta_{\mathcal{D}}^*\|_2 \le \frac{4rL}{\lambda n}. \tag{106}$$

Therefore, from $(\lambda + \beta)$-smoothness of $\mathcal{L}_{\mathcal{D} \circ u}$ and $\lambda$-strong convexity of $\mathcal{L}_{\mathcal{D}}$, we have

$$\text{err}(\Theta; \mathcal{D} \circ u) = \mathbb{E}\left[\mathcal{L}_{\mathcal{D} \circ u}(\Theta) - \mathcal{L}_{\mathcal{D} \circ u}(\theta^*_{\mathcal{D} \circ u})\right]$$

$$\leq \frac{\lambda + \beta}{2} \mathbb{E}\left[\|\Theta - \theta^*_{\mathcal{D} \circ u}\|_2^2\right]$$

$$\leq (\lambda + \beta)\left[\mathbb{E}\left[\|\Theta - \theta^*_{\mathcal{D}}\|_2^2\right] + \|\theta^*_{\mathcal{D}} - \theta^*_{\mathcal{D} \circ u}\|_2^2\right]$$

$$\leq \left(1 + \frac{\beta}{\lambda}\right)\left[2\mathbb{E}\left[\mathcal{L}_{\mathcal{D}}(\Theta) - \mathcal{L}_{\mathcal{D}}(\theta^*_{\mathcal{D}})\right] + \frac{16 r^2 L^2}{\lambda n^2}\right].$$

$\square$

**Lemma 33** (Accuracy of Noisy-GD). *For convex, L-Lipschitz, and, $\beta$-smooth loss function $\ell(\theta; \mathbf{x})$ and regularizer $\mathbf{r}(\theta) = \frac{\lambda}{2}\|\theta\|_2^2$, if learning rate $\eta < \frac{1}{\lambda + \beta}$, the excess empirical risk of $\Theta_{\eta K} = \text{Noisy-GD}(\mathcal{D}, \Theta_0, K)$ for any $\mathcal{D} \in \mathcal{X}^n$ is bounded as*

$$\text{err}(\Theta_{\eta K}; \mathcal{D}) \leq \text{err}(\Theta_0; \mathcal{D}) e^{-\lambda \eta K / 2} + \left(1 + \frac{\beta}{\lambda}\right) d\sigma^2. \tag{107}$$

*Proof.* Let $\Theta_{\eta k}$ denote the $k$th iteration parameter of Noisy-GD run. Recall that $k + 1$th noisy gradient update step is

$$\Theta_{\eta(k+1)} = \Theta_{\eta k} - \eta \nabla \mathcal{L}_{\mathcal{D}}(\Theta_{\eta k}) + \sqrt{2\eta \sigma^2} \mathbf{Z}_k. \tag{108}$$

From $(\beta + \lambda)$-smoothness of $\mathcal{L}_{\mathcal{D}}$, we have

$$\mathcal{L}_{\mathcal{D}}(\Theta_{\eta(k+1)}) \leq \mathcal{L}_{\mathcal{D}}(\Theta_{\eta k}) + \left\langle \nabla \mathcal{L}_{\mathcal{D}}(\Theta_{\eta k}), \Theta_{\eta(k+1)} - \Theta_{\eta k}\right\rangle + \frac{\beta + \lambda}{2}\|\Theta_{\eta(k+1)} - \Theta_{\eta k}\|_2^2$$

$$= \mathcal{L}_{\mathcal{D}}(\Theta_{\eta k}) - \eta \|\nabla \mathcal{L}_{\mathcal{D}}(\Theta_{\eta k})\|_2^2 + \sqrt{2\eta \sigma^2}\left\langle \nabla \mathcal{L}_{\mathcal{D}}(\Theta_{\eta k}), \mathbf{Z}_k\right\rangle$$

$$+ \frac{\eta^2 (\beta + \lambda)}{2}\|\nabla \mathcal{L}_{\mathcal{D}}(\Theta_{\eta k})\|_2^2 + \eta \sigma^2 (\beta + \lambda)\|\mathbf{Z}_k\|_2^2$$

$$- \eta \sqrt{2\eta \sigma^2}(\beta + \lambda)\left\langle \nabla \mathcal{L}_{\mathcal{D}}(\Theta_{\eta k}), \mathbf{Z}_k\right\rangle$$

On taking expectation over the joint distribution of $\Theta_{\eta k}, \Theta_{\eta(k+1)}, \mathbf{Z}_k$, the above simplifies to

$$\mathbb{E}\left[\mathcal{L}_{\mathcal{D}}(\Theta_{\eta(k+1)})\right] \leq \mathbb{E}\left[\mathcal{L}_{\mathcal{D}}(\Theta_{\eta k})\right] - \eta\left(1 - \frac{\eta(\lambda + \beta)}{2}\right)\mathbb{E}\left[\|\nabla \mathcal{L}_{\mathcal{D}}(\Theta_{\eta k})\|_2^2\right] + \eta d\sigma^2(\beta + \lambda). \tag{109}$$

Let $\theta^*_{\mathcal{D}} = \underset{\theta \in \mathbb{R}^d}{\arg\min} \mathcal{L}_{\mathcal{D}}(\theta)$. From $\lambda$-strong convexity of $\mathcal{L}_{\mathcal{D}}$, for any $\theta \in \mathbb{R}^d$, we have

$$\|\nabla \mathcal{L}_{\mathcal{D}}(\theta)\|_2^2 \geq 2\lambda(\mathcal{L}_{\mathcal{D}}(\theta) - \mathcal{L}_{\mathcal{D}}(\theta^*_{\mathcal{D}})). \tag{110}$$

Let $\gamma = \lambda \eta(2 - \eta(\lambda + \beta))$. Plugging this in the above inequality, and substracting $\mathcal{L}_{\mathcal{D}}(\theta^*_{\mathcal{D}})$ on both sides, for $\eta < \frac{1}{\lambda + \beta}$, we get

$$\mathbb{E}\left[\mathcal{L}_{\mathcal{D}}(\Theta_{\eta(k+1)}) - \mathcal{L}_{\mathcal{D}}(\theta^*_{\mathcal{D}})\right] \leq (1 - \gamma)\mathbb{E}\left[\mathcal{L}_{\mathcal{D}}(\Theta_{\eta k}) - \mathcal{L}_{\mathcal{D}}(\theta^*_{\mathcal{D}})\right] + \eta d\sigma^2(\beta + \lambda)$$

$$\leq (1 - \gamma)^{k+1}\mathbb{E}\left[\mathcal{L}_{\mathcal{D}}(\Theta_0) - \mathcal{L}_{\mathcal{D}}(\theta^*)\right] + \eta d\sigma^2(\beta + \lambda)(1 + \cdots + (1 - \gamma)^{k+1})$$

$$\leq e^{-\gamma(k+1)/2}\mathbb{E}\left[\mathcal{L}_{\mathcal{D}}(\Theta_0) - \mathcal{L}_{\mathcal{D}}(\theta^*_{\mathcal{D}})\right] + \frac{\eta d\sigma^2(\beta + \lambda)}{\gamma}.$$

For $\eta < \frac{1}{\lambda + \beta}$, note that $\gamma \geq \lambda \eta$, and so

$$\text{err}(\Theta_{\eta K}; \mathcal{D}) \leq \text{err}(\Theta_0; \mathcal{D}) e^{-\lambda \eta K / 2} + \left(1 + \frac{\beta}{\lambda}\right) d\sigma^2. \tag{111}$$

$\square$

Finally, we are ready to prove our main Theorem 6 showing that algorithm pair $(\mathrm{A}_{\text{Noisy-GD}}, \bar{\mathrm{A}}_{\text{Noisy-GD}})$ solves the data-deletion problem as described in Section 4. We basically combine the RDP guarantee in Theorem 25, non-adaptive data-deletion guarantee in Theorem 31, and prove excess empirical risk bound using Lemma 33 and Lemma 32.

**Theorem 6** (Accuracy, privacy, deletion, and computation tradeoffs)**.** *Let constants $\lambda, \beta, L > 0$, $q > 1$, and $0 < \varepsilon_{\text{dd}} \leq \varepsilon_{\text{dp}}$. Define constant $\kappa = \frac{\lambda+\beta}{\lambda}$. Let the loss function $\ell(\theta; \mathbf{x})$ be twice differentiable, convex, $L$-Lipschitz, and $\beta$-smooth, the regularizer be $\mathbf{r}(\theta) = \frac{\lambda}{2}\|\theta\|_2^2$. If the learning rate be $\eta = \frac{1}{2(\lambda+\beta)}$, the gradient noise variance is $\sigma^2 = \frac{4qL^2}{\lambda\varepsilon_{\text{dp}}n^2}$, and the weight initialization distribution is $\rho = \mathcal{N}\left(0, \frac{\sigma^2}{\lambda(1-\eta\lambda/2)\mathbb{I}_d}\right)$, then*

> **(1.)** *both $\mathrm{A}_{\text{Noisy-GD}}$ and $\bar{\mathrm{A}}_{\text{Noisy-GD}}$ are $(q, \varepsilon_{\text{dp}})$-RDP for any $K_A, K_{\bar{A}} \geq 0$,*
>
> **(2.)** *pair $(\mathrm{A}_{\text{Noisy-GD}}, \bar{\mathrm{A}}_{\text{Noisy-GD}})$ satisfies $(q, \varepsilon_{\text{dd}})$-data-deletion all non-adaptive $r$-requesters*
> $$\text{if} \quad K_{\bar{A}} \geq 4\kappa \log \frac{\varepsilon_{\text{dp}}}{\varepsilon_{\text{dd}}}, \tag{112}$$
>
> **(3.)** *and all models in $(\hat{\Theta}_i)_{i\geq 0}$ produced by $(\mathrm{A}_{\text{Noisy-GD}}, \bar{\mathrm{A}}_{\text{Noisy-GD}}, \mathcal{Q})$ on any $\mathcal{D}_0 \in \mathcal{X}^n$, where $\mathcal{Q}$ is any $r$-requester, have an excess empirical risk $\text{err}(\hat{\Theta}_i; \mathcal{D}_i) = O\left(\frac{qd}{\varepsilon_{\text{dp}}n^2}\right)$*
> $$\text{if} \quad K_A \geq 4\kappa \log\left(\frac{\varepsilon_{\text{dp}}n^2}{4qd}\right), \quad \text{and} \quad K_{\bar{A}} \geq 4\kappa \log \max\left\{5\kappa, \frac{8\varepsilon_{\text{dp}}r^2}{qd}\right\}. \tag{113}$$

*Proof.* **(1.) Privacy.** By Theorem 25, the Noisy-GD with $K$ iterations will be $(q, \varepsilon_{\text{dp}})$-RDP for the stated choice of loss function, regularizer, and learning rate as long as $\sigma^2 \geq \frac{4qL^2}{\lambda\varepsilon_{\text{dp}}n^2}\left(1 - e^{-\lambda\eta K/2}\right)$. Therefore, if we set $\sigma^2 = \frac{4qL^2}{\lambda\varepsilon_{\text{dp}}n^2}$, Noisy-GD is $(q, \varepsilon_{\text{dp}})$-RDP for any $K$. For the same $\sigma^2$, both $\mathrm{A}_{\text{Noisy-GD}}$ and $\bar{\mathrm{A}}_{\text{Noisy-GD}}$ are also $(q, \varepsilon_{\text{dp}})$-RDP for any $K_A$ and $K_{\bar{A}}$ as they run Noisy-GD on respective databases for generating the output.

**(2.) Deletion.** By Theorem 31, for the stated choice of loss function, regularizer, learning rate, and weight initialization distribution, the algorithm pair $(\mathrm{A}_{\text{Noisy-GD}}, \bar{\mathrm{A}}_{\text{Noisy-GD}})$ satisfies $(q, \varepsilon_{\text{dd}})$-data-deletion under all non-adaptive $r$-requesters $\mathcal{Q}$ if $K_{\bar{A}} \geq \frac{2}{\eta\lambda} \log\left(\frac{4qL^2}{\lambda\varepsilon_{\text{dd}}\sigma^2 n^2}\right)$. By plugging in $\sigma^2 = \frac{4qL^2}{\lambda\varepsilon_{\text{dp}}n^2}$ and $\eta = \frac{1}{2(\lambda+\beta)}$, this constraint simplifies to $K_{\bar{A}} \geq 4\kappa \log \frac{\varepsilon_{\text{dp}}}{\varepsilon_{\text{dd}}}$.

**(3.) Accuracy.** We prove the induction hypothesis that under the conditions stated in the theorem, $\text{err}(\hat{\Theta}_i; \mathcal{D}_i) \leq \frac{10\kappa qdL^2}{\lambda\varepsilon_{\text{dp}}n^2}$ for all $i \geq 0$.

*Base case:* The minimizer $\theta^*_{\mathcal{D}_0}$ of $\mathcal{L}_{\mathcal{D}_0}$ satisfies

$$\nabla\mathcal{L}_{\mathcal{D}_0}(\theta^*_{\mathcal{D}_0}) = \frac{1}{n}\sum_{\mathbf{x}\in\mathcal{D}_0}\nabla\ell(\theta^*_{\mathcal{D}_0}; \mathbf{x}) - \lambda\theta^*_{\mathcal{D}_0} = 0 \implies \|\theta^*_{\mathcal{D}_0}\|_2 \leq \frac{L}{\lambda}. \tag{114}$$

As a result, the excess empirical risk of initialization weights $\Theta_0 \sim \rho = \mathcal{N}\left(0, \frac{\sigma^2}{\lambda(1-\eta\lambda/2)\mathbb{I}_d}\right)$ on $\mathcal{L}_{\mathcal{D}_0}$ is bounded as

$$
\begin{aligned}
\text{err}(\Theta_0; \mathcal{D}_0) &= \mathbb{E}\left[\mathcal{L}_{\mathcal{D}_0}(\Theta_0) - \mathcal{L}_{\mathcal{D}_0}(\theta^*_{\mathcal{D}_0})\right] \\
&\leq \frac{(\lambda+\beta)}{2}\mathbb{E}\left[\|\Theta_0 - \theta^*_{\mathcal{D}_0}\|_2^2\right] && \text{(From $(\lambda+\beta)$-smoothness of $\mathcal{L}_{\mathcal{D}_0}$)} \\
&= \frac{(\lambda+\beta)}{2}\left[\|\theta^*_{\mathcal{D}_0}\|_2^2 + \mathbb{E}\left[\|\Theta_0\|_2^2\right] - 2\mathbb{E}\left[\langle\theta^*_{\mathcal{D}_0}, \Theta_0\rangle\right]\right] \\
&\leq \left(1 + \frac{\beta}{\lambda}\right)\left[\frac{L^2}{2\lambda} + \frac{\sigma^2 d}{2-\lambda\eta}\right] && \text{(From (114) and $\mathbb{E}\left[\|\mathbf{Z}\|_2^2\right] = d$ if $\mathbf{Z}\sim\mathcal{N}(0, \mathbb{I}_d)$.)} \\
&\leq \kappa\left[\frac{L^2}{2\lambda} + d\sigma^2\right].
\end{aligned}
$$

Since $\hat{\Theta}_0 = A_{\text{Noisy-GD}}(\mathcal{D}_0) = \text{Noisy-GD}(\mathcal{D}_0, \Theta_0, K_A)$, by Lemma 33, running $K_A \geq 2\kappa \log\left(\frac{\varepsilon_{\text{dp}} n^2}{4qd}\right)$ iterations gives

$$\text{err}(\hat{\Theta}_0; \mathcal{D}_0) \leq \text{err}(\Theta_0; \mathcal{D}_0) e^{-\lambda \eta K_A/2} + \kappa d\sigma^2$$

$$\leq \kappa \left[\frac{L^2}{2\lambda} + d\sigma^2\right] e^{-\lambda \eta K_A/2} + \kappa d\sigma^2$$

$$\leq \frac{\kappa L^2}{2\lambda} e^{-\lambda \eta K_A/2} + \frac{8\kappa qdL^2}{\lambda \varepsilon_{\text{dp}} n^2} \qquad \text{(On substituting } \sigma^2 = \frac{4qL^2}{\lambda \varepsilon_{\text{dp}} n^2}\text{)}$$

$$\leq \frac{10\kappa qdL^2}{\lambda \varepsilon_{\text{dp}} n^2} \qquad \text{(Since } K_A \geq 4\kappa \log\left(\frac{\varepsilon_{\text{dp}} n^2}{4qd}\right)\text{)}$$

*Induction step:* Assume that $\text{err}(\hat{\Theta}_{i-1}; \mathcal{D}_{i-1}) \leq \frac{10\kappa qdL^2}{\lambda \varepsilon_{\text{dp}} n^2}$. Since $\hat{\Theta}_i = \bar{A}_{\text{Noisy-GD}}(\mathcal{D}_{i-1}, u_i, \hat{\Theta}_{i-1}) = \text{Noisy-GD}(\mathcal{D}_i, \hat{\Theta}_{i-1}, K_{\bar{A}})$, by Lemma 33 and Lemma 32, running $K_{\bar{A}} \geq 2\kappa \log \max\left\{5\kappa, \frac{8r^2}{qd}\right\}$ iterations gives

$$\text{err}(\hat{\Theta}_i; \mathcal{D}_i) \leq \kappa \left[2\text{err}(\hat{\Theta}_{i-1}; \mathcal{D}_{i-1}) + \frac{16r^2 L^2}{\lambda n^2}\right] e^{-\lambda \eta K_{\bar{A}}/2} + \kappa d\sigma^2$$

$$\leq \kappa \left[\frac{20\kappa qdL^2}{\lambda \varepsilon_{\text{dp}} n^2} + \frac{16r^2 L^2}{\lambda n^2}\right] e^{-\lambda \eta K_{\bar{A}}/2} + \frac{4\kappa qdL^2}{\lambda \varepsilon_{\text{dp}} n^2} \qquad \text{(Substituting } \sigma^2\text{)}$$

$$\leq \frac{16\kappa r^2 L^2}{\lambda n^2} e^{-\lambda \eta K_{\bar{A}}/2} + \frac{8\kappa qdL^2}{\lambda \varepsilon_{\text{dp}} n^2} \qquad \text{(Since } K_{\bar{A}} \geq 4\kappa \log(5\kappa)\text{)}$$

$$\leq \frac{10\kappa qdL^2}{\lambda \varepsilon_{\text{dp}} n^2} \qquad \text{(Since } K_{\bar{A}} \geq 4\kappa \log \frac{8\varepsilon_{\text{dp}} r^2}{qd}\text{)}$$

$\square$

### G.5 PROOFS FOR SUBSECTION 5.2

In this Appendix, we provide a proof of our data-deletion and utility guarantee in Theorem 7 which applies to non-convex but bounded losses $\ell(\theta; \mathbf{x})$ under $L2$ regularizer $\mathbf{r}(\theta)$. Suppose $\mathcal{D}_0 \in \mathcal{X}^n$ is an arbitrary database, $\mathcal{Q}$ is any non-adaptive $r$-requester, and $(\hat{\Theta}_i)_{i \geq 0}$ is the model sequence generated by the interaction of $(A_{\text{Noisy-GD}}, \bar{A}_{\text{Noisy-GD}}, \mathcal{Q})$. Our first goal will be to prove $(q, \varepsilon_{\text{dd}})$-data deletion guarantee on $(A_{\text{Noisy-GD}}, \bar{A}_{\text{Noisy-GD}})$ and we will later use it for arguing utility as well. Recall from Definition 4.1 that to prove $(q, \varepsilon_{\text{dd}})$-data-deletion, we need to construct a map $\pi_i^{\mathcal{Q}} : \mathcal{X}^n \to \mathcal{O}$ such that for all $i \geq 1$ and any $u_i \in \mathcal{U}^r$,

$$R_q\left(\bar{A}(\mathcal{D}_{i-1}, u_i, \hat{\Theta}_{i-1}) \middle\| \pi_i^{\mathcal{Q}}(\mathcal{D}_0 \circ \langle \text{ind}, \mathbf{y} \rangle)\right) \leq \varepsilon_{\text{dd}} \quad \text{for all } \langle \text{ind}, \mathbf{y} \rangle \in u_i. \tag{115}$$

Our construction of $\pi_i^u$ for this proof is completely different from the one described in Appendix G.4. As discussed in Remark 3, since $\mathcal{Q}$ is non-adaptive, it suffices to show that there exists a map $\pi : \mathcal{X}^n \to \mathcal{O}$ such that for all $i \geq 1$,

$$R_q\left(\bar{A}(\mathcal{D}_{i-1}, u_i, \hat{\Theta}_{i-1}) \middle\| \pi(\mathcal{D}_i)\right) \leq \varepsilon_{\text{dd}}, \tag{116}$$

for all $\mathcal{D}_0 \in \mathcal{X}^n$ and all edit sequences $(u_i)_{i \geq 1}$ from $\mathcal{U}^r$.

Our mapping of choice for the purpose is the Gibbs distribution with the following density:

$$\pi(\mathcal{D})(\theta) \propto e^{-\mathcal{L}_{\mathcal{D}}(\theta)/\sigma^2}. \tag{117}$$

The high-level intuition for this construction is that Noisy-GD can be interpreted as Unadjusted Langevin Algorithm (ULA) (Roberts & Tweedie, 1996), which is a discretization of the Langevin diffusion (described in eqn. (63)) that eventually converges to this Gibbs distribution (see Appendix G.1 for a quick refresher). However, showing a convergence for ULA (in indistinguishability notions like Rényi divergence) to this Gibbs distribution, especially in form of non-asymptotic bounds on the

mixing time and discretization error has been a long-standing open problem. Recent breakthrough results by Vempala & Wibisono (2019) followed by Chewi et al. (2021) resolved this problem with an elegant argument, relying solely on isoperimetric assumptions over (117) that hold for non-convex losses. Our data-deletion argument leverages this rapid convergence result to basically show that once Noisy-GD reaches near-indistinguishability to its Gibbs mixing distribution, maintaining indistinguishability to subsequent Gibbs distribution corresponding to database edits require much fewer Noisy-GD iterations than fresh retraining (i.e. data deletion is faster than retraining).

We start by presenting Chewi et al. (2021)'s convegence argument adapted to our Noisy-GD formulation, with a slighlty tighter analysis that results in a $\log(q)$ improvement in the discretization error over the original. Consider the discrete stochastic process $(\Theta_{\eta k})_{0 \leq k \leq K}$ induced by parameter update step in Noisy-GD algorithm when run for $K$ iterations on a database $\mathcal{D}$ with an arbitrary start distribution $\Theta_0 \sim \mu_0$. We interpolate each discrete update from $\Theta_{\eta k}$ to $\Theta_{\eta(k+1)}$ via a diffusion process $\Theta_t$ defined over time $\eta k \leq t \leq \eta(k+1)$ as

$$\Theta_t = \Theta_{\eta k} - (t - \eta k)\nabla \mathcal{L}_{\mathcal{D}}(\Theta_{\eta k}) + \sqrt{2\sigma^2}(\mathbf{Z}_t - \mathbf{Z}_{\eta k}), \tag{118}$$

where $\mathbf{Z}_t$ is a Weiner process. Note that if $\Theta_{\eta k}$ models the parameter distribution after the $k^{th}$ update, then $\Theta_{\eta(k+1)}$ models the parameter distribution after the $k+1^{th}$ update. On repeating this construction for all $k = 0, \cdots, K$, we get a *tracing diffusion* $\{\Theta_t\}_{t \geq 0}$ for Noisy-GD (which is different from (84)). We denote the distribution of random variable $\Theta_t$ with $\mu_t$. The tracing diffusion during the duration $\eta k \leq t \leq \eta(k+1)$ is characterized by the following Fokker-Planck equation.

**Lemma 34** (Proposition 14 (Chewi et al., 2021)). *For tracing diffusion $\Theta_t$ defined in (118), the equivalent Fokker-Planck equation in the interval $\eta k \leq t \leq \eta(k+1)$ is*

$$\partial_t \mu_t(\theta) = \text{div}\left(\left\{\mathbb{E}\left[\nabla \mathcal{L}_{\mathcal{D}}(\Theta_{\eta k}) - \nabla \mathcal{L}_{\mathcal{D}}(\Theta_t)|\Theta_t = \theta\right] + \sigma^2 \nabla \log \frac{\mu_t(\theta)}{\pi(\mathcal{D})(\theta)}\right\} \mu_t(\theta)\right), \tag{119}$$

*where $\pi(\mathcal{D})$ is the Gibbs distribution defined in (117).*

*Proof.* Conditioned on observing parameter $\Theta_{\eta k} = \theta_{\eta k}$, the process $(\Theta_t)_{\eta k \leq t \leq \eta(k+1)}$ is a Langevin diffusion along a constant Vector field $\nabla \mathcal{L}_{\mathcal{D}}(\theta_{\eta k})$. Therefore, the conditional probability density $\mu_{t|\eta k}(\cdot|\theta_{\eta k})$ of $\Theta_t$ given $\theta_{\eta k}$ follows the following Fokker-Planck equation.

$$\partial_t \mu_{t|\eta k}(\cdot|\theta_{\eta k}) = \sigma^2 \Delta \mu_{t|\eta k}(\cdot|\theta_{\eta k}) + \text{div}\left(\mu_{t|\eta k}(\cdot|\theta_{\eta k})\nabla \mathcal{L}_{\mathcal{D}}(\theta_{\eta k})\right) \tag{120}$$

Taking expectation over $\Theta_{\eta k}$, we have

$$\partial_t \mu_t(\cdot) = \int \mu_{\eta k}(\theta_{\eta k}) \left\{\sigma^2 \Delta \mu_{t|\eta k}(\cdot|\theta_{\eta k}) + \text{div}\left(\mu_{t|\eta k}(\cdot|\theta_{\eta k})\nabla \mathcal{L}_{\mathcal{D}}(\theta_{\eta k})\right)\right\} d\theta_{\eta k}$$

$$= \sigma^2 \Delta \mu_t(\cdot) + \text{div}\left(\mu_t(\cdot)\nabla \mathcal{L}_{\mathcal{D}}(\cdot)\right) + \text{div}\left(\mu_t(\cdot)\int \left[\nabla \mathcal{L}_{\mathcal{D}}(\theta_{\eta k}) - \nabla \mathcal{L}_{\mathcal{D}}(\cdot)\right]\mu_{\eta k|t}(\theta_{\eta k}|\cdot)d\theta_{\eta k}\right)$$

$$= \sigma^2 \text{div}\left(\mu_t(\cdot)\nabla \log \frac{\mu_t(\cdot)}{\pi(\mathcal{D})(\cdot)}\right) + \text{div}\left(\mathbb{E}\left[\nabla \mathcal{L}_{\mathcal{D}}(\Theta_{\eta k}) - \nabla \mathcal{L}_{\mathcal{D}}(\cdot)|\Theta_t = \cdot\right]\mu_t(\cdot)\right),$$

where $\mu_{\eta k|t}$ is the conditional density of $\Theta_{\eta k}$ given $\Theta_t$. For the last equality, we have used the fact that $\nabla \mathcal{L}_{\mathcal{D}} = -\sigma^2 \nabla \log \pi(\mathcal{D})$ from (117). $\square$

The following lemma provides a partial differential inequality that bounds the rate of change in Rényi divergence $R_q(\mu_t \| \pi(\mathcal{D}))$ using Fokker-Planck equation (119) of Noisy GD's tracing diffusion.

**Lemma 35** (Proposition 15 (Chewi et al., 2021)). *Let $\rho_t := \mu_t/\pi(\mathcal{D})$ where $\pi(\mathcal{D})$ is the Gibbs distribution defined in (117) and $\psi_t := \rho_t^{q-1}/E_q(\rho_t\|\pi(\mathcal{D}))$. The rate of change in $R_q(\mu_t\|\pi(\mathcal{D}))$ along racing diffusion in time $\eta k \leq t \leq \eta(k+1)$ is bounded as*

$$\partial_t R_q(\mu_t\|\pi(\mathcal{D})) \leq -\frac{3q\sigma^2}{4}\frac{I_q(\mu_t\|\pi(\mathcal{D}))}{E_q(\mu_t\|\pi(\mathcal{D}))} + \frac{q}{\sigma^2}\mathbb{E}\left[\psi_t(\Theta_t)\|\nabla \mathcal{L}_{\mathcal{D}}(\Theta_{\eta k}) - \nabla \mathcal{L}_{\mathcal{D}}(\Theta_t)\|_2^2\right]. \tag{121}$$

*Proof.* For brevity, let $\Delta_t(\cdot) = \mathbb{E}\left[\nabla\mathcal{L}_\mathcal{D}(\Theta_{\eta k}) - \nabla\mathcal{L}_\mathcal{D}(\Theta_t)|\Theta_t = \cdot\right]$ in context of this proof. From Lebinz integral rule, we have

$$
\begin{aligned}
\partial_t \mathrm{R}_q\left(\mu_t \| \pi(\mathcal{D})\right) &= \frac{q}{(q-1)\mathrm{E}_q\left(\mu_t \| \pi(\mathcal{D})\right)} \int \left(\frac{\mu_t}{\pi(\mathcal{D})}\right)^{q-1} \partial_t \mu_t \mathrm{d}\theta \\
&= \frac{q}{(q-1)\mathrm{E}_q\left(\mu_t \| \pi(\mathcal{D})\right)} \int \rho_t^{q-1} \mathrm{div}\left(\left\{\Delta_t + \sigma^2\nabla\log\rho_t\right\}\mu_t\right)\mathrm{d}\theta \quad \text{(From (119))} \\
&= -\frac{q}{(q-1)\mathrm{E}_q\left(\mu_t \| \pi(\mathcal{D})\right)} \int \left\langle\nabla\left(\rho_t^{q-1}\right), \Delta_t + \sigma^2\nabla\log\rho_t\right\rangle \mu_t \mathrm{d}\theta \\
&= -\frac{q}{\mathrm{E}_q\left(\mu_t \| \pi(\mathcal{D})\right)} \int \rho_t^{q-2}\left\langle\nabla\rho_t, \Delta_t + \sigma^2\frac{\nabla\rho_t}{\rho_t}\right\rangle \mu_t \mathrm{d}\theta \\
&= -\frac{q}{\mathrm{E}_q\left(\mu_t \| \pi(\mathcal{D})\right)}\left\{\sigma^2\mathrm{I}_q\left(\mu_t \| \pi(\mathcal{D})\right) + \frac{2}{q}\underbrace{\mathbb{E}_{\mu_t}\left[\rho_t^{q/2-1}\left\langle\nabla\left(\rho_t^{q/2}\right), \Delta_t\right\rangle\right]}_{\stackrel{\text{def}}{=}F_1}\right\}
\end{aligned}
$$

$$\text{(From (40))}$$

Note that the expectation in $\Delta_t(\cdot)$ is over the conditional distribution $\mu_{\eta k|t}$ while the expectation in $F_1$ is over $\mu_t$. Therefore, we can combine the two to get an expectation over the unconditional joint distribution over $\Theta_t$ and $\Theta_{\eta k}$ as follows.

$$
\begin{aligned}
-F_1 &= \mathbb{E}_{\Theta_t\sim\mu_t}\left[\rho_t^{q/2-1}(\Theta_t)\left\langle\nabla\left(\rho_t^{q/2}\right)(\Theta_t), \mathbb{E}_{\Theta_{\eta k}\sim\mu_{\eta k|t}}\left[\nabla\mathcal{L}_\mathcal{D}(\Theta_t) - \nabla\mathcal{L}_\mathcal{D}(\Theta_{\eta k})\right]\right\rangle\right] \\
&= \mathbb{E}_{\mu_{\eta k,t}}\left[\rho_t^{q/2-1}(\Theta_t)\left\langle\nabla\left(\rho_t^{q/2}\right)(\Theta_t), \nabla\mathcal{L}_\mathcal{D}(\Theta_t) - \nabla\mathcal{L}_\mathcal{D}(\Theta_{\eta k})\right\rangle\right] \\
&\leq \frac{\sigma^2}{2q}\mathbb{E}\left[\rho_t^{-1}(\Theta_t)\left\|\nabla\left(\rho_t^{q/2}\right)(\Theta_t)\right\|_2^2\right] + \frac{q}{2\sigma^2}\mathbb{E}\left[\rho_t^{q-1}(\Theta_t)\left\|\nabla\mathcal{L}_\mathcal{D}(\Theta_t) - \nabla\mathcal{L}_{B_k}(\Theta_{\eta k})\right\|_2^2\right] \\
&= \frac{q\sigma^2}{8}\mathrm{I}_q\left(\rho_t \| \mu\right) + \frac{q}{2\sigma^2}\mathbb{E}\left[\rho_t^{q-1}(\Theta_t)\left\|\nabla\mathcal{L}_\mathcal{D}(\Theta_t) - \nabla\mathcal{L}_{B_k}(\Theta_{\eta k})\right\|_2^2\right] \quad \text{(From (40))}
\end{aligned}
$$

Substituting it in the preceding inequality proves the proposition. $\qquad\square$

We need to solve the PDI (121) to get a convergence bound for Noisy-GD. To help in that, we first introduce the change of measure inequalities shown in Chewi et al. (2021).

**Lemma 36** (Change of measure inequality (Chewi et al., 2021)). *If $\ell(\theta; \mathbf{x})$ is $\beta$-smooth, and regularizer is $\mathbf{r}(\theta) = \frac{\lambda}{2}\|\theta\|_2^2$, then for any probability density $\mu$ on $\mathbb{R}^d$,*

$$
\mathbb{E}_\mu\left[\|\nabla\mathcal{L}_\mathcal{D}\|_2^2\right] \leq 4\sigma^4\mathbb{E}_{\pi(\mathcal{D})}\left[\left\|\nabla\sqrt{\frac{\mu}{\pi(\mathcal{D})}}\right\|_2^2\right] + 2d\sigma^2(\beta + \lambda), \tag{122}
$$

*where $\pi(\mathcal{D})$ is the Gibbs distribution defined in (117).*

*Proof.* Consider the Langevin diffusion (63) described in Appendix G.1 over the potential $\mathcal{L}_\mathcal{D}$. The Gibbs distribution $\pi(\mathcal{D})$ is its stationary distribution, and the diffusion's infintesimal generator $\mathcal{G}$ applied on the $\mathcal{L}_\mathcal{D}$ gives

$$
\mathcal{G}\mathcal{L}_\mathcal{D} = \sigma^2\Delta\mathcal{L}_\mathcal{D} - \|\nabla\mathcal{L}_\mathcal{D}\|_2^2. \tag{123}
$$

Therefore,

$$\underset{\mu}{\mathbb{E}}\left[\|\nabla\mathcal{L}_{\mathcal{D}}\|_2^2\right] = \sigma^2\underset{\mu}{\mathbb{E}}\left[\Delta\mathcal{L}_{\mathcal{D}}\right] - \underset{\mu}{\mathbb{E}}\left[\mathcal{G}\mathcal{L}_{\mathcal{D}}\right] \qquad\qquad \text{(From (123))}$$

$$\leq d\sigma^2(\beta+\lambda) - \int \mathcal{G}\mathcal{L}_{\mathcal{D}}\left(\frac{\mu}{\pi(\mathcal{D})}-1\right)\pi(\mathcal{D})\mathrm{d}\theta \qquad \text{(From }\beta\text{-smoothness and (71))}$$

$$= d\beta\sigma^2(\beta+\lambda) + \int\left[\|\nabla\mathcal{L}_{\mathcal{D}}\|_2^2 - \sigma^2\Delta\mathcal{L}_{\mathcal{D}}\right]\left(\frac{\mu}{\pi(\mathcal{D})}-1\right)\pi(\mathcal{D})\mathrm{d}\theta$$

$$= d\beta\sigma^2(\beta+\lambda) + \int\|\nabla\mathcal{L}_{\mathcal{D}}\|_2^2\left(\mu-\pi(\mathcal{D})\right)\mathrm{d}\theta$$

$$\quad + \sigma^2\int\left\langle\nabla\mathcal{L}_{\mathcal{D}},\nabla\left[\left(\frac{\mu}{\pi(\mathcal{D})}-1\right)\pi(\mathcal{D})\right]\right\rangle\mathrm{d}\theta \qquad\qquad \text{(From (19))}$$

$$= d\beta\sigma^2(\beta+\lambda) + \int\|\nabla\mathcal{L}_{\mathcal{D}}\|_2^2\left(\mu-\pi(\mathcal{D})\right)\mathrm{d}\theta + \sigma^2\int\left\langle\nabla\mathcal{L}_{\mathcal{D}},-\frac{\nabla\mathcal{L}_{\mathcal{D}}}{\sigma^2}\right\rangle\left(\mu-\pi(\mathcal{D})\right)\mathrm{d}\theta$$

$$\quad + \sigma^2\int\left\langle\nabla\mathcal{L}_{\mathcal{D}},\nabla\frac{\mu}{\pi(\mathcal{D})}\right\rangle\pi(\mathcal{D})\mathrm{d}\theta \qquad\qquad \left(\text{Since }\nabla\pi(\mathcal{D})=-\frac{\nabla\mathcal{L}_{\mathcal{D}}}{\sigma^2}\pi(\mathcal{D})\right)$$

$$= d\beta\sigma^2(\beta+\lambda) + 0 + 2\sigma^2\int\left\langle\sqrt{\frac{\mu}{\pi(\mathcal{D})}}\nabla\mathcal{L}_{\mathcal{D}},\nabla\sqrt{\frac{\mu}{\pi(\mathcal{D})}}\right\rangle\pi(\mathcal{D})\mathrm{d}\theta$$

$$\leq d\beta\sigma^2(\beta+\lambda) + \frac{1}{2}\underset{\mu}{\mathbb{E}}\left[\|\nabla\mathcal{L}_{\mathcal{D}}\|_2^2\right] + 2\sigma^4\underset{\pi(\mathcal{D})}{\mathbb{E}}\left[\left\|\nabla\sqrt{\frac{\mu}{\pi(\mathcal{D})}}\right\|_2^2\right]$$

$$\qquad\qquad\qquad\qquad \text{(From (20) with } a=2\sigma^2)$$

$$\square$$

Another change in measure inequality needed for the proof is the Donkser-Vardhan variational principle.

**Lemma 37** (Donsker-Vardhan Variational principle (Donsker & Varadhan, 1983)). *If $\nu$ and $\nu'$ are two distributions on $\mathbb{R}^d$ such that $\nu \ll \nu'$, then for all functions $f : \mathbb{R}^d \to \mathbb{R}$,*

$$\underset{\theta\sim\nu}{\mathbb{E}}\left[f(\theta)\right] \leq \mathrm{KL}\left(\nu\|\nu'\right) + \log\underset{\theta'\sim\nu'}{\mathbb{E}}\left[\exp(f(\theta'))\right]. \tag{124}$$

We are now ready to prove the rate of convergence guarantee for Noisy-GD following Chewi et al. (2021)'s method, but with a more refined analysis that leads to a improvement of $\log q$ factor in the discretization error (compared to the original (Chewi et al., 2021, Theorem 4)).

**Theorem 38** (Convergence of Noisy-GD in Rényi divergence). *Let constants $\beta, \lambda, \sigma^2 \geq 0$ and $q, B > 1$. Suppose the loss function $\ell(\theta; \mathbf{x})$ is $(\sigma^2 \log(B)/4)$-bounded and $\beta$-smooth, and regularizer is $\mathbf{r}(\theta) = \frac{\lambda}{2}\|\theta\|_2^2$. If step size is $\eta \leq \frac{\lambda}{64Bq^2(\beta+\lambda)^2}$, then for any database $\mathcal{D} \in \mathcal{X}^n$ and any weight initialization distribution $\mu_0$ for $\Theta_0$, the Rényi divergence of distribution $\mu_{\eta K}$ of output model $\Theta_{\eta K} = \text{Noisy-GD}(\mathcal{D}, \Theta_0, K)$ with respect to the Gibbs distribution $\pi(\mathcal{D})$ defined in (117) shrinks as follows:*

$$\mathrm{R}_q\left(\mu_{\eta K}\|\pi(\mathcal{D})\right) \leq q\exp\left(-\frac{\lambda\eta K}{2B}\right)\mathrm{R}_q\left(\mu_0\|\pi(\mathcal{D})\right) + \frac{32d\eta qB(\beta+\lambda)^2}{\lambda}. \tag{125}$$

*Proof.* From $(\beta + \lambda)$-smoothness of loss $\mathcal{L}_{\mathcal{D}}$ we have that for any $\eta k \leq t \leq \eta(k+1)$,

$$\|\nabla\mathcal{L}_{\mathcal{D}}(\Theta_{\eta k}) - \nabla\mathcal{L}_{\mathcal{D}}(\Theta_t)\|_2^2 \leq (\beta+\lambda)^2\|\Theta_{\eta k} - \Theta_t\|_2^2$$

$$= (\beta+\lambda)^2\left\|(t-\eta k)\nabla\mathcal{L}_{\mathcal{D}}(\Theta_{\eta k}) - \sqrt{2(t-\eta k)\sigma^2}\mathbf{Z}_k\right\|_2^2$$

$$\text{(From (118))}$$

$$\leq 2\eta^2(\beta+\lambda)^2\|\nabla\mathcal{L}_{\mathcal{D}}(\Theta_{\eta k})\|_2^2 + 4\eta\sigma^2(\beta+\lambda)^2\|\mathbf{Z}_k\|_2^2$$

$$\leq 4\eta^2(\beta+\lambda)^2\|\nabla\mathcal{L}_{\mathcal{D}}(\Theta_{\eta k}) - \nabla\mathcal{L}_{\mathcal{D}}(\Theta_t)\|_2^2$$

$$\quad + 4\eta^2(\beta+\lambda)^2\|\nabla\mathcal{L}_{\mathcal{D}}(\Theta_t)\|_2^2 + 4\eta\sigma^2(\beta+\lambda)^2\|\mathbf{Z}_k\|_2^2$$

Let $\rho_t := \frac{\mu_t}{\pi(\mathcal{D})}$ and $\psi_t := \rho_t^{q-1}/\mathrm{E}_q\left(\rho_t\|\pi(\mathcal{D})\right)$. If $\eta \le \frac{1}{2\sqrt{2}(\beta+\lambda)}$, we rearrange to get the following and use it to get the following bound on the discretization error in (121):

$$\mathbb{E}\left[\psi_t(\Theta_t)\left\|\nabla\mathcal{L}_{\mathcal{B}_k}(\Theta_{\eta k}) - \nabla\mathcal{L}_{\mathcal{D}}(\Theta_t)\right\|_2^2\right] \le 8\eta^2(\beta+\lambda)^2 \underbrace{\mathbb{E}\left[\psi_t(\Theta_t)\left\|\nabla\mathcal{L}_{\mathcal{D}}(\Theta_t)\right\|_2^2\right]}_{\overset{\mathrm{def}}{=}F_1}$$

$$+ 32\eta\sigma^2(\beta+\lambda)^2 \underbrace{\mathbb{E}\left[\psi_t(\Theta_t)\left\|\mathbf{Z}_k\right\|_2^2/4\right]}_{\overset{\mathrm{def}}{=}F_2}.$$

Hence, for solving the PDI (121), we have to bound the three expectations $F_1$ and $F_2$.

1. **Bounding $F_1$.** Note that $\underset{\Theta_t \sim \mu_t}{\mathbb{E}}\left[\psi_t(\Theta_t)\right] = \int \psi_t(\theta)\mu_t(\theta)\mathrm{d}\theta = \frac{1}{\mathrm{E}_q(\rho_t\|\pi(\mathcal{D}))}\int \frac{\mu_t^q}{\pi(\mathcal{D})^{q-1}}\mathrm{d}\theta = 1$.
   So, $\psi_t\mu_t(\theta) := \psi_t(\theta)\mu_t(\theta)$ is a probablity density function on $\mathbb{R}^d$. On applying the measure change Lemma 36 on it, we get

   $$F_1 = \underset{\psi_t\mu_t}{\mathbb{E}}\left[\|\nabla\mathcal{L}_{\mathcal{D}}\|_2^2\right] \le 4\sigma^4 \underset{\pi(\mathcal{D})}{\mathbb{E}}\left[\left\|\nabla\sqrt{\frac{\psi_t\mu_t}{\pi(\mathcal{D})}}\right\|_2^2\right] + 2d\sigma^2(\beta+\lambda) \qquad \text{(From (122))}$$

   $$= 4\sigma^4 \underset{\pi(\mathcal{D})}{\mathbb{E}}\left[\frac{\left\|\nabla\sqrt{\rho_t^q}\right\|_2^2}{\mathrm{E}_q\left(\mu_t\|\pi(\mathcal{D})\right)}\right] + 2d\sigma^2(\beta+\lambda)$$

   $$= \sigma^4 q^2 \frac{\mathrm{I}_q\left(\mu_t\|\pi(\mathcal{D})\right)}{\mathrm{E}_q\left(\mu_t\|\pi(\mathcal{D})\right)} + 2d\sigma^2(\beta+\lambda). \qquad \text{(From (40))}$$

2. **Bounding $F_2$.** Since $\psi_t\mu_t$ is a valid density on $\mathbb{R}^d$, the joint density $\psi_t\mu_{t,z}(\theta,z) := \psi_t(\theta)\mu_{t,z}(\theta,z)$ where $\mu_{t,z}$ is the joint density of $\Theta_t$ and $\mathbf{Z}_k$ is also a valid density. Note that the $F_2$ is an expectation on $\|\mathbf{Z}_k\|_2^2$ taken over the joint density $\psi_t\mu_{t,z}$. We can perform a measure change operation using Donsker-Vardhan principle to get

   $$F_2 = \underset{\psi_t\mu_{t,z}}{\mathbb{E}}\left[\|\mathbf{Z}_k\|_2^2/4\right] \le \mathrm{KL}\left(\psi_t\mu_{t,z}\|\mu_{t,z}\right) + \log\underset{\mu_z}{\mathbb{E}}\left[\exp(\|\mathbf{Z}_k\|_2^2/4)\right],$$

   where we simplified the second term using the fact that the marginal $\mu_z$ of $\mu_{t,z}$ is a standard normal Gaussian. The random variable $\|\mathbf{Z}_k\|_2^2$ is distributed according to the Chi-squared distribution $\chi_d^2$ with $d$ degrees of freedom. Since the moment generating function of Chi-squared distribution is $\mathrm{M}_{\chi_d^2}(t) = \underset{X\sim\chi_d^2}{\mathbb{E}}\left[\exp(tX)\right] = (1-2t)^{-d/2}$ for $t < \frac{1}{2}$, we can simplify the second term in $F_2$ as

   $$\log\underset{\mu_z}{\mathbb{E}}\left[\exp(\|\mathbf{Z}_k\|_2^2/4)\right] = \log\mathrm{M}_{\chi_d^2}\left(\frac{1}{4}\right) = \frac{d\log 2}{2}. \qquad (126)$$

   The KL divergence term can be simplified as follows.

   $$\mathrm{KL}\left(\psi_t\mu_{t,z}\|\mu_{t,z}\right) = \int\int \psi_t\mu_{t,z}(\theta_t,z)\log\psi_t(\theta_t)\mathrm{d}\theta_t\mathrm{d}z$$

   $$= \int \psi_t\mu_t \log\frac{\rho_t^{q-1}}{\mathrm{E}_q\left(\mu_t\|\pi(\mathcal{D})\right)}\mathrm{d}\theta_t \qquad \text{(On marginalization of $z$)}$$

   $$= \frac{q-1}{q}\int \mu_t\psi_t \log\left\{\frac{\rho_t^q}{\mathrm{E}_q\left(\mu_t\|\pi(\mathcal{D})\right)} - \log\mathrm{E}_q\left(\mu_t\|\pi(\mathcal{D})\right)^{1/(q-1)}\right\}\mathrm{d}\theta_t$$

   $$= \frac{q-1}{q}\left\{\mathrm{KL}\left(\mu_t\psi_t\|\pi(\mathcal{D})\right) - \mathrm{R}_q\left(\mu_t\|\pi(\mathcal{D})\right)\right\}$$

   $$\le \mathrm{KL}\left(\mu_t\psi_t\|\pi(\mathcal{D})\right) \qquad \text{(Since $\mathrm{R}_q\left(\mu_t\|\pi(\mathcal{D})\right) > 0$)}$$

Note that under the assumptions of the Theorem, $\pi(\mathcal{D})$ satisfies log-Sobolev inequality (74) with constant $\lambda/B$ (i.e. satisfies $\mathrm{LS}(\lambda/B)$). To see this, recall from Lemma 18 that the Gaussian distribution $\rho(\theta) = \mathcal{N}\left(0, \frac{\sigma^2}{\lambda}\mathbb{I}_d\right)$ satisfies $\mathrm{LS}(\lambda)$ inequality. Since loss $\ell(\theta; \mathbf{x})$ is $(\sigma^2 \log(B)/4)$-bounded, the density ratio $\frac{\pi(D)(\theta)}{\rho(\theta)} \in \left[\frac{1}{\sqrt{B}}, \sqrt{B}\right]$. The claim therefore follows from Lemma 20. Using this inequality, from Lemma 21 we have

$$
\begin{aligned}
\mathrm{KL}\left(\mu_t \psi_t \| \pi(\mathcal{D})\right) &\leq \frac{\sigma^2 B}{2\lambda} \int \mu_t \psi_t \left\| \nabla \log\left(\frac{\mu_t \psi_t}{\pi(\mathcal{D})}\right) \right\|_2^2 \mathrm{d}\theta_t \\
&= \frac{\sigma^2 B}{2\lambda} \int \frac{\rho_t^q}{\mathrm{E}_q\left(\mu_t\|\pi(\mathcal{D})\right)} \left\| \nabla \log(\rho_t^q) \right\|_2^2 \pi(\mathcal{D})\mathrm{d}\theta_t \\
&= \frac{2\sigma^2 B}{\lambda} \frac{1}{\mathrm{E}_q\left(\mu_t\|\pi(\mathcal{D})\right)} \int \left\| \nabla(\rho_t^{q/2}) \right\|_2^2 \pi(\mathcal{D})\mathrm{d}\theta_t \\
&= \frac{q^2 \sigma^2 B}{2\lambda} \frac{\mathrm{I}_q\left(\mu_t\|\pi(\mathcal{D})\right)}{\mathrm{E}_q\left(\mu_t\|\pi(\mathcal{D})\right)}
\end{aligned}
$$

On combining all the two bounds on $F_1$ and $F_2$ and rearranging, we have

$$
\begin{aligned}
\mathbb{E}\left[\psi_t(\Theta_t) \|\nabla \mathcal{L}_{\mathcal{D}}(\Theta_{\eta k}) - \nabla \mathcal{L}_{\mathcal{D}}(\Theta_t)\|_2^2\right] &\leq 8\eta q^2 \sigma^4 (\beta + \lambda)^2 \frac{\mathrm{I}_q\left(\mu_t\|\pi(\mathcal{D})\right)}{\mathrm{E}_q\left(\mu_t\|\pi(\mathcal{D})\right)} \left(\eta + \frac{2B}{\lambda}\right) \\
&\quad + 16\eta d\sigma^2 (\beta + \lambda)^2 \left(\eta(\beta + \lambda) + \log 2\right)
\end{aligned}
$$

Let step size be $\eta \leq \min\left\{\frac{2B}{\lambda}, \frac{\lambda}{64 B q^2 (\beta+\lambda)^2}\right\}$. Then, the first term above is bounded as

$$
8\eta q^2 \sigma^4 (\beta + \lambda)^2 \frac{\mathrm{I}_q\left(\mu_t\|\pi(\mathcal{D})\right)}{\mathrm{E}_q\left(\mu_t\|\pi(\mathcal{D})\right)} \left(\eta + \frac{2B}{\lambda}\right) \leq \frac{\sigma^4}{2} \frac{\mathrm{I}_q\left(\mu_t\|\pi(\mathcal{D})\right)}{\mathrm{E}_q\left(\mu_t\|\pi(\mathcal{D})\right)}. \tag{127}
$$

Let $\eta \leq \frac{1}{4(\beta+\lambda)}$. Then, in the third term, $(\eta(\beta + \lambda) + \log 2) \leq 1$. Plugging the bound on discretization error back in the PDI (121), we get

$$
\partial_t \mathrm{R}_q\left(\mu_t\|\pi(\mathcal{D})\right) \leq -\frac{q\sigma^2}{4} \frac{\mathrm{I}_q\left(\mu_t\|\pi(\mathcal{D})\right)}{\mathrm{E}_q\left(\mu_t\|\pi(\mathcal{D})\right)} + 16\eta dq(\beta + \lambda)^2. \tag{128}
$$

Since $\pi(\mathcal{D})$ satisfies $\mathrm{LS}(\lambda/B)$ inequality, from Lemma 23 this PDI reduces to

$$
\partial_t \mathrm{R}_q\left(\mu_t\|\pi(\mathcal{D})\right) + \frac{\lambda}{2B}\left(\frac{\mathrm{R}_q\left(\mu_t\|\pi(\mathcal{D})\right)}{q} + (q-1)\partial_q \mathrm{R}_q\left(\mu_t\|\pi(\mathcal{D})\right)\right) \leq 16 d\eta q(\beta + \lambda)^2. \tag{129}
$$

Let $c_1 = \frac{\lambda}{2B}$ and $c_2 = 16 d\eta(\beta + \lambda)^2$. Additionally, let $u(q, t) = \frac{\mathrm{R}_q(\mu_t\|\pi(\mathcal{D}))}{q}$. Then,

$$
\partial_t \mathrm{R}_q\left(\mu_t\|\pi(\mathcal{D})\right) + c_1\left(\frac{\mathrm{R}_q\left(\mu_t\|\pi(\mathcal{D})\right)}{q} + (q-1)\partial_q \mathrm{R}_q\left(\mu_t\|\pi(\mathcal{D})\right)\right) \leq c_2 q
$$

$$
\implies \frac{\partial_t \mathrm{R}_q\left(\mu_t\|\pi(\mathcal{D})\right)}{q} + c_1 \frac{\mathrm{R}_q\left(\mu_t\|\pi(\mathcal{D})\right)}{q} + c_1(q-1)\left(\frac{\partial_q \mathrm{R}_q\left(\mu_t\|\pi(\mathcal{D})\right)}{q} - \frac{\mathrm{R}_q\left(\mu_t\|\pi(\mathcal{D})\right)}{q^2}\right) \leq c_2
$$

$$
\implies \partial_t u(q, t) + c_1 u(q, t) + c_1(q - 1)\partial_q u(q, t) \leq c_2.
$$

For some constant $\bar{q} \geq 1$, let $q(s) = (\bar{q} - 1)\exp(c_1(s - \eta K)) + 1$, and $t(s) = s$. Note that $\frac{\mathrm{d}q(s)}{\mathrm{d}s} = c_1(q(s) - 1)$ and $\frac{\mathrm{d}t(s)}{\mathrm{d}s} = 1$. Therefore, for any $0 \leq t \leq \eta K$, the PDI above implies the following differential inequality is followed along the path $u(s) = u(q(s), t(s))$.

$$
\begin{aligned}
\frac{\mathrm{d}u(s)}{\mathrm{d}s} + c_1 u(s) \leq c_2 &\implies \frac{\mathrm{d}}{\mathrm{d}s}\{e^{c_1 s} u(s)\} \leq c_2 e^{c_1 s} \\
&\implies [e^{c_1 s} u(s)]_0^{\eta K} \leq \int_0^{\eta K} c_2 e^{c_1 s}\mathrm{d}s \\
&\implies e^{c_1 \eta K} u(\eta K) - u(0) \leq \frac{c_2(e^{c_1 \eta K} - 1)}{c_1} \\
&\implies u(\eta K) \leq e^{-c_1 \eta K} u(0) + \frac{c_2}{c_1}(1 - e^{-c_1 \eta K}).
\end{aligned}
$$

On reversing the parameterization of $q$ and $t$, we get

$$\mathrm{R}_{q(\eta K)}\left(\mu_{\eta K}\|\pi(\mathcal{D})\right) \leq \frac{q(\eta K)}{q(0)}e^{-c_1\eta K}\mathrm{R}_{q(0)}\left(\mu_0\|\pi(\mathcal{D})\right) + \frac{c_2}{c_1}q(\eta K)$$

$$\leq \frac{q(\eta K)}{q(0)}\exp\left(-\frac{\lambda\eta K}{2B}\right)\mathrm{R}_{q(0)}\left(\mu_0\|\pi(\mathcal{D})\right) + \frac{32d\eta B(\beta+\lambda)^2}{\lambda}q(\eta K).$$

Since $q(0) > 1$ and $\bar{q} = q(\eta K) > q(0)$, from monotonicity of Rényi divergence in $q$, we get

$$\mathrm{R}_{\bar{q}}\left(\mu_{\eta K}\|\pi(\mathcal{D})\right) \leq \bar{q}\exp\left(-\frac{\lambda\eta K}{2B}\right)\mathrm{R}_{\bar{q}}\left(\mu_0\|\pi(\mathcal{D})\right) + \frac{32d\eta\bar{q}B(\beta+\lambda)^2}{\lambda}. \tag{130}$$

Finally, noting that for constants $B, q > 1$ and $\beta, \lambda \geq 0$, step size $\eta \leq \min\{\frac{1}{2\sqrt{2}(\beta+\lambda)}, \frac{1}{4(\beta+\lambda)}, \frac{2B}{\lambda}, \frac{\lambda}{64Bq^2(\beta+\lambda)^2}\} = \frac{\lambda}{64Bq^2(\beta+\lambda)^2}$ completes the proof. $\qquad\square$

We will use Theorem 38 for proving the data-deletion and utility gaurantee on $(\mathrm{A}_{\text{Noisy-GD}}, \bar{\mathrm{A}}_{\text{Noisy-GD}})$. We need the following result that shows that Gibbs distributions enjoy strong indistinguishability on bounded perturbations to its potential function (which is basically why the exponential mechanism satisfies $(\varepsilon, 0)$-DP (Wang et al., 2015; Dwork et al., 2014)).

**Lemma 39** (Indistinguishability under bounded perturbations)**.** *For two potential functions $\mathcal{L}, \mathcal{L}' : \mathbb{R}^d \to \mathbb{R}$ and some constant $\sigma^2$, let $\nu \propto e^{-\mathcal{L}/\sigma^2}$ and $\nu' \propto e^{-\mathcal{L}'/\sigma^2}$ be the respective Gibbs distributions. If $|\mathcal{L}(\theta) - \mathcal{L}'(\theta)| \leq c$ for all $\theta \in \mathbb{R}^d$, then $\mathrm{R}_q\left(\nu\|\nu'\right) \leq \frac{2c}{\sigma^2}$ for all $q > 1$.*

*Proof.* The Gibbs distributions $\nu, \nu'$ have a density

$$\nu(\theta) = \frac{1}{\Lambda}e^{-\mathcal{L}(\theta)/\sigma^2}, \quad \text{and} \quad \nu'(\theta) = \frac{1}{\Lambda'}e^{-\mathcal{L}'(\theta)/\sigma^2},$$

where $\Lambda, \Lambda'$ are the respective normalization constants. If for all $\theta \in \mathbb{R}^d$, the potential difference $|\mathcal{L}(\theta) - \mathcal{L}'(\theta)| \leq c$, then

$$\mathrm{R}_q\left(\nu\|\nu'\right) = \frac{1}{q-1}\log\int\frac{\nu^q}{\nu'^{q-1}}\mathrm{d}\theta$$

$$= \frac{1}{q-1}\log\int\left(\frac{\Lambda'}{\Lambda}\right)^{q-1}\exp\left(\frac{q-1}{\sigma^2}(\mathcal{L}'(\theta) - \mathcal{L}(\theta))\right)\times\nu(\theta)\mathrm{d}\theta$$

$$\leq \frac{1}{q-1}\left\{(q-1)\log\frac{\Lambda'}{\Lambda} + \log\exp\left(\frac{c(q-1)}{\sigma^2}\int\nu\mathrm{d}\theta\right)\right\}$$

$$= \frac{1}{q-1}\left\{(q-1)\log\frac{\int\exp\left(-\frac{\mathcal{L}(\theta)}{\sigma^2} + \frac{\mathcal{L}(\theta)-\mathcal{L}'(\theta)}{\sigma^2}\right)\mathrm{d}\theta}{\int\exp\left(-\frac{\mathcal{L}(\theta)}{\sigma^2}\right)\mathrm{d}\theta} + \frac{c(q-1)}{\sigma^2}\right\}$$

$$\leq \frac{2c}{\sigma^2}.$$

$\qquad\square$

In Theorem 7, we show that $(\mathrm{A}_{\text{Noisy-GD}}, \bar{\mathrm{A}}_{\text{Noisy-GD}})$ solves the data-deletion problem described in Section 4 even for non-convex losses. Our proof uses the convergence Theorem 38 and indistinguishability for bounded perturbation Lemma 39 to show that the data-deletion algorithm $\bar{\mathrm{A}}_{\text{Noisy-GD}}$ can consistently produce models indistinguishable to the corresponding Gibbs distribution (117) in the online setting at a fraction of computation cost of $\mathrm{A}_{\text{Noisy-GD}}$. As discussed in Remark 3, such an indistinguishability is sufficient for ensuring data-deletion for non-adaptive requests. As for adaptive requests, the well-known RDP guarantee of Abadi et al. (2016) combined with our reduction Theorem 5 offers a data-deletion guarantee for $(\mathrm{A}_{\text{Noisy-GD}}, \bar{\mathrm{A}}_{\text{Noisy-GD}})$ under adaptivity.

Our proof of accuracy for the data-deleted models leverages the fact that Gibbs distribution (117) is an almost excess risk minimizer as shown in the following Theorem 40. Since our data-deletion guarantee is based on near-indistinguishability to (117), this property also ensures near-optimal excess risk of data-deleted models.

**Theorem 40** (Near optimality of Gibbs sampling). *If the loss function $\ell(\theta; \mathbf{x})$ is $\sigma^2 \log(B)/4$-bounded and $\beta$-smooth, the regularizer is $\mathbf{r}(\theta) = \frac{\lambda}{2} \|\theta\|_2^2$, then the excess empirical risk for a model $\bar{\Theta}$ sampled from the Gibbs distribution $\pi(\mathcal{D}) \propto e^{-\mathcal{L}_\mathcal{D}/\sigma^2}$ is*

$$\text{err}(\bar{\Theta}; \mathcal{D}) = \mathbb{E}\left[\mathcal{L}_\mathcal{D}(\bar{\Theta}) - \mathcal{L}_\mathcal{D}(\theta_\mathcal{D}^*)\right] \le \frac{d\sigma^2}{2}\left(\log \frac{\beta + \lambda}{\lambda} + \sqrt{B}\right). \tag{131}$$

*Proof.* We simplify expected loss as

$$\mathbb{E}\left[\mathcal{L}_\mathcal{D}(\bar{\Theta})\right] = \int \mathcal{L}_\mathcal{D}\pi(\mathcal{D})\mathrm{d}\theta = \sigma^2(\mathrm{H}(\pi(\mathcal{D})) - \log(\Lambda_\mathcal{D})), \tag{132}$$

where

$$\mathrm{H}(\pi(\mathcal{D})) = -\int \pi(\mathcal{D})\log \pi(\mathcal{D})\mathrm{d}\theta = -\int \frac{e^{-\mathcal{L}_\mathcal{D}/\sigma^2}}{\Lambda_\mathcal{D}}\log \frac{e^{-\mathcal{L}_\mathcal{D}/\sigma^2}}{\Lambda_\mathcal{D}}\mathrm{d}\theta \tag{133}$$

is the differential entropy of $\pi(\mathcal{D})$, and $\Lambda_\mathcal{D} = \int e^{-\mathcal{L}_\mathcal{D}/\sigma^2}\mathrm{d}\theta$ is the normalization constant. Since the potential function $\mathcal{L}_\mathcal{D}$ is $(\lambda + \beta)$-smooth, we have

$$-\sigma^2 \log(\Lambda_\mathcal{D}) = -\sigma^2 \log \int e^{-\mathcal{L}_\mathcal{D}/\sigma^2}\mathrm{d}\theta$$

$$= \mathcal{L}_\mathcal{D}(\theta_\mathcal{D}^*) - \sigma^2 \log \int e^{(\mathcal{L}_\mathcal{D}(\theta_\mathcal{D}^*) - \mathcal{L}_\mathcal{D}(\theta))/\sigma^2}\mathrm{d}\theta$$

$$\le \mathcal{L}_\mathcal{D}(\theta_\mathcal{D}^*) - \sigma^2 \log \int e^{-(\beta+\lambda)\|\theta - \theta_\mathcal{D}^*\|_2^2/2\sigma^2}\mathrm{d}\theta$$

$$= \mathcal{L}_\mathcal{D}(\theta_\mathcal{D}^*) - \frac{d\sigma^2}{2}\log\left(\frac{2\pi\sigma^2}{\lambda + \beta}\right).$$

Since $\ell(\theta; \mathbf{x})$ is $\sigma^2 \log(B)/4$-bounded, note that for the Gaussian distribution $\rho \sim \mathcal{N}\left(0, \frac{\sigma^2}{\lambda}\mathbb{I}_d\right)$, the density ratio lies in $\frac{\pi(\mathcal{D})(\theta)}{\rho(\theta)} \in \left[\frac{1}{\sqrt{B}}, \sqrt{B}\right]$ for all $\theta \in \mathbb{R}^d$. We decompose entropy $\mathrm{H}(\pi(\mathcal{D}))$ into cross-entropy and KL divergence to get

$$\mathrm{H}(\pi(\mathcal{D})) = -\int \pi(\mathcal{D})\log \rho\,\mathrm{d}\theta - \mathrm{KL}\left(\pi(\mathcal{D})\|\rho\right)$$

$$\le -\int \pi(\mathcal{D})\log\left[\left(\frac{\lambda}{2\pi\sigma^2}\right)^{d/2}e^{-\frac{\lambda\|\theta\|_2^2}{2\sigma^2}}\right]\mathrm{d}\theta \qquad \text{(Since KL}\left(\pi(\mathcal{D})\|\rho\right) \ge 0\text{)}$$

$$= \frac{d}{2}\log \frac{2\pi\sigma^2}{\lambda} + \frac{\lambda}{2\sigma^2}\int \|\theta\|_2^2\pi(\mathcal{D})(\theta)\mathrm{d}\theta$$

$$\le \frac{d}{2}\log \frac{2\pi\sigma^2}{\lambda} + \frac{\lambda\sqrt{B}}{2\sigma^2}\int \|\theta\|_2^2\rho(\theta)\mathrm{d}\theta \qquad \text{(Since } \frac{\pi(\mathcal{D})(\theta)}{\rho(\theta)} \in \left[\frac{1}{\sqrt{B}}, \sqrt{B}\right]\text{)}$$

$$= \frac{d}{2}\log \frac{2\pi\sigma^2}{\lambda} + \frac{d\sqrt{B}}{2}.$$

On combining the bounds, we get

$$\text{err}(\bar{\Theta}; \mathcal{D}) = \mathbb{E}\left[\mathcal{L}_\mathcal{D}(\bar{\Theta}) - \mathcal{L}_\mathcal{D}(\theta_\mathcal{D}^*)\right] \le \frac{d\sigma^2}{2}\left(\log \frac{\beta + \lambda}{\lambda} + \sqrt{B}\right). \tag{134}$$

$$\square$$

**Theorem 7** (Accuracy, privacy, deletion, and computation tradeoffs). *Let constants $\lambda, \beta, L, \sigma^2, \eta > 0$, $q, B > 1$, and $0 < \varepsilon_{\mathrm{dd}} \le \varepsilon_{\mathrm{dp}} < d$. Let the loss function $\ell(\theta; \mathbf{x})$ be $\frac{\sigma^2 \log(B)}{4}$-bounded, $L$-Lipschitz and $\beta$-smooth, the regularizer be $\mathbf{r}(\theta) = \frac{\lambda}{2}\|\theta\|_2^2$, and the weight initialization distribution be $\rho = \mathcal{N}\left(0, \frac{\sigma^2}{\lambda}\mathbb{I}_d\right)$. Then,*

**(1.)** *both* $A_{Noisy\text{-}GD}$ *and* $\bar{A}_{Noisy\text{-}GD}$ *are* $(q, \varepsilon_{\mathrm{dp}})$-*RDP for any* $\eta \geq 0$ *and any* $K_A, K_{\bar{A}} \geq 0$ *if*

$$\sigma^2 \geq \frac{qL^2}{\varepsilon_{\mathrm{dp}} n^2} \cdot \eta \max\{K_A, K_{\bar{A}}\}, \tag{135}$$

**(2.)** *pair* $(A_{Noisy\text{-}GD}, \bar{A}_{Noisy\text{-}GD})$ *satisfy* $(q, \varepsilon_{\mathrm{dd}})$-*data-deletion under all non-adaptive* $r$-*requesters for any* $\sigma^2 > 0$, *if learning rate is* $\eta \leq \frac{\lambda \varepsilon_{\mathrm{dd}}}{64 d q B (\beta + \lambda)^2}$ *and number of iterations satisfy*

$$K_A \geq \frac{2B}{\lambda \eta} \log\left(\frac{q \log(B)}{\varepsilon_{\mathrm{dd}}}\right), \quad K_{\bar{A}} \geq K_A - \frac{2B}{\lambda \eta} \log\left(\frac{\log(B)}{2\left(\varepsilon_{\mathrm{dd}} + \frac{r}{n}\log(B)\right)}\right), \quad (136)$$

**(3.)** *and all models in sequence* $(\hat{\Theta}_i)_{i \geq 0}$ *output by* $(A_{Noisy\text{-}GD}, \bar{A}_{Noisy\text{-}GD}, \mathcal{Q})$ *on any* $\mathcal{D}_0 \in \mathcal{X}^n$, *where* $\mathcal{Q}$ *is an* $r$-*requester, satisfy* $\mathrm{err}(\hat{\Theta}_i; \mathcal{D}_i) = \tilde{O}\left(\frac{dq}{\varepsilon_{\mathrm{dp}} n^2} + \frac{1}{n}\sqrt{\frac{q \varepsilon_{\mathrm{dd}}}{\varepsilon_{\mathrm{dp}}}}\right)$ *when inequalities in* (136) *and* (135) *are equalities.*

*Proof.* **(1.) Privacy.** By Theorem 24, Noisy-GD with $K$ iterations on an $L$-Lipschitz loss function satisfies $(q, \varepsilon_{\mathrm{dp}})$-RDP for any initial weight distribution $\rho$ and learning rate $\eta \geq 0$ if $\sigma^2 = \frac{qL^2}{\varepsilon_{\mathrm{dp}} n^2} \cdot \eta K$. Since, both $A_{\text{Noisy-GD}}$ and $\bar{A}_{\text{Noisy-GD}}$ run Noisy-GD for $K_A$ and $K_{\bar{A}}$ iterations respectively, setting the noise variance given in the Theorem statement ensures $(q, \varepsilon_{\mathrm{dp}})$-RDP for both.

**(2.) Deletion.** For showing data-deletion under non-adaptive requests, recall that it is sufficient to show that there exists a map $\pi : \mathcal{X}^n \to \mathcal{O}$ such that for all $i \geq 1$,

$$\mathrm{R}_q\left(\bar{A}(\mathcal{D}_{i-1}, u_i, \hat{\Theta}_{i-1}) \middle\| \pi(\mathcal{D}_i)\right) \leq \varepsilon_{\mathrm{dd}}, \tag{137}$$

for all edit sequences $(u_i)_{i \geq 1}$ from $\mathcal{U}^r$, where $(\hat{\Theta}_i)_{i \geq 0}$ is the sequence of models generated by the interaction of $(A_{\text{Noisy-GD}}, \bar{A}_{\text{Noisy-GD}}, \mathcal{Q})$ on any database $\mathcal{D}_0 \in \mathcal{X}^n$. For all $i \geq 0$, let $\hat{\mu}_i$ denote the distribution of $\hat{\Theta}_i$. We prove (137) via induction.

*Base step:* Note that the initial weight distribution $\rho = \mathcal{N}\left(0, \frac{\sigma^2}{\lambda}\mathbb{I}_d\right)$ has a density proportional to $e^{-\mathbf{r}(\theta)/\sigma^2}$ and the distribution $\pi(\mathcal{D}_0)$ has a density proportional to $e^{-\mathcal{L}_{\mathcal{D}_0}(\theta)/\sigma^2}$. Since both of these are Gibbs distributions with their potential difference $|\mathcal{L}_{\mathcal{D}_0}(\theta) - \mathbf{r}(\theta)| \leq \sigma^2 \log(B)/4$ for all $\theta \in \mathbb{R}^d$ due to boundedness assumption on $\ell(\theta; \mathbf{x})$, we have from Lemma 39 that

$$\mathrm{R}_q\left(\rho \| \pi(\mathcal{D}_0)\right) \leq \frac{2}{\sigma^2} \times \frac{\sigma^2 \log(B)}{4} = \frac{\log(B)}{2}. \tag{138}$$

Under the stated assumptions on loss $\ell(\theta; \mathbf{x})$ and learning rate $\eta$, note that the convergence Theorem 38 holds. Since $\hat{\Theta}_0 = A_{\text{Noisy-GD}}(\mathcal{D}_0) = \text{Noisy-GD}(\mathcal{D}_0, \Theta_0, K_A)$, where $\Theta_0 \sim \rho$, we have

$$\begin{aligned}
\mathrm{R}_q\left(\hat{\mu}_0 \| \pi(\mathcal{D}_0)\right) &\leq q \exp\left(-\frac{\lambda \eta K_A}{2B}\right) \mathrm{R}_q\left(\rho \| \pi(\mathcal{D}_0)\right) + \frac{32 d \eta q B (\beta + \lambda)^2}{\lambda} \\
&\leq q \exp\left(-\frac{\lambda \eta K_A}{2B}\right)\left(\frac{\log(B)}{2}\right) + \frac{\varepsilon_{\mathrm{dd}}}{2} \qquad \text{(Since } \eta \leq \frac{\lambda \varepsilon_{\mathrm{dd}}}{64 d q B (\beta + \lambda)^2}\text{)} \\
&\leq \varepsilon_{\mathrm{dd}} \qquad\qquad\qquad\qquad \text{(Since } K_A \geq \frac{2B}{\lambda \eta} \log\left(\frac{q \log(B)}{\varepsilon_{\mathrm{dd}}}\right)\text{)}
\end{aligned}$$

*Induction step:* Suppose $\mathrm{R}_q\left(\hat{\mu}_{i-1} \| \pi(\mathcal{D}_{i-1})\right) \leq \varepsilon_{\mathrm{dd}}$. Again, from boundedness of $\ell(\theta; \mathbf{x})$, we have $|\mathcal{L}_{\mathcal{D}_{i-1}}(\theta) - \mathcal{L}_{\mathcal{D}_i}(\theta)| \leq \frac{r \sigma^2 \log B}{2n}$ for all $\theta \in \mathbb{R}^d$. Therefore, from Lemma 39 we have for all $q > 1$ that

$$\mathrm{R}_q\left(\pi(\mathcal{D}_{i-1}) \| \pi(\mathcal{D}_i)\right) \leq \frac{r \log(B)}{n}. \tag{139}$$

So from the weak triangle inequality Theorem 17 of Rényi divergence,

$$\mathrm{R}_q\left(\hat{\mu}_{i-1} \| \pi(\mathcal{D}_i)\right) \leq \mathrm{R}_q\left(\hat{\mu}_{i-1} \| \pi(\mathcal{D}_{i-1})\right) + \mathrm{R}_\infty\left(\pi(\mathcal{D}_{i-1}) \| \pi(\mathcal{D}_i)\right) \leq \varepsilon_{\mathrm{dd}} + \frac{r \log(B)}{n}. \tag{140}$$

Note that $K_{\bar{\mathrm{A}}} \geq K_{\mathrm{A}} - \frac{2B}{\lambda\eta} \log\left(\frac{\log(B)}{2\left(\varepsilon_{\mathrm{dd}} + \frac{r}{n}\log(B)\right)}\right) \geq \frac{2B}{\lambda\eta} \log\left(\frac{2q\left(\varepsilon_{\mathrm{dd}} + \frac{r}{n}\log(B)\right)}{\varepsilon_{\mathrm{dd}}}\right)$. Since $\hat{\Theta}_i = \bar{\mathrm{A}}_{\mathrm{Noisy\text{-}GD}}(\mathcal{D}_{i-1}, u_i, \hat{\Theta}_{i-1}) = \mathrm{Noisy\text{-}GD}(\mathcal{D}_i, \hat{\Theta}_{i-1}, K_{\bar{\mathrm{A}}})$, from convergence Theorem 38 we have

$$
\begin{aligned}
\mathrm{R}_q\left(\hat{\mu}_i \| \pi(\mathcal{D}_i)\right) &\leq q\exp\left(-\frac{\lambda\eta K_{\bar{\mathrm{A}}}}{2B}\right)\mathrm{R}_q\left(\hat{\mu}_{i-1}\|\pi(\mathcal{D}_i)\right) + \frac{32 d\eta q B(\beta+\lambda)^2}{\lambda} \\
&\leq q\exp\left(-\frac{\lambda\eta K_{\bar{\mathrm{A}}}}{2B}\right)\left(\varepsilon_{\mathrm{dd}} + \frac{r\log(B)}{n}\right) + \frac{\varepsilon_{\mathrm{dd}}}{2} \\
&\qquad\qquad \text{(From (140) and constraint } \eta \leq \frac{\lambda\varepsilon_{\mathrm{dd}}}{64 dqB(\beta+\lambda)^2}) \\
&\leq \varepsilon_{\mathrm{dd}}. \qquad\qquad\qquad \text{(Since } K_{\bar{\mathrm{A}}} \geq \frac{2B}{\lambda\eta}\log\left(\frac{2q\left(\varepsilon_{\mathrm{dd}} + \frac{r}{n}\log(B)\right)}{\varepsilon_{\mathrm{dd}}}\right))
\end{aligned}
$$

Hence, by induction, $\mathrm{R}_q\left(\hat{\mu}_i\|\pi(\mathcal{D}_i)\right) \leq \varepsilon_{\mathrm{dd}}$ holds for all $i \geq 0$.

**(3.) Accuracy.** Let $\theta^*_{\mathcal{D}_i} = \underset{\theta\in\mathbb{R}^d}{\arg\min}\, \mathcal{L}_{\mathcal{D}_i}(\theta)$, and $\bar{\Theta}_i \sim \pi(\mathcal{D}_i)$. We decompose the excess empirical risk of Noisy-GD as follows:

$$
\mathrm{err}(\hat{\Theta}_i; \mathcal{D}_i) = \mathbb{E}\left[\mathcal{L}_{\mathcal{D}_i}(\hat{\Theta}_i) - \mathcal{L}_{\mathcal{D}_i}(\bar{\Theta}_i)\right] + \mathbb{E}\left[\mathcal{L}_{\mathcal{D}_i}(\bar{\Theta}_i) - \mathcal{L}_{\mathcal{D}_i}(\theta^*_{\mathcal{D}_i})\right]. \tag{141}
$$

The second term is the suboptimality of Gibbs distribution and by Theorem 40, it is bounded as

$$
\mathbb{E}\left[\mathcal{L}_{\mathcal{D}_i}(\bar{\Theta}_i) - \mathcal{L}_{\mathcal{D}_i}(\theta^*_{\mathcal{D}_i})\right] \leq \frac{d\sigma^2}{2}\left(\log\frac{\beta+\lambda}{\lambda} + \sqrt{B}\right). \tag{142}
$$

Due to $L$-Lipschitzness and $(\lambda+\beta)$-smoothness of $\mathcal{L}_{\mathcal{D}_i}$, for any coupling $\Pi$ of $\hat{\Theta}_i$ and $\bar{\Theta}_i$, the first term is bounded as

$$
\begin{aligned}
\mathbb{E}\left[\mathcal{L}_{\mathcal{D}_i}(\hat{\Theta}_i) - \mathcal{L}_{\mathcal{D}_i}(\bar{\Theta}_i)\right] &\leq \underset{\Pi}{\mathbb{E}}\left[\left\langle \nabla\mathcal{L}_{\mathcal{D}_i}(\bar{\Theta}_i), \hat{\Theta}_i - \bar{\Theta}_i\right\rangle + \frac{\lambda+\beta}{2}\left\|\hat{\Theta}_i - \Theta_i\right\|_2^2\right] \\
&\leq L\sqrt{\underset{\Pi}{\mathbb{E}}\left[\left\|\hat{\Theta}_i - \bar{\Theta}_i\right\|_2^2\right]} + \frac{\lambda+\beta}{2}\underset{\Pi}{\mathbb{E}}\left[\left\|\hat{\Theta}_i - \bar{\Theta}_i\right\|_2^2\right].
\end{aligned}
$$
$$
\text{(From Jensen's inequality)}
$$

Recall that the distribution $\pi(\mathcal{D})$ satisfies $\mathrm{LS}(\lambda/B)$ inequality. On choosing the coupling $\Pi$ to be the infimum, we get the following bound on Wasserstein's distance from Lemma 22.

$$
\underset{\Pi}{\inf}\sqrt{\underset{\hat{\Theta}_i, \bar{\Theta}_i \sim \Pi}{\mathbb{E}}\left[\left\|\hat{\Theta}_i - \bar{\Theta}_i\right\|_2^2\right]} = \mathrm{W}_2\left(\hat{\Theta}_i, \bar{\Theta}_i\right) \leq \sqrt{\frac{2B\sigma^2}{\lambda}\mathrm{KL}\left(\mu_i\|\pi(\mathcal{D}_i)\right)} \leq \sqrt{\frac{2\varepsilon_{\mathrm{dd}}B\sigma^2}{\lambda}}. \tag{143}
$$

The last inequality above follows from monotonicity of Rényi divergence in $q$ and the fact that $\lim_{q\to 1}\mathrm{R}_q\left(\nu\|\nu'\right) = \mathrm{KL}\left(\nu\|\nu'\right)$. Therefore, on combining all the bounds we get

$$
\mathrm{err}(\hat{\Theta}; \mathcal{D}) \leq L\sigma\sqrt{\frac{2\varepsilon_{\mathrm{dd}}B}{\lambda}} + \frac{\varepsilon_{\mathrm{dd}}B\sigma^2(\lambda+\beta)}{\lambda} + \frac{d\sigma^2}{2}\left(\log\frac{\beta+\lambda}{\lambda} + \sqrt{B}\right) = O\left(\sigma\sqrt{\varepsilon_{\mathrm{dd}}} + d\sigma^2\right). \tag{144}
$$

Note that if the constraints on $K_{\mathrm{A}}$ and $K_{\bar{\mathrm{A}}}$ in (136) and on $\sigma^2$ in (135) are equalities instead, we have

$$
\sigma^2 = \frac{2qBL^2}{\lambda\varepsilon_{\mathrm{dp}}n^2}\log\left(\frac{q\log(B)}{\varepsilon_{\mathrm{dd}}}\right) = \tilde{O}\left(\frac{q}{\varepsilon_{\mathrm{dp}}n^2}\right), \tag{145}
$$

where $\tilde{O}(\cdot)$ hides logarithmic factors. Therefore, the excess empirical risk has an order

$$
\mathrm{err}(\hat{\Theta}; \mathcal{D}) = \tilde{O}\left(\frac{1}{n}\sqrt{\frac{q\varepsilon_{\mathrm{dd}}}{\varepsilon_{\mathrm{dp}}}} + \frac{dq}{\varepsilon_{\mathrm{dp}}n^2}\right). \tag{146}
$$

$\square$

