# OpenReview forum: "Forget Unlearning: Towards True Data-Deletion in Machine Learning"
_ICLR.cc/2023/Conference — Submitted to ICLR 2023_

### Official Review · Reviewer_hke4 · 2022-10-25

**Confidence:** 4
**Correctness:** 3
**Technical Novelty And Significance:** 4
**Empirical Novelty And Significance:** Not applicable
**Recommendation:** 6

**Clarity, Quality, Novelty And Reproducibility:**

I think the paper is well-written and coherent. Overall, I found the flow of ideas natural. There are some places where it gets a little heavy on notation but it is perhaps unavoidable. The problem studied as well the ideas presented are indeed novel, interesting and of high quality. For detailed concerns/questions regarding some of the claims and how they tie together in the overall story, please see the preceding question.
Reproducibility: Does not apply.

**Strength And Weaknesses:**

Strengths

1. The problem and the message of the paper is definitely interesting and important especially since the area of machine unlearning is up and coming. It rightly distinguishes between the unlearning criterion used in prior works (indistinguishibility to retraining) and the informal goal of implementing the right to be forgotten guideline.

2. The paper is well-written and presents a systematic investigation of deficiencies in prior abstraction of the problem. Furthermore, it contributes new and compelling language to capture various important entities and their interaction in unlearning properties and to imbibe desirable properties towards a definition.

Weaknesses and questions:

Admittedly, some of the weaknesses I list below may stem from my potentially incorrect understanding, so I would be grateful if the authors can correct me if I am missing something.

**Proposed definition**: The authors show that the prior definition of unlearning is neither complete nor sound. To remedy this, the authors propose a new definition. However, it seems that the authors do not show that this new definition satisfies (some degree of) soundness and completeness, apart from informal justifications. Is there a rigorous argument here? If yes, please point me to it in case I missed it.

**Problem with internal data strucutres**: One of the key requirements that the authors posit in desirable unlearning algorithms is that they don't maintain any internal data-structure. Their key argument towards this are Thms 3 and 4 wherein they construct problems and a valid learning/unlearning pair with internal state such that after some updates, an adversary can infer a deleted record. However, it seems to me that this is possible even with no internal state. Consider the problem of mean (or sum, for simplicity) estimation and the perfect learning/unlearning pair for it. For all 1-non-adaptive requester, an adversary can completely recover its data record by subtracting the updated estimate from the original estimate, right?
If the above reasoning is correct, then there is something missing in the deduction of the stateless requirements from Thm3/4.
What am I missing here?

Relatedly, I don't understand what *unsafe* means in the context of internal data-structures? For example, in Table 1, the authors correctly list that the prior work of Ullah et al. maintain internal data-structure, but therin, the unlearning being indistinguishable to retraining criterion is applied not just to the model output but the internal data structure as well. In that case, is this still considered unsafe? If yes, then can the authors provide a rigorous definition of safety here?


**Reduction from adaptive to non-adaptive updates**: The recent work of Gupta et al. 2021 also studied machine unlearning against adaptive updates and similarly proved  reductions (as the authors do) using differential privacy. However, a key difference is that Gupta et al. required DP to hold with with regard to a change of description of internal randomness as opposed to standard data item replacement. I understand that the notion of unlearning is different in both papers, but I wonder if there is something more which allows the authors to use standard DP?
In any case, the authors should provide a discussion on differences with the result in the aforementioned work, which is currently missing.

**New definition but similar algorithmic techniques**:
As an example, authors study the design of learning and unlearning algorithms for smooth convex empirical risk minimization under their proposed unlearning criterion.
 The authors' proposed method is based on Noisy gradient descent from the differential privacy literature. This (or its variants) is the method used in many prior works on unlearning (albeit some are stateful, which the authors discard owing "safety" issues). I wonder therefore if it is possible to a-posteriori reason about their unlearning properties (with possibly worse parameters) in the language proposed by the authors. Or is there something fundamental in the algorithmic differences of the authors proposed method?

**Computational savings in Noisy GD**:
 The authors argue that their unlearning method based on Noisy GD demonstrates non-trivial savings of computation compared to retraining with their learning algorithm (Noisy GD).
However, noisy "full" gradient descent is an inefficient (in the sense of gradient complexity) method for the problem of (privately) optimizing smooth convex losses -- it seems that the gradient complexity is $O(\kappa n)$ whereas there exists private methods based on SVRG with $O(n+\kappa)$ gradient complexity or even better.
Now, if retraining were a viable unlearning method, the this computational savings of retraining over a slow learning method may be problematic.
In any case, ideally, we would want unlearning methods for fast learning methods for the problem.
So, it would be good if the authors can expand on if there is something important about the structure of Noisy GD updates, or is it possible to obtain the same results via a faster but (a somewhat) black-box DP optimization routine.


**Elaboration of proof techniques**:
The authors mention that "...techniques we develop for analyzing data deletion differ tremendously from those used to argue differential privacy". It would have been useful if the authors could include some more details on this claim (maybe some key lemmas) in the main text itself.

Minor:

In the conclusion, the authors state that "we showed that differential privacy is necessary for data deletion when requests are adaptive". I did not see such a strong claim formally in the paper.

Typo? -- In page 2, the authors say "B) generates models that are indistinguishable from any distribution independent of the deleted records". Based on their definition 4, shouldn't it be "from some distribution independent of the deleted record"?


**Summary Of The Paper:**

The paper challenges the unlearning criterion used in prior works, indistinguishibility to retraining, for ensuring the right to be forgotten guideline. They argue that this criterion itself is neither complete nor sound. The authors then propose a new unlearning criterion and design learning and unlearning algorithms for the problem of optimizing smooth convex and non-convex losses.

**Summary Of The Review:**

Overall, I think that this paper rigorously challenges the existing formulation of a problem which has been used in multiple prior works and thus is important. The paper is well-written and I enjoyed reading the paper. I have some concerns and suggestions about some technical details of their claims (see above), and I hope that the authors response would help towards addressing these.

-----
Update: Score changed from 8 to 6 owing to potentially significant update to the paper post release of first set of reviews.

---

> ### Author Response · Authors · 2022-11-14
> **Response to reviewer hke4**
>
> We thank the reviewer for their positive feedback on our work and answer their questions as follows.
>
> **On Soundness and Completeness of Proposed Definition.** We acknowledge that we didn't provide formal proof of soundness or completeness for our data-deletion Definition 4.1. To remedy that, we provide proof of soundness in our [response](https://openreview.net/forum?id=goLFJ0ZNwl&noteId=M0QMHr_kbwl) to reviewer JXqD. We also clarify that our definition might not be complete, although it captures more valid data-deletion algorithms than some of the prior definitions.
>
> **Problem with internal data structures.**  We clarify that Section 3 identifies two independent reasons for the violation of RTBF in the considered threat model. Only one of these vulnerabilities is due to the reliance on internal data structures, while the other is due to the possibility of adaptive edit requests. Specifically, our Theorem 3 presents a construction of a data-deletion algorithm that maintains internal data-dependent states, leading to blatant non-privacy in the long run. On the other hand, our Theorem 4 presents a construction for a *stateless deletion-via-retraining* algorithm that also leads to blatant non-privacy when the edit requests are generated adaptively.
>
> The example that the reviewer describes for reconstructing the deleted record by looking at both the models before and after deletion may not technically be a violation of RTBF, as one can argue that the source of information for this attack is the outcome published before the deletion. Our threat model focuses on only the leakage from the model/outcome releases *after* the deletion of the target record. For more details on our threat model, we refer the reviewer to our comment [here](https://openreview.net/forum?id=goLFJ0ZNwl&noteId=wYYuMFiBU-).
>
> We clarify that by *unsafe*, we mean any hidden computation that is *not independent* of the previously deleted records but affects the observable outcomes. Under this definition of *safety*, we agree with the reviewer that Ullah et al.'s algorithms indeed do not maintain any *unsafe* hidden data structures as the perfect indistinguishability they guarantee applies to all the internal states as well. We will fix this mistake in Table 1.
>
> **Reduction from adaptive to non-adaptive updates.** Thank you for this question. Yes, our reduction theorem is fundamentally different than that of Gupta et al., 2021. We refer the reviewer to our [this comment](https://openreview.net/forum?id=goLFJ0ZNwl&noteId=dvw6JdqPe-) for details. We also plan on including this discussion in our paper's appendix.
>
> **New definition but similar algorithmic techniques.** We emphasize that our paper challenges the data-deletion definitions in the literature and not the algorithms themselves. We believe it should be possible to prove data-deletion bounds (as per our Definition 4.1) for some existing algorithms.
>
> **Computational savings in Noisy GD.** To the best of our knowledge, Noisy-GD is the *fastest optimal* method for privately optimizing strongly-convex smooth losses. The DP-SVRG algorithm by Wang et al., 2017, that the reviewer is referencing indeed has a runtime of $O(n + \kappa)$ but provides a sub-optimal utility of $O(\frac{d \log(n)\log(1/\delta)}{n^2\epsilon^2})$ under an $(\epsilon,\delta)$-DP constraint. In contrast, Noisy-GD has an optimal utility guarantee of $O(\frac{d\log(1/\delta)}{n^2\epsilon^2})$ with a runtime of $O(n\kappa)$, as proved in Chourasia et al., 2021, which matches the known lower bound of $\Omega(\frac{d}{n^2\epsilon^2})$ for any $(\epsilon, \delta)$-differentially private empirical risk minimization on strongly convex losses.
>
> > it would be good if the authors can expand on if there is something important about the structure of Noisy GD updates, or is it possible to obtain the same results via a faster but (a somewhat) black-box DP optimization routine.
>
> We don't think Noisy-GD has any special properties, and there could be equally useful (if not better) solutions to the data-deletion problem.
>
> **Elaboration of proof techniques.** Due to space constraints, we can only elaborate on the proof techniques in the appendix.
>
> **DP as a necessary condition to deletion.** We acknowledge that we did not formally prove this claim. To remedy that, we provide a Theorem in [our response](https://openreview.net/forum?id=goLFJ0ZNwl&noteId=axMPNvFaww) to reviewer JXqD that we will include in the paper.
>
> > Typo? -- In page 2, the authors say "B) generates models that are indistinguishable from any distribution independent of the deleted records". Based on their definition 4, shouldn't it be "from some distribution independent of the deleted record"?
>
> Thank you for identifying this typo; we will fix it. We also refer the reviewer to our [updated definition](https://openreview.net/forum?id=goLFJ0ZNwl&noteId=WpMjPtFvB_) that avoids using the term 'independence' for mathematical consistency.

---

> > ### Comment · Reviewer_hke4 · 2022-11-20
> > **Thank you**
> >
> > I thank the authors for their detailed response to all my questions. I still like the paper and will keep my score as is. In the response, the authors presented multiple new results which were not part of the original submission -- these include  soundness and completeness discussion of their definition and necessity of DP. I encourage the authors to incorporate (some parts of) these results into the main text.

---

> > > ### Author Response · Authors · 2022-11-21
> > > **Thank you for the discussions**
> > >
> > > Thank you for appreciating our research and the insightful comments that helped us bridge the gaps in our claims. We have included these new results and updated our presentation of them in our revised version. Hopefully, this updated version and our discussions have convinced you of the correctness of our claims and that you would consider updating your correctness score to 4.

---

### Official Review · Reviewer_vRVP · 2022-10-25

**Confidence:** 3
**Correctness:** 2
**Technical Novelty And Significance:** 3
**Empirical Novelty And Significance:** Not applicable
**Recommendation:** 6

**Clarity, Quality, Novelty And Reproducibility:**

Threat model: The paper is missing a clear threat model: who is the attacker, what can they do, what information they have. As a result it is difficult to evaluate claims that other papers do not satisfy the proper definition of deletion.

State: please consider expanding this discussion. How can a server not having any information about original data "delete" a point that it forgot was there? how would it know that the request is legitimate and retrieve data is needs to delete?

Adaptivity: the results presented in the paper seem to be in a non-adaptive case, however arguments showing that previous work does not satisfy the definition seem to be adaptive. It would be great if the authors could clarify this aspect.

Algorithm: as this work presents a way to unlearn data, an algorithm showing the steps for learning and unlearning would facilitate the presentation.

Please consider summarising the properties that previous works fail to achieve in your definition. Counter-example of executing previous algorithms and showing where they would fail help.

Please consider how re-training from scratch would or would not satisfy your definition. It would be good to be clear from the beginning that not all definitions fail (if it is the case).

**Strength And Weaknesses:**

Strengths:
- the problem of unlearning is important
- observation that the data maybe added later on is an important one

Weaknesses:
- missing threat model
- the definition requiring an algorithm to be stateless does not seem to be intuitive (see below)
- the adaptive vs non-adaptive aspect is confusing
- summary of where and which previous definitions fail is missing


Observation: this paper provides a theoretical result.

**Summary Of The Paper:**

The authors consider the definition of unlearning developed in previous work. The paper claims that these definitions do not satisfy the true meaning of unlearning as data could be added back in at later points in time. To this end, the propose that the new definition should be stateless and algorithms for deletion and training should satisfy DP-like guarantees.

**Summary Of The Review:**

The paper makes an observation that some of the existing unlearning definitions may not capture all properties one needs from unlearning. This is an important observation. It suggests a new definition. However, without clear threat model and attack capabilities it is difficult to evaluate whether it is the definition that fails or the assumptions on the attacker that fail. The method and the paper could also be presented in a clearer way (e.g., with counter-examples that follow previous definitions an show how they fail).

---

> ### Author Response · Authors · 2022-11-13
> **Response 1 to reviewer vRVP**
>
> We thank the reviewer for all the constructive feedback for improving our paper's clarity. We address the reviewer's doubts here and describe our planned changes to the paper.
>
> > Threat model: The paper is missing a clear threat model: who is the attacker, what can they do, what information they have. As a result it is difficult to evaluate claims that other papers do not satisfy the proper definition of deletion.
>
> $$
> \newcommand{\nc}{\newcommand}
> \nc{\ind}{\text{ind}}
> $$
>
> Our paper investigates the failure of prior definitions of *certified data-deletion* proposed for ensuring the "Right to be Forgotten" guideline. To do so, we consider the following threat model. The *data curator* has access to an initial database $\mathcal D_0$ consisting of $n$ ordered records contributed by individual users. The curator trains a hidden ML model (plus any additional associated metadata that can be used to speed up data-deletion later) $\hat\Theta_0 = \mathcal A(\mathcal D_0)$ using a learning algorithm $\mathcal A$ and publishes the outcome $\phi_0 = f_{\text{publish}}(\hat\Theta_0)$. At any step, $i \geq 1$, any set of ($r$ many) users may request their records to be deleted, and (an equal number of) new users may request their records to be added. We model such a transaction through an edit request $u_i = \\{\langle \ind_1, x_1 \rangle, \cdots, \langle \ind_r, x_r \rangle\\}$ that, when applied to the current database $\mathcal D_{i-1}$, replaces $r$ records at indices $\ind_1, \cdots, \ind_r$ with new records $x_1, \cdots, x_r$. In response to the edit request $u_i$, the curator runs a *data-deletion* algorithm $\mathcal R$ that updates the current hidden ML model (as well as the associated metadata) $\hat\Theta_{i-1}$ to produce the next model (plus metadata) $\hat\Theta_i = \mathcal R(\mathcal D_{i-1}, u_i, \hat\Theta_{i-1})$, and releases the updated outcome $\phi_i = f_{\text{publish}}(\hat\Theta_i)$.
>
> In our threat model, for an arbitrary step $i \geq 1$, an adversary is interested in finding out the identity of a record in the database $\mathcal D_{i-1}$ that was deleted by the edit request $u_i$. Our adversary only has access to the releases by the curator *after deletion*, i.e., the infinite post-deletion sequence $\phi_i, \phi_{i+1}, \cdots$. Additionally, this adversary is assumed to know how users might react to a published outcome but doesn't observe either the published outcome or the users' requests. That is to say, our adversary *knows some dependence relationship* between random variables $\phi_0, \cdots, \phi_{i-1}$ and $u_1, \cdots, u_{i}$, but does not explicitly observe these random variables. For instance, an adversary might know that if the outcome $\phi_1$ predicts that "Donald Trump is winning the election," then some democratic users might delete their data while some new republican users might contribute their data to the curator. So, even though the adversary does not observe the actual outcome $\phi_1$ or the ensuing edit request $u_2$, and so on, it can still exploit knowledge about this dependence a-posteriori to infer the identity of a deleted record.
>
> To capture the *worst-case knowledge* about the dependence between unobserved outcomes and edit requests, we model our adversary to have the power to design an adaptive requester $\mathcal Q$ that sends the edit requests $u_1, \cdots, u_{i-1}$  in response to published outcomes $\phi_1, \cdots, \phi_{i-1}$ following some rule:
> $$
> u_1 = \mathcal Q(\phi_0), u_2 = \mathcal Q(\phi_0, u_1, \phi_1), \cdots, u_{i-1} = \mathcal Q(\phi_0, u_1, \phi_1, u_2, \cdots, u_{i-2}, \phi_{i-2}).
> $$
> However, the adversary *does not* observe interaction transcript $\\{\phi_0, u_1, \phi_1, \cdots u_{i-1}, \phi_{i-1}\\}$.
>
> *Note.* We will modify Section 3 to make the threat model clear.
>
> > State: please consider expanding this discussion. How can a server not having any information about original data "delete" a point that it forgot was there? how would it know that the request is legitimate and retrieve data is needs to delete?
>
> When processing a deletion request $u_i$, the curator has access to the database $\mathcal D_{i-1}$ that contains the records that are being requested to be deleted as well as the hidden ML model + metadata trained on $\mathcal D_{i-1}$. To verify that request $u_i$ is legitimate, the curator can simply check it against the current database $\mathcal D_{i-1}$. To process the deletion request $u_i$, the curator needs to do three things:
> - Update the hidden ML model + corresponding metadata (which we refer to as the hidden state) by running the data-deletion mechanism $\hat\Theta_i = \mathcal R(\mathcal D_{i-1}, u_1, \hat\Theta_{i-1})$.
> - Update the current database by executing $\mathcal D_{i-1} \circ u_i \rightarrow \mathcal D_i$.
> - Release the publishable outcome $\phi_i = f_{\text{publish}}(\hat\Theta_i)$.

---

> ### Author Response · Authors · 2022-11-13
> **Response 2 to reviewer vRVP**
>
> > Please consider summarising the properties that previous works fail to achieve in your definition. Counter-example of executing previous algorithms and showing where they would fail help.
>
> Our work challenges prior certifications proposed for the "Right to be Forgotten" in the form of unlearning/data-deletion guarantees by [Guo et al., 2019, Sekhari et al., 2021, Neel et al., 2021, Gupta et al., 2021]. For these definitions, we identify two distinct causes of critical failures leading to blatant non-privacy that we summarise as follows.
>
> - **Violation of RTBF due to hidden states.** The definitions of Gupta et al., 2021, and Neel et al., 2021 require indistinguishability over only the published outcomes released after applying $f_{\text{publish}}(\cdot)$ to the hidden ML model + metadata. In Theorem 3, we demonstrate that algorithms certified under this definition can potentially lead to blatant non-privacy of a deleted record through the post-deletion releases in the long run, even when the edit requests are non-adaptive.
> - **Violation of RTBF due to adaptive requests.** The definitions of Guo et al., 2019, Sekhari et al., 2021, Gupta et al.,2021, Neel et al.,2021 are all vulnerable to attacks that exploit knowledge about the dependency between edit requests and released models, even when the indistinguishability is over the ML model + metadata, i.e., no publish function $f_{\text{publish}}$ is used, or in other words, the deletion mechanism does not have hidden states. In Theorem 4, we demonstrate that for algorithms certified under their definitions, an adversary observing the post-deletion releases may infer the identity of a deleted record by looking for the dependence pattern encoded by the adaptive requests in the post-deletion releases.
>
> Additionally, we also demonstrate that these definitions are incomplete by providing examples of valid data-deletion mechanisms that fail to satisfy them. We refer the reviewer to additional comments [here](https://openreview.net/forum?id=goLFJ0ZNwl&noteId=j_YW6nigsLc) and [here](https://openreview.net/forum?id=goLFJ0ZNwl&noteId=aezU4gdYDf) for additional details about unsoundness and incompleteness of existing definitions.
>
> We remark that our criticisms over the data-deletion algorithms proposed by these works are indirect; the only reason to not trust existing data-deletion algorithms is because the certifications provided on them are unsound. The algorithms themselves could be valid and might potentially be certified a-posteriori under our sound data-deletion definition.
>
> *Note.* We will clarify the two critical vulnerabilities that existing definitions overlook by modifying Section 3.
>
> > Adaptivity: the results presented in the paper seem to be in a non-adaptive case, however arguments showing that previous work does not satisfy the definition seem to be adaptive. It would be great if the authors could clarify this aspect.
>
> Our Definition 4.1 of data-deletion that we propose to tackle the two aforementioned critical vulnerabilities in existing definitions and offers a useful reduction. In Theorem 5, we show that if $(\mathcal A, \mathcal R)$ satisfies data-deletion under non-adaptive requests and is also differentially private with respect to the replacement of single records, then $(\mathcal A, \mathcal R)$ satisfy data-deletion under adaptive requests as well. Since the proposed learning and data-deletion mechanisms are based on Noisy-GD, a well-known DP algorithm, our reduction Theorem 5 directly extends the non-adaptive deletion guarantees in our Theorems 6 and 7 to the adaptive setting.
>
> > Algorithm: as this work presents a way to unlearn data, an algorithm showing the steps for learning and unlearning would facilitate the presentation.
>
> We describe our proposed learning and data-deletion algorithm pair $(\mathcal A_{\text{Noisy-GD}}, \mathcal R_{\text{Noisy-GD}})$ in Definition 5.1.
>
> *Note.* To aid the presentation, we will provide algorithmic pseudocode for these algorithms instead.
>
> > Please consider how re-training from scratch would or would not satisfy your definition. It would be good to be clear from the beginning that not all definitions fail (if it is the case).
>
> We clarify that retraining from scratch using a non-DP learning algorithm would not satisfy our data-deletion definition. We refer the reviewer to [this discussion](https://openreview.net/forum?id=goLFJ0ZNwl&noteId=axMPNvFaww) that DP is necessary for satisfying our data-deletion guarantee.

---

### Official Review · Reviewer_JXqD · 2022-10-26

**Confidence:** 4
**Correctness:** 2
**Technical Novelty And Significance:** 2
**Empirical Novelty And Significance:** 3
**Recommendation:** 5

**Clarity, Quality, Novelty And Reproducibility:**

All proofs are included, and except for a few minor typos the paper seems to be well-written. Please refer to other sections for more details about clarity.

**Strength And Weaknesses:**

Strengths: The paper considers the important question of properly defining data deletion and gives algorithms for data deletion, in the adaptive and online unlearning setting. The paper provides a rigorous theoretical exploration of issues of a prior definition, and a fix to it.

Weaknesses:
1. I think the paper creates a misleading impression of the prior works, as:
  - They only consider the setting of iterative unlearning (multiple {adaptive or non-adaptive} unlearning requests) which, in my opinion, does not comprise the fundamental deletion setting. In particular, the two-stage setting that consists of a single learning state, and a single unlearning stage (to unlearn K many points) has been completely ignored in the paper. All their constructions to invalidate the soundness of machine unlearning definition completely breaks down in this setting.


  - They only consider the indistinguishability definition of Neel et. al. 2021, Gupta et. al. 2021, and extrapolate the failure of these definitions under adaptive queries to argue that all prior machine unlearning works are wrong/incomplete. Contrary to how they pose the prior works, there already exist definitions in the literature that are both sound and complete. Consider the definition of Sekhari et. al. 2021 - It is sound as there is no f_publish and only two stages of interaction, and complete since it only compares the output after the unlearning state (on both original and updated datasets) and hence, an unlearning algorithm that outputs a fixed untrained model is a valid unlearning algorithm.




2. Following up on the above comment, Theorem 3 only holds when there is a separate f_publish, but not all unlearning definitions distinguish f_publish and algorithms output (e.g. Sekhari et. al. 2021, Ginart et. al. 2019).

3. The authors claim in the abstract, and in various places in the paper, that it is necessary to satisfy DP to ensure true data deletion. I do not see any formal claim anywhere arguing that DP is necessary. From the definitions and results, it seems to be a sufficient condition rather than a necessary one. Please elaborate.

4. What are the DP guarantees for the noisy SGD algorithm in Algorithm 1. Is it the exact same noise as DP-SGD? Are the authors essentially running a pure DP algorithm, and then just arguing in hindsight that it is also unlearning?

5. In the analysis for noisy SGD, the authors only consider empirical risk guarantees. Do the benefits over retraining (reported in Table 1) also hold when we care about test loss performance?

6. Do the authors have any proof that their definition of data deletion is sound or complete?

7. The risk of adaptive unlearning requests was also previously explored by Gupta et. al.

8. Minor issues and typos:
   - Definition 2.5, eqn 8 is not clear. I feel it will help to elaborate a bit more on what it means.
   - Definition 4.1 has a typo “all edit requests u = … \in U^{i}” instead of U^1.
   - Independence is defined for two random variables (and not two distributions).  Please fix the paper to not use phrases like “there exists a distribution \pi_i^u that is independent of the record being deleted” (e.g. in defn 4.1). This is not mathematically consistent.
   - Change the notation R to denote the unlearning algorithm (it is confusing as R should ideally be used to denote the retraining algorithm).

I would be happy to engage further with the authors on any of the above comments.

**Summary Of The Paper:**

The paper looks at the problem of ensuring the right-to-be-forgotten (RTBF), under multiple adaptive edit requests to the dataset. The paper has three main contributions (a) they show that the machine unlearning definitions used in the prior works are not sufficient to ensure RTBF, especially when the edit requests are adaptive (they show that it is neither sound nor complete), (b) They provide a new definition for data deletion, (c) They provide an analysis of noisy-SGD algorithm under this new definition. This is a completely theoretical paper and there are no experiments.



**Summary Of The Review:**

As the paper stands, I recommend rejecting it. I feel that the authors are trying to create a wrong impression of the current machine-unlearning literature, which is my primary reason for rejection.  I also feel that the paper is incomplete (as discussed more in the weaknesses). In particular, it is not clear whether DP is actually necessary for deletion in the adaptive setting (as claimed by the authors), and whether their new definition is sound or complete.

---

> ### Author Response · Authors · 2022-11-10
> **Response 1 to reviewer JXqD ([Q1])**
>
> **[Q1]** We show here that our criticisms on both soundness and completeness of existing unlearning notions in Section 3 also apply to the definitions of Ginart et al., 2019, Sekhari et al., 2021, Guo et al., 2019, even in the two-stage deletion setting. We elaborate on this claim here.
>
> **Definition ($(\epsilon, \delta)$-certified removal [Guo et al., 2019]).** A removal mechanism $\mathcal R$ performs *$(\epsilon, \delta)$-certified removal* for learning algorithm $\mathcal A$ if for all databases $\mathcal D \subset \mathcal X$ and deletion subset $S \subset D$,
> $$
> \mathcal R(\mathcal D, S, \mathcal A(\mathcal D)) \stackrel{\epsilon, \delta}{\approx} \mathcal A(\mathcal D\setminus S).
> $$
> **Definition ($(\epsilon, \delta)$-unlearning [Sekhari et al., 2021]).** For all $\mathcal D \subset \mathcal X$ of size $n$ and deletion subset $S \subset \mathcal D$ such that $|S| \leq m$, a learning algorithm $\mathcal A$ and an unlearning algorithm $\mathcal{R}$ is *$(\epsilon, \delta)$-unlearning* if
> $$
> \mathcal{R}(T(\mathcal D), S, \mathcal A(\mathcal D)) \stackrel{\epsilon, \delta}{\approx} \mathcal{R}( T(\mathcal D \setminus S), \varnothing, \mathcal{A}(\mathcal D \setminus S)),
> $$
> where $\varnothing$ denotes the empty set and $T(\mathcal D)$ denotes the data statistics available to $\mathcal{R}$.
>
> **Definition (Data Deletion Operation [Ginart et al.,2019]).** Fix any dataset $\mathcal D \subset \mathcal X$ and learning algorithm $\mathcal A$. Operation $\mathcal R$ is a *deletion operation* for $\mathcal A$ if $\mathcal R(\mathcal D, S, \mathcal A(\mathcal D)) \stackrel{0,0}{\approx} \mathcal A(\mathcal D \setminus S)$ for any set $S \subset \mathcal D$ selected *independently* of $\mathcal A(\mathcal D)$.
>
> **Unsoundness.** The definitions of Guo et al., 2019, and Sekhari et al., 2021, make no assumptions about how the deletion batch $S$ is selected. So, request $S$ can depend on $\mathcal A(\mathcal D)$. This dependence is common in the real world; for example, a user deletes her information if she doesn't like what model $\mathcal A(\mathcal D)$ reveals about her. We present the following construction along the same lines as our proof in Theorem 4 to show that the definitions of Guo et al., 2019, and Sekhari et al., 2021, are unsound.
>
> For the universe of records $\mathcal X = \\{-2, -1, 1, 2\\}$, consider the following learning and unlearning algorithms.
> $$
> \mathcal A(\mathcal D) = \sum_{x \in \mathcal D} x, \quad \text{and} \quad \mathcal R(\mathcal D, S, \mathcal A(\mathcal D)) = \sum_{x \in \mathcal D \setminus S} x
> $$
> Note that for any $\mathcal D \subset \mathcal X$, the above algorithm pair $(\mathcal A, \mathcal R)$ satisfies all the aforementioned unlearning guarantees for $\epsilon = \delta = 0$ and $T(\mathcal D) = \mathcal D$. Now consider two neighboring databases $\mathcal D = \\{-2, -1, 2\\}$, $\mathcal D' = \\{-2, 1, 2\\}$ and the following dependence between the learned model $\mathcal A(\mathcal{\bar D})$ and request $S$:
> $$
> S = \\{x < 0: \forall x \in \mathcal X\\}\ \text{if} \ \mathcal A(\mathcal{\bar D}) < 0,\ \text{otherwise}\ S = \\{x > 0: \forall x \in \mathcal X\\}.
> $$
> Knowing this dependence, an adversary can distinguish whether $\mathcal{\bar D} = \mathcal D$ or $\mathcal{\bar D} = \mathcal D'$ by looking only at $\mathcal R(\mathcal{\bar D}, S, \mathcal A(\mathcal{\bar D}))$. This is because if $\mathcal{\bar D = \mathcal D}$, then the outcome after unlearning is $2$, and if $\mathcal{\bar D} = \mathcal D'$, the observation after unlearning is $-2$.
>
> So, even though $(\mathcal A, \mathcal R)$ satisfies the guarantees of Guo et al., 2019, and Sekhari et al., 2021, it blatantly reveals the identity ($-1$ or $1$ ) of a deleted record to an adversary observing only the post-deletion release.
>
> The definition of Ginart et al., 2019, assumes that the requests $S$ are selected independently of $\mathcal A(\mathcal D)$. So, our construction does not apply, keeping the possibility that their definition is sound. We remark, however, that algorithms satisfying their definitions cannot be trusted in settings where we expect some dependence between deletion requests and the learned models.
>
> **Incompleteness.** All the above deletion definitions are incomplete. Consider an unlearning algorithm $\mathcal R$ that outputs a fixed output $x_1 \in \mathcal X$ if the deletion request $S = \varnothing$ and outputs another fixed output $x_2 \in \mathcal X$ if the deletion request $S \neq \varnothing$. It is easy to see that $\mathcal R$ is a valid deletion algorithm as its output does not depend on the input database or the learned model. However, note that $\mathcal R$ does not satisfy the unlearning definition of Sekhari et al., 2021, for any learning algorithm $\mathcal A$. And, for a learning algorithm $\mathcal A(\mathcal D) = \sum_{x \in \mathcal D} x$, one can also verify that the pair $(\mathcal A, \mathcal R)$ does not satisfy the definitions of Guo et al., 2019, and Ginart et al., 2019.

---

> ### Author Response · Authors · 2022-11-10
> **Response 2 to reviewer JXqD ([Q2])**
>
> **[Q2]** Our Theorem 3 shows a violation of the "Right to be Forgotten," specifically in the online setting when the unlearning mechanism $\mathcal R$ is allowed to maintain secret data-dependent states. We clarify that the necessary condition for the vulnerability we demonstrate in Theorem 3 is that the deletion algorithm relies on hidden data-dependent states. For example, if the curator caches statistics like database mean and variance, but does not update it alongside modifications to the database after processing the edit requests and uses it to calibrate future models, then the violation we demonstrate might occur. These hidden data-dependent states may provide an unbounded channel for information leakage regarding the deleted record that seeps through every future release after deletion, causing blatant non-privacy in the long run. We point out that as per the problem formulation of Ginart et al., 2019, in Section 3 of their paper, they permit an unlearning algorithm to have access to "arbitrary metadata that is not necessarily used at inference time." They further clarify that "such metadata could include data structures or partial computations that can be leveraged to help with subsequent deletions." Our Theorem 3 shows the privacy risk of doing precisely that, which is both a novel and an extremely important vulnerability to consider while designing data-deletion algorithms.

---

> ### Author Response · Authors · 2022-11-10
> **Response 3 to reviewer JXqD ([Q3])**
>
> $$
> \newcommand{\nc}{\newcommand}
> \nc{\mc}{\mathcal}
> \nc{\def}{\stackrel{\text{def}}{=}}
> \nc{\TV}[1]{\mathbf{TV}\left(#1\right)}
> \nc{\D}{\mc{D}}
> \nc{\Db}{\mc{\bar{D}}}
> \nc{\R}{\mc{R}}
> \nc{\A}{\mc{A}}
> \nc{\U}{\mc{U}}
> \nc{\Q}{\mc{Q}}
> \nc{\X}{\mc{X}}
> \nc{\O}{\mc{O}}
> \nc{\ep}{\epsilon}
> \nc{\de}{\delta}
> \nc{\H}[3]{#3\left(#1\middle\Vert#2\right)}
> \nc{\Re}[2]{\H{#1}{#2}{\mathrm{R}_q}}
> \nc{\KL}[2]{\H{#1}{#2}{\mathbf{KL}}}
> \nc{\T}{\text{Test}}
> \nc{\Tb}{\overline{\T}}
> \nc{\Th}{\Theta}
> \nc{\th}{\theta}
> \nc{\Thh}{\hat\Th}
> \nc{\Ad}{\text{Adv}}
> \nc{\M}{\text{MI}}
> \nc{\ra}{\rightarrow}
> $$
> **[Q3]** We acknowledge that we did not provide formal proof that DP is necessary for data deletion. To remedy this, we present the following theorem.
>
> **Theorem (DP is necessary for data-deletion).** If any randomized learning algorithm $\A:\X^n\ra\O$ is not $(0,\de)$-DP with respect to the replacement of a single record and deletion algorithm $\R:\X^n\times\U\times\O\ra\O$ is not $(0,\de)$-DP with respect to the replacement of a single record that is not being deleted, then the pair $(\A,\R)$ does not satisfy $(q,\frac{\de^4}{2})$-data-deletion guarantee under $1$-adaptive $1$-requesters for any $q>1$.
>
> *Proof.* If $\A$ is not $(0,\de)$-DP with respect to the replacement of a single record, then there exists a pair of neighboring databases $\D, \D'$ such that
> $$
> \TV{\A(\D);\A(\D')}=\sup_{O\subset\O}|\Pr[\A(\D)\in O]-\Pr[\A(\D')\in O]|>\de.
> $$
> Similarly, if $\R$ is not $(0,\de)$-DP with respect to replacement of a single record that is not being deleted, then there exists a pair of databases $\Db,\Db'$ and an update request $u\in\U^1$ such that $\Db\circ u$ and $\Db'\circ u$ are neighboring and for all $\th\in\O$,
> $$
> \TV{\R(\Db,u,\th); \R(\Db',u,\th)}=\sup_{O\subset\O}|\Pr[\R(\Db,u,\th)\in O]-\Pr[\R(\Db',u,\th)\in O]|>\de.
> $$
> Since $\TV{}$ distance is bounded from below in both cases, there exist tests $\T,\Tb:\O\ra\\{0,1\\}$ such that
> $$
> \Ad(\T;\D,\D')\def\Pr[\T(\A(\D))=1]-\Pr[\T(\A(\D'))=1]>\de,
> $$
> and for all $\th\in\O$,
> $$
> \Ad(\Tb;\Db,\Db',u)\def\Pr[\Tb(\R(\Db,u,\th))=1]-\Pr[\Tb (\R(\Db',u,\th))=1]>\de.
> $$
> Assume W.L.O.G. that $\Db,\Db'$ differs at index $1$, while $\D,\D'$ differs at index $n$. To prove the theorem statement, we show that for a starting database $\D_0\in\\{\D,\D'\\}$ and an edit request $u_n$ deleting the differing record at step $n$, there exists a 1-adaptive 1-requester $\Q$ that sends adaptive edit requests $u_1,\cdots,u_{n-1}$ for the first $n-1$ rounds such that there exists no distribution $\pi$ satisfying $\Re{\R(\D_{n-1},u_n,\Thh_{n-1})}{\pi}\leq\frac{\de^4}{2}$ for both choices of $\D_0$, implying that $(\A,\R)$ can't satisfy $(q,\frac{\de^4}{2})$-data-deletion.
>
> Consider the following construction of a 1-adaptive 1-requester $\Q$ that only observes the first model $\Thh_0=\A(\D_0)$.
> $$
> \Q(\Thh_0,u_1,u_2,\cdots,u_{i-1})=\langle i,\Db[i]\rangle\ \text{if}\ \T(\Thh_0)=1,\ \text{otherwise}\ \langle i,\Db'[i]\rangle.
> $$
> This requester $\Q$ transforms any initial database $\D_0$ to $\D_n=\Db$ if the outcome of $\T(\Thh_0)=1$, or to $\D_n=\Db'$ if  $\T(\Thh_0)=0$. Consider an adversary that does not see the interaction transcript $(\Thh_0,u_1,\Thh_1,\cdots,u_n)$, but is interested in identifying whether $\D_0$ was $\D$ or $\D'$. The adversary only gets to observe the output $\Thh_n=\R(\D_{n-1},u_n,\Thh_{n-1})$ after processing the edit request $u_n$ that deletes the differing record in $\D_0$. On this observation, the adversary runs the membership inference test $\M(\Thh_n)=\Tb(\Thh_n)$. For membership advantage defined as $\Ad(\M;\D,\D')\def\Pr[\M(\Thh_n)=1|\D_0=\D]-\Pr[\M(\Thh_n)=1|\D_0=\D']$, this test satisfy
> $$
> \Ad(\M;\D,\D')=\sum_{b\in\\{0,1\\}} \Pr[\M(\Thh_n)=1| \T(\Thh_0)=b,\D_0=\D]\Pr[\T(\Thh_0)=b|\D_0=\D]-\sum_{b\in\\{0,1\\}}\Pr[\M(\Thh_n)=1|\T(\Thh_0)=b,\D_0=\D'] \Pr[\T(\Thh_0)=b|\D_0=\D']
> $$
>
> $$
> \iff\Ad(\M;\D,\D')=\Pr[\Tb(\Thh_n)=1|\D_n=\Db]\Ad(\T;\D,\D')-\Pr[\Tb(\Thh_n)=1|\D_n=\Db']\Ad(\T;\D,\D')
> $$
>
> $$
> \iff\Ad(\M;\D,\D')=\Ad(\Tb;\Db,\Db',u_n)\Ad(\T;\D,\D')>\de^2.
> $$
>
> So, it must be true that the total variation between $\Thh_n$ given $\D_0=\D$ and $\D_0=\D'$ is lower bounded as
> $$
> \TV{\Thh_n\mid_{\D_0=\D};\Thh_n\mid_{\D_0=\D'}}>\de^2.
> $$
> From triangle inequality for total variation and Pinsker's inequality, note that for a random variable $\Th\in\O$ with any arbitrary distribution $\pi$,
> $$
> \de^2<\TV{\Thh_n\mid_{\D_0=\D};\Thh_n\mid_{\D_0=\D'}}\leq\TV{\Thh_n\mid_{\D_0=\D};\Th}+\TV{\Thh_n\mid_{\D_0=\D'};\Th}\leq\sqrt{2\max\left(\KL{\Thh_n\mid_{\D_0=\D}}{\Th},\KL{\Thh_n\mid_{\D_0=\D'}}{\Th}\right)}.
> $$
> Since R\'enyi divergence increases with $q$, for any $q>1$,
> $$
> \max\left(\Re{\Thh_n\mid_{\D_0=\D}}{\Th},\Re{\Thh_n\mid_{\D_0=\D'}}{\Th}\right)>\frac{\de^4}{2}.
> $$
> But, to satisfy $(q,\de^4/2)$-data-deletion, there must exist a mapping $\pi$ for which
> $$
> \Re{\R(\D_{n-1},u_n,\Thh_{n-1})}{\pi(\D_0\circ u_n)}\leq\frac{\de^4}{2}
> $$
> hold for both choices of $\D_0\in\\{\D,\D'\\}$, which isn't possible.

---

> > ### Comment · Reviewer_hke4 · 2022-11-18
> > **Thanks and some clarification questions**
> >
> > Sorry for responding on the thread of response to another reviewer. I had the same question and the authors pointed to this comment in their response. I first thank the authors for providing the above details. However, I think there may be some issues (or I am misunderstanding something) but with a potential fix. It would be good if the authors could clarify these.
> >
> > **$(0,\delta)$-DP is *not really* DP and a mistake(?) in the proof**
> >
> > The original informal claim (without proof) in the paper was that DP is necessary for unlearning. The formal claim pertains to $(0,\delta)$-DP. However, in the $(\epsilon, \delta)$ DP literature, $\delta$ is taken to be very small, often a negligible function in the number of data-points (otherwise, one can just release a record with probability $\delta$ and satisfy $(0,\delta)$-DP, however, this clearly violates privacy).
> > With such choice of $\delta$, I suspect if there are any *interesting* $(0,\delta)$-DP algorithms. So, in that case, the necessary condition would be too strong to design any meaningful schemes. But then the authors indeed propose methods with non-trivial guarantees..
> >
> >
> > The problem (I think) is the very first line of the proof where the authors claim that if an algorithm is **not** $(0,\delta)$, then there exists datasets such that TV distance is at least $\delta$. This is not true -- a $(0.1,\delta)$-DP algorithm is also **not** $(0,\delta)$-DP, but the TV distance relation may not hold here. For a concrete counter-example, consider Gaussian mechanism for $1$-sensitive query with variance $\sigma^2 = 100\log{1/\delta}$; this satisfies (roughly)  $\left(0.1, \delta \right)$-DP but the TV distance, by standard calculations, is (roughly) $\frac{1}{\sigma} = \frac{0.1}{\sqrt{\log{1/\delta}}}$ which may be smaller than $\delta$ for certain values of $\delta$.
> >
> > **Fix by replacing $(0,\delta)$-DP with $\delta$-TV stability**:
> >
> > I think the premise in the statement can be weakened while still allowing for non-trivial schemes such as the authors use.
> > Basically, if you change $(0,\delta)$-DP to $\delta$-TV stability, then the first step of the proof holds trivially. However, importantly, there are $\delta$-TV schemes, for relatively large $\delta$, which are not $(0,\delta)$-DP but $(\epsilon', \delta')$-DP with *meaningful* $\epsilon'$ and $\delta'$. The most basic example (as above) is Gaussian mechanism, and so noisySGD. Hence, this fixed necessary condition would apply to methods like the authors use.
> >
> > However, if this is indeed the fix, then the claim that DP is necesaary is incorrect and should not be made (in the present form), I think.
> >
> > Interestingly, this TV stability condition is also used in the prior work of Ullah et al. (which the authors cite), albeit for a seemingly different purpose of computational efficiency during unlearning.
> >
> >
> > **An unclear step in the proof**:
> >
> > There is still one part of the proof which is unclear to me -- this is when you convert the TV bound to Renyi Divergence via Pinsker's inequality -- Pinsker's inequality only relates TV distance and KL divergence and KL divergence is $q$-Renyi divergence when $\alpha\rightarrow 1$. But, how do you go from here to general $q$? If the authors can provide more details to this step, then that would be helpful.

---

> > > ### Author Response · Authors · 2022-11-19
> > > **Clarifications and Fixes**
> > >
> > > Thank you for your comment! We derived this theorem very recently after reading the reviews, and admittedly we made a mistake (not in the first line of the proof, but in using the extension of Pinsker inequality for R\'enyi divergence). Here we provide answers to the reviewer's questions and also update our original comment with the fixed proof.
> > >
> > > > The problem (I think) is the very first line of the proof where the authors claim that if an algorithm is not $(0,\delta)$, then there exists datasets such that TV distance is at least $\delta$. This is not true -- a $(0.1,\delta)$-DP algorithm is also **not** $(0,\delta)$-DP, but the TV distance relation may not hold here.
> > >
> > > By definition, a mechanism $\mathcal A: \mathcal X^n \rightarrow \mathcal O$ satisfies $(0,\delta)$-DP if and only if for all neighbouring databases $\mathcal D, \mathcal D' \in \mathcal X^n$ and all events $O \subset \mathcal O$, the following inequality holds:
> > > $$
> > > \Pr[\mathcal A(\mathcal D) \in O] \leq \Pr[\mathcal A(\mathcal D') \in O] + \delta.
> > > $$
> > > So, from the contra-positive of this statement, we have that a mechanism $\mathcal A: \mathcal X^n \rightarrow \mathcal O$ **does not** satisfy $(0,\delta)$-DP if and only if there exist neighboring databases $\mathcal D, \mathcal D' \in \mathcal X^n$ and an event $O \subset \mathcal O$ such that
> > > $$
> > > \Pr[\mathcal A(\mathcal D) \in O] > \Pr[\mathcal A(\mathcal D') \in O] + \delta.
> > > $$
> > > The first line in our proof simply uses the above inequality, i.e.
> > > $$
> > > \mathbf{TV}(\mathcal A(\mathcal D); \mathcal A(\mathcal D')) = \sup_{O' \in \mathcal O} | \Pr[\mathcal A(\mathcal D) \in O'] -\Pr[\mathcal A(\mathcal D') \in O']| \geq \Pr[\mathcal A(\mathcal D) \in O] - \Pr[\mathcal A(\mathcal D') \in O] > \delta.
> > > $$
> > >
> > > **Problem with reviewer's counter-example.**
> > > To disprove our claim in the first statement, one needs to show that there exists a mechanism that is **not** $(0,\delta)$-DP but has a total variation smaller than $\delta$. In the example presented, the reviewer argues that there exists a mechanism that satisfies $(0.1, \delta)$-DP but also has a TV distance smaller than $\delta$. This counter-example does not disprove our claim as $(0.1, \delta)$-DP does not imply **not** $(0,\delta)$-DP.
> > >
> > > **Remark.** The purpose of our theorem is to show that if $\mathcal A$ and $\mathcal R$ leak non-negligible information about individual records, then under adaptive requests, a pair $(\mathcal A, \mathcal R)$ can be made to leak non-negligible information about a deleted record even after deletion, meaning that it cannot satisfy any sound data-deletion guarantee. Our assumption of **not** $(0,\delta)$-DP is meant to capture such a non-negligible leakage by $\mathcal A$ and $\mathcal R$. One should interpret our theorem as a simple way to determine if any algorithm pair $(\mathcal A, \mathcal R)$ is a plausible deletion algorithm. If they don't satisfy $(0, \delta)$-DP with negligible $\delta$, then their data-deletion capacity has to be limited.
> > >
> > > **What does it mean for a mechanism to be **not** $(0, \delta)$-DP with non-negligible $\delta$?** We show that if a mechanism $\mathcal A$ is **not** $(0, 1 - e^{-\epsilon} + \delta e^{-\epsilon})$-DP for some $\epsilon$ and $\delta$, then it is also **not** $(\epsilon, \delta)$-DP.
> > >
> > > *Proof.* If $\mathcal A$ is not $(0, 1 - e^{-\epsilon} + \delta e^{-\epsilon})$-DP, then, following the preceding argument, there exists a pair of neighboring databases $\D, \D'$ and an event $O \subset \mathcal O$ such that
> > > $$
> > > \Pr[\mathcal A(\mathcal D) \in O] - \Pr[\mathcal A(\mathcal D') \in O] > 1 - e^{-\epsilon} + \delta e^{-\epsilon}.
> > > $$
> > > So, the advantage for a membership inference attack $\text{MI}(\theta) = \mathbb{I}\\{\theta \in O\\}$ is
> > > $$
> > > \text{Adv}(\text{MI};\mathcal A) = \Pr[\text{MI}(\mathcal A(\mathcal D)) =1] - \Pr[\text{MI}(\mathcal A(\mathcal D')) =1] > 1 - e^{-\epsilon} + \delta e^{-\epsilon}.
> > > $$
> > > Erlingsson et al., 2020, show in their Proposition 2 that for any $(\epsilon,\delta)$-DP mechanism $\mathcal A$, the membership inference advantage of any attack $\text{MI}$ is upper bounded as
> > > $$
> > > \text{Adv}(\text{MI}; \mathcal A) \leq 1 - e^{-\epsilon} + \delta e^{-\epsilon}.
> > > $$
> > > Since that's not the case, the mechanism $\mathcal A$ must not be $(\epsilon,\delta)$-DP.
> > >
> > > **On incorrect use of Pinsker Inequality.** Thank you for directing our attention to this. We used [Theorem 31, Erven et al., 2014] that extends Pinsker inequality to R\'enyi divergences without realizing that it applies only to orders $0 < q < 1$. We fix it in the previous comment by instead using the standard Pinsker inequality and monotonicity property of R\'enyi divergence w.r.t. order $q$.

---

> > > > ### Comment · Reviewer_hke4 · 2022-11-20
> > > > **Thank you and final question on chain of reasoning**
> > > >
> > > > I thank the authors for the clarification and I apologize for the confusion; the authors correctly identified the mistake in my reasoning. A final question I have is how all the arguments above fit together. If I understand correctly, the authors want to show:
> > > >
> > > > Claim: Not $(\epsilon, \delta)$-DP for reasonable $\epsilon, \delta$ implies *poor* deletion guarantee.
> > > >
> > > > The authors show:
> > > > 1. Not $(0, \delta')$-DP for non-negligible $\delta'$ implies poor deletion guarantee.
> > > > 2. Not $(0, \delta')$-DP for non-negligible $\delta'$ implies not $(\epsilon, \delta)$-DP for reasonable $\epsilon, \delta$.
> > > >
> > > > However, the above two together doesn't give the claimed statement, right? So, it does not prevent existence of methods which are not $(\epsilon, \delta)$-DP for reasonable $\epsilon, \delta$ and yet have *good* deletion guarantee. Correct me if I am missing something.

---

> > > > > ### Author Response · Authors · 2022-11-21
> > > > > **Further Clarifications**
> > > > >
> > > > > Thank you for your comments!
> > > > >
> > > > > To clarify your doubt, we highlight a common misconception about the $(\epsilon,\delta)$ notion of differential privacy--not satisfying $(\epsilon, \delta)$-DP for reasonable values of $\epsilon, \delta$ (i.e. negligible $\delta$ but non-negligible $\epsilon$) does not mean that the mechanism necessarily leaks violating information about individual records. To see this, consider a mechanism $\mathcal A:\mathcal X^* \rightarrow [0,1]$ that returns a uniform sample from $[0,1]$ for a database $\mathcal D \in \mathcal X^*$ but for a neighbouring database $\mathcal D' \in \mathcal X^*$ it returns a uniform sample from $[0, 1/2) \cup (1/2, 1]$. This mechanism does not satisfy $(\epsilon, 0)$-DP for any $\epsilon < \infty$, simply because $1/2 \not\in \text{supp}(\mathcal A(\mathcal D'))$ but $1/2 \in \text{supp}(\mathcal A(\mathcal D))$.
> > > > >
> > > > > However, the advantage of any membership inference attack $\text{MI}(o) = \mathbb{I}\\{o \in O\\}$ for any $O \subset [0,1]$ is
> > > > > $$
> > > > > \text{Adv}(\text{MI}; \mathcal D, \mathcal D') = \Pr[\text{MI}(\mathcal A(\mathcal D)) = 1] - \Pr[\text{MI}(\mathcal A(\mathcal D')) = 1] = \Pr_{o \sim U([0,1]}[o \in O] - \Pr_{o \sim U([0, 1/2) \cup (1/2, 1]}[o \in O] = 0.
> > > > > $$
> > > > > Our example here shows that it is incorrect to say that an algorithm is privacy-violating if it **does not** satisfy $(\epsilon, \delta)$-DP with reasonably large values of $\epsilon$ and negligible $\delta$. Using a similar construction, one can also show that **not** $(\epsilon,0)$-DP does not mean that $(\mathcal A, \mathcal R)$ cannot be valid data-deletion algorithms.
> > > > >
> > > > > Now, when we claim that DP is necessary for doing data-deletion, what we really mean is that there should not exist adversaries that can do membership inference on outputs of $\mathcal A$ and $\mathcal R$ (which is implied by $(\epsilon, \delta)$-DP for reasonable $\epsilon$ and negligible $\delta$). And to show this claim, we assume there exists an adversary that can do membership inference (which is equivalent to **not** $(0, \delta)$-DP with non-negligible $\delta$) and exhibit a contradiction showing that a data-deletion guarantee is impossible.

---

> > > > > > ### Comment · Reviewer_hke4 · 2022-11-21
> > > > > > **Thanks for the response**
> > > > > >
> > > > > > Thanks for the clarifications. Indeed, a DP guarantee is not necessary to prevent successful membership inference attacks (or ensure TV stability). However, the claims in the paper such as "it is necessary to satisfy differential privacy to ensure true data deletion" (in the abstract), and "we also showed that differential privacy is necessary for data deletion" (conclusion) are not true in the, by now standard, formal sense of DP, which assumes reasonable settings of $\epsilon$ and $\delta$, unless explicitly specified. I would suggest that the authors revise them to either pertain to membership inference attacks or TV stability.

---

> > > > > > > ### Author Response · Authors · 2022-11-24
> > > > > > > **Thanks for the suggestion**
> > > > > > >
> > > > > > > Thank you for the suggestion. We agree that saying "differential privacy (with standard budget values) is necessary for data deletion" is not what our theorem shows, as the reviewer rightly pointed out. From our discussions, the problem with such a claim is that DP is not really a necessary condition for bounded membership inference (which is a problem with the standard DP definition, in our opinion). But our theorem still shows that if membership inference is possible for existing records, then membership inference would be possible for deleted records. Consequently, following the suggestion of the reviewer, we will amend our claim in the abstract and the conclusion that "deletion algorithms must ensure the privacy of existing records for providing privacy of deleted records when edit requests are adaptive."

---

> ### Author Response · Authors · 2022-11-10
> **Response 4 to reviewer JXqD ([Q4])**
>
> $$
> \newcommand{\nc}{\newcommand}
> \nc{\mc}{\mathcal}
> \nc{\Re}[2]{\mathrm{R}_q\left(#1\middle\Vert#2\right)}
> \nc{\ep}{\epsilon}
> \nc{\de}{\delta}
> \nc{\Th}{\Theta}
> \nc{\th}{\theta}
> \nc{\D}{\mc{D}}
> \nc{\X}{\mc{X}}
> $$
> **[Q4]**
> > What are the DP guarantees for the noisy SGD algorithm in Algorithm 1?
>
> The DP guarantee for Noisy-GD we refer to in Theorem 1 is a restatement of Abadi et al.'s, 2016, DP guarantee for their DP-SGD algorithm, but translated for our Noisy-GD variant described in Algorithm 1 and stated in terms of R\'enyi DP rather than $(\ep,\de)$-DP. The DP guarantee we refer to in Theorem 2 is the one proved by Chourasia et al., 2021, for Noisy-GD Algorithm 1.
>
> For the sake of completeness, we present the following proof for Theorem 1 and will include it in our paper's appendix.
>
> **Theorem 1 (RDP guarantee for Noisy-GD Algorithm 1).** If the loss function $\ell(\theta;x)$ is $L$-Lipschitz (or gradients are clipped to a magnitude of $L$), then Noisy-GD Algorithm 1 satisfies $(q,\ep)$-R\'enyi DP with $\ep=\frac{qL^2}{\sigma^2 n^2}\cdot\eta K$.
>
> *Proof.* The $L_2$ sensitivity of gradient $\nabla\mathcal L_{\D}(\th)\stackrel{\text{def}}{=}\frac{1}{|\D|}\sum_{x \in \D}\text{Clip}_L(\nabla \ell(\th;x))+\nabla\mathbf{r}(\th)$ computed in step 2 of Algorithm 1 for neighboring databases in $\X^n$ that differ in a single record is $\frac{2L}{n}$ since individual gradients are clipped to a magnitude of $L$ and divided by $|\D| = n$.
>
> Conditioned on observing the intermediate model $\Th_{\eta k}=\th_k$ at step $k$, the next model $\Th_{\eta(k+1)}$ after the noisy gradient update is a Gaussian mechanism with noise variance $2\sigma^2/\eta$. So, for neighboring databases $\D, \D' \in \X^n$, we have from the R\'enyi DP bound of Gaussian mechanisms proposed in [Proposition 7, Mironov, 2017] that
> $$
> \Re{\Th_{\eta(k+1)}\mid_{\Th_{\eta k}=\th_k}}{\Th_{\eta(k+1)}'\mid_{\Th_{\eta k}'=\th_k}}\leq\frac{\eta qL^2}{n^2\sigma^2},
> $$
> where $\\{\Th_{\eta k}\\}\_{0\leq k\leq K}$ and $\\{\Th_{\eta k}'\\}\_{0\leq k\leq K}$ are intermediate parameters in Algorithm 1 when run on databases $\D$ and $\D'$ respectively. Finally, from R\'enyi composition [Proposition 1, Mironov, 2017], we have
> $$
> \Re{\Th_{\eta K}}{\Th_{\eta K}'}\leq\Re{(\Th_0,\Th_\eta,\cdots,\Th_{\eta K})}{(\Th_0',\Th_\eta',\cdots,\Th_{\eta K}')}\leq\sum_{k=0}^{K-1}\Re{\Th_{\eta(k+1)}\mid_{\Th_{\eta k}=\th_k}}{\Th_{\eta(k+1)}'\mid_{\Th_{\eta k}'=\th_k}}\leq\frac{qL^2}{n^2\sigma^2}\cdot\eta K
> $$
> > Is it the exact same noise as DP-SGD?
>
> Different papers discussing Noisy-GD variants adopt different notational conventions for the total noise added to the gradients. The noise variance in our Algorithm 1 is $2\eta\sigma^2$; but is $\frac{\eta^2\sigma^2L^2}{n^2}$ in the full-batch setting of DP-SGD in Abadi et al., 2016. To translate the bound in Theorem 1, one can simply rescale $\sigma$ across different conventions to have the same noise variance, i.e., $2\eta\sigma^2=\frac{\eta^2\hat\sigma^2L^2}{n^2}$.
>
> To verify that our Theorem 1 is a restatement of Abadi et al.'s $(\ep,\de)$-DP bound, recall from R\'enyi DP to DP conversion in [Proposition 3, Mironov, 2017] that $(1+\frac{2}{\ep}\log\frac{1}{\de},\frac{\ep}{2})$-R\'enyi DP implies $(\ep,\de)$-DP. So, setting the bound in Theorem 1 to be smaller than $\frac{\ep}{2}$ and substituting $q=1+\frac{2}{\ep}\log\frac{1}{\de}$, we get
> $$
> \left(\frac{\ep+2\log\frac{1}{\de}}{\ep}\right)\frac{L^2}{n^2\sigma^2}\cdot\eta K\leq\frac{\ep}{2}\iff\frac{\sqrt{K(\ep+2\log\frac{1}{\de}})}{\ep}\leq\hat\sigma.
> $$
> For $\ep\leq2\log\frac{1}{\de}$, we get the same noise bound as in [Theorem 1, Abadi et al., 2016] for their (full-batch) DP-SGD.
> > Are the authors essentially running a pure DP algorithm and then just arguing in hindsight that it is also unlearning?
>
> Our proposed approach falls under the Descent-to-Delete framework proposed by Neel et al., 2021, wherein, after each deletion request, we run Noisy-GD starting from the previous model and perform a small number of gradient descent steps over records in the modified database; sufficient to ensure that the information about the deleted records is reduced within a desired bound.
>
> We emphasize that we do not argue that Noisy-GD is a reliable data-deletion algorithm simply because it is differentially private. Instead, in Section 5, we provide information bounds explicitly for the deleted records in the form of our data-deletion guarantees for Noisy-GD. Our data-deletion guarantees are much smaller than Noisy-GD's DP guarantees presented in Theorems 1 and 2. An interesting distinction between a DP guarantee and a data-deletion guarantee for Noisy-GD is that while the DP guarantee worsens on doing more iterations, the data-deletion guarantee constantly improves. This is because performing a noisy gradient step transfers information from the records in the current database to the model, but it only removes information about the deleted records from the model due to the injection of noise.

---

> ### Author Response · Authors · 2022-11-10
> **Response 5 to reviewer JXqD ([Q5])**
>
> **[Q5]** We do not know if the computational benefits of our data-deletion algorithm over our retraining algorithm would also hold when we care about the test loss performance. In general, the population risk of any unlearning algorithm depends on the quality of edit requests being sent. If the edit requests are replacing the records in the database with out-of-distribution samples, then the updated model would progressively perform worse on fresh samples drawn from the population. However, if the edit requests replace the records with i.i.d. samples taken from the same population, the population risk should ideally remain constant. The generalization guarantee for our proposed deletion mechanism is left as future research.

---

> ### Author Response · Authors · 2022-11-11
> **Response 6 to reviewer JXqD ([Q6])**
>
> $$
> \newcommand{\nc}{\newcommand}
> \nc{\mc}{\mathcal}
> \nc{\D}{\mc{D}}
> \nc{\R}{\mc{R}}
> \nc{\A}{\mc{A}}
> \nc{\Q}{\mc{Q}}
> \nc{\X}{\mc{X}}
> \nc{\O}{\mc{O}}
> \nc{\Ad}{\text{Adv}}
> \nc{\M}{\text{MI}}
> \nc{\T}{\text{Test}}
> \nc{\ep}{\epsilon}
> \nc{\de}{\delta}
> \nc{\Th}{\Theta}
> \nc{\th}{\theta}
> \nc{\Thh}{\hat\Th}
> \nc{\Thb}{\bar\Th}
> \nc{\ind}{\text{ind}}
> \nc{\Re}[2]{\mathrm{R}_q\left(#1\middle\Vert#2\right)}
> \nc{\ra}{\rightarrow}
> $$
> **[Q6]** Our Definition 4.1 of data deletion is sound but not complete.
>
> **Soundness.** We acknowledge that we did not provide proof of soundness for our Definition 4.1 in the paper. We present the following Theorem here to remedy that.
>
> **Theorem (Data-deletion guarantee is sound).** If the algorithm pair $(\A,\R)$ satisfies $(q,\ep)$-data-deletion guarantee under all $p$-adaptive $r$-requesters, then even with the power of designing an $p$-adaptive $r$-requester $\Q$ that interacts with the curator before deletion of a target record at any step $i$, no adversary observing only the post-deletion releases $\Thh_{i},\Thh_{i+1},\cdots$, can do membership inference for the deleted target with an advantage
> $$
> \Ad(\M)\geq\frac{qe^{\ep(q-1)/q}}{q-1}[2(q-1)]^{1/q} - 1.
> $$
>
> *Note:* The above bound on advantage approaches $0$ as $q\ra\infty$ and $\ep\ra0$, implying that our definition is sound.
>
> *Proof.* Suppose W.L.O.G. that an edit request $u_i$ deletes a single record at index '$\ind$' in $\D_{i-1}$. In the worst case, this record $\D_{i-1}[\ind]$ might have been there from the start, i.e. $\D_0[\ind]=\D_{i-1}[\ind]$, and influenced all the decision of the adaptive requester $\Q$ in the edit steps $1,\cdots,i-1$. To show that our data-deletion definition is sound, we need to prove that if $(\A,\R)$ satisfies our definition, then even in this worst-case scenario, no adaptive adversary can design a $\M(\Thh_i,\Thh_{i+1},\cdots)\in\\{0,1\\}$ that can distinguish the null hypothesis $H_0=\\{\D_{0}[\ind]=x\\}$ from the alternate hypothesis $H_1=\\{\D_0[\ind]=x'\\}$ for any $x,x'\in\X$ with high probability. That is, the advantage of any $\M$ attack, defined as
> $$
> \Ad(\M)\stackrel{\text{def}}{=}\Pr[\M(\Thh_i,\Thh_{i+1},\cdots)=1|H_0]-\Pr[\M(\Thh_i,\Thh_{i+1},\cdots)=1|H_1],
> $$
> must be small. Since $\R$ is stateless and after processing $u_i$ the databases $\D_i,\D_{i+1},\cdots$ no longer contain the deleted record $\D_{i-1}[\ind]$, the data processing inequality implies that $\Thh_{i+1},\Thh_{i+2},\cdots$ generated by $\R$ cannot have more information about $\D_{i-1}[\ind]$ than what is present in $\Thh_i$. Therefore any $\M(\Thh_i,\Thh_{i+1},\cdots)$ has an advantage smaller than the optimal $\M^*(\Thh_i)\in\\{0,1\\}$ that only uses $\Thh_i$.
>
> Since $(\A,\R)$ satisfy $(q,\ep)$-data-deletion for any $\Q$, we know there exists a mapping $\pi^\Q_i$ such that for all $\D_0\in\X^n$, the model $\Thh_i$ generated by the interaction between $(\A,\R,\Q)$ on $\D_0$ after $i$ rounds satisfy $\Re{\Thh_i}{\pi^\Q_i(\D_0\circ u_i)}\leq\ep$. As the databases $\D_0\circ u_i$ is identical under both hypotheses $H_0$ and $H_1$, we have $\Re{\Thh_i\mid_{H_b}}{\Thb}\leq\ep$ for $b\in\\{0,1\\}$, where $\Thb \sim \pi^\Q_i(\D_0\circ u_i)$. From R\'enyi divergence to $(\ep,\de)$-indistinguishability conversion described in Remark 1, we get
> $$
> \Pr[\M^*(\Thh_i)=1|H_0]\leq e^{\ep'(\de)}\Pr\[\M^*(\Thb_i)=1]+\de,
> $$
> $$
> \Pr[\M^*(\Thh_i)=0|H_1]\leq e^{\ep'(\de)}\Pr[\M^*(\Thb_i)=0]+\de,
> $$
> where $\ep'(\de)=\ep+\frac{\log 1/\de}{q-1}$ for any $0<\de<1$. Therefore,
> $$
> \Ad(\M)\leq\Ad(\M^*)=\Pr[\M^*(\Thh_i)=1|H_0]-\Pr[\M^*(\Thh_i)=1|H_1]\leq\min_\de e^{\ep'(\de)}-1+2\de=\frac{qe^{\ep(q-1)/q}}{q-1}[2(q-1)]^{1/q}-1.
> $$
>
> **Completeness.** Our Definition 4.1 of data-deletion applies to several valid deletion mechanisms that do not satisfy unlearning guarantees of Ginart et al., 2019; Guo et al., 2019; Neel et al., 2021; Gupta et al., 2021. For example, the trivial example of a deletion mechanism $\R$ that outputs a fixed untrained model in $\th\in\O$ regardless of its inputs satisfies $(0,0)$-data-deletion for all adaptive/non-adaptive requesters for a mapping $\pi^\Q_i(\D)=\th$. However, we do not claim that our data-deletion guarantee is complete, and there could be valid data-deletion mechanisms that do not satisfy our definition.
>
> *Note:* We will include the above theorem on soundness and a discussion on completeness in our paper.

---

> ### Author Response · Authors · 2022-11-11
> **Response 7 to reviewer JXqD ([Q7])**
>
> **[Q7]** The recent work of Gupta et al., 2021, also studies adaptive data deletion and proves a reduction from adaptive unlearning guarantee to non-adaptive unlearning guarantee under DP. As reviewer [hke4](https://openreview.net/forum?id=goLFJ0ZNwl&noteId=RbY89eRsFzI) points out, the reduction [Theorem 3.1, Gupta et al., 2021] relies on DP with regards to a change in the description of learning/unlearning algorithm's coins and not with regards to the standard replacement of records. In contrast, our Theorem 5 presents a reduction from adaptive to non-adaptive data-deletion guarantee under DP with respect to the standard replacement of records. We emphasize that these two Theorems are fundamentally different, which we elaborate on here.
>
> The adaptive unlearning definition of Gupta et al., 2021, is designed to ensure with a high probability that no adaptive adversary $\mathcal Q$ can force the output distribution of the unlearning algorithm $\mathcal R(\mathcal D_{i-1}, u_i, \hat\Theta_{i-1})$ to diverge substantially from that of retraining using the learning algorithm $\mathcal A(\mathcal D_i)$. Such an attack is possible in stateful unlearning algorithms that rely on persistent structures that are only randomized once during initialization, for example, the initial partitioning of start database $\mathcal D_0$ in Bourtoule et al., 2021's SISA algorithm. Gupta et al., 2021, show in their Theorem 5.1 that an adaptive update requester $\mathcal Q$ can interactively send deletion requests $u_1, \cdots, u_i$ to SISA so that the partitioning of remaining records in $\mathcal D_i = \mathcal D_0 \circ u_1 \cdots u_i$ follows a pattern that is unlikely to occur on repartitioning of $\mathcal D_i$ when executing $\mathcal A(\mathcal D_i)$. As proved in their reduction in Theorem 3.1, a straightforward way to prevent this is by ensuring that the uncertainty regarding the persistent structures remains private for long periods of time to an adversary observing the unlearned model. Hence the proof of their reduction from adaptive unlearning guarantee to non-adaptive unlearning guarantee relies on DP with regards to the coins of the unlearning algorithm.
>
> Our paper shows that satisfying Gupta et al.'s adaptive unlearning definition still does not guarantee data deletion. In Theorem 4, we demonstrate that there exists an algorithm pair $(\mathcal A, \mathcal R)$ satisfying (a strictly stronger version) of adaptive unlearning [Definition 2.3, Gupta et al., 2021], but still causes blatant non-privacy of deleted records in post-deletion release. The vulnerability we identify occurs because an adaptive requester can learn the identity of any target record before it is deleted and re-encode it back in the curator's database by sending edit requests. Because of this, an adversary (who knows how the adaptive requester works but does not have access to the requester's interaction transcript) can extract the identity of the target record from the model released after processing the deletion request. We argue that a reliable (and necessary) way to prevent this attack is to make sure that no adaptive requester ever learns the identity of a target record from the $p$ pre-deletion model releases it has access to. Consequently, our reduction in Theorem 5 from adaptive to non-adaptive requests relies on DP with respect to the replacement of records instead.

---

> ### Author Response · Authors · 2022-11-11
> **Response 8 to reviewer JXqD ([Q8])**
>
> $$
> \newcommand{\nc}{\newcommand}
> \nc{\mc}{\mathcal}
> \nc{\Re}[2]{\mathrm{R}_q\left(#1\middle\Vert#2\right)}
> \nc{\R}{\mc{R}}
> \nc{\A}{\mc{A}}
> \nc{\X}{\mc{X}}
> \nc{\U}{\mc{U}}
> \nc{\D}{\mc{D}}
> \nc{\Q}{\mc{Q}}
> \nc{\O}{\mc{O}}
> \nc{\Thh}{\hat\Theta}
> \nc{\ep}{\epsilon}
> \nc{\de}{\delta}
> $$
> **[Q8]** We address the reviewer's comments on notations and inconsistencies as follows.
>
> > Definition 2.5, eqn 8 is not clear. I feel it will help to elaborate a bit more on what it means.
>
> Definition 2.5 describes an adaptive update requester that interacts with the curator by observing the published outcomes and sending edit requests. We quantify the *strength* of an adaptive requester by two integers $(p, q)$. The integer $p$ denotes the maximum number of published outcomes an update requester sees in the entire interaction for generating the edit requests. The integer $r$ denotes the maximum number of records an update requester can edit in a single request. Equation (7), restated below, defines an adaptive update requester $\mathcal Q$ that sees *all* the previously published outcomes and issued requests.
> $$
> u_1 = \Q(\phi_0), u_2 = \Q(\phi_0, u_1, \phi_1), \cdots, u_i = \Q(\phi_0, u_1, \phi_1, u_2, \cdots, u_{i-1}, \phi_{i-1})
> $$
> Equation (8), on the other hand, defines an adaptive update requester $\Q$ that only sees a maximum of $p$ of the published outcomes at arbitrary edit steps $s^1, s^2, \cdots, s^p$. Such an adaptive requester is equivalent to some function $\Q'$ that depends only on the available outcomes from the subset $\{\phi_{s^1}, \phi_{s^2}, \cdots \phi_{s^p}\}$ of observations generated during the entire interaction and the previously issued edit requests. That is, for any $i \geq 1$, a $p$-adaptive update requester $\Q$ must be equivalent to
> $$
> u_i = \Q(\phi_0, u_1,\phi_1 \cdots, u_{i-1}, \phi_{i-1}) = \Q'(u_1, u_2, \cdots, u_{i-1}; \phi_{s^1}, \phi_{s^2}, \cdots, \phi_{s^j})
> $$
> such that $s^j < i$. We will rewrite equation (8) to improve clarity.
>
> > - Definition 4.1 has a typo "all edit requests u = ... \in U^i " instead of U^1.
> >
> > - Independence is defined for two random variables (and not two distributions). Please fix the paper to not use phrases like “there exists a distribution \pi_i^u that is independent of the record being deleted” (e.g., in defn 4.1). This is not mathematically consistent.
>
> We thank the reviewer for identifying this typo and inconsistency. We provide the following fixed definition below.
>
> **Definition 4.1 ($(q,\ep)$-data-deletion under $p$-adaptive $r$-requesters).** Let $q>1$, $\ep\geq 0$, and $p, r \in \mathbb{N}$. We say that an algorithm pair $(\A, \R)$ satisfies $(q,\ep)$-data-deletion under $p$-adaptive $r$-requesters if the following condition holds for all edit requests $u \in \U^1$ and all $p$-adaptive $r$-requesters $\Q$. For every step $i \geq 1$, their exists a randomized mapping $\pi_i^\Q: \X^n \rightarrow \O$ such that for all initial databases $\D_0 \in \X^n$,
> $$
> \Re{\R(\D_{i-1},u,\Thh_{i-1})}{\pi_i^Q(\D_0\circ u)}\leq\ep.
> $$
> This change only addresses the descriptional issue and does not affect any proof ideas in the paper. We will include this change in our updated rebuttal paper.
>
> > Change the notation R to denote the unlearning algorithm (it is confusing as R should ideally be used to denote the retraining algorithm).
>
> We will update our notation for the deletion algorithm to make it less confusing.

---

### Official Review · Reviewer_4yeU · 2022-10-28

**Confidence:** 3
**Correctness:** 4
**Technical Novelty And Significance:** 3
**Empirical Novelty And Significance:** Not applicable
**Recommendation:** 6

**Clarity, Quality, Novelty And Reproducibility:**

* The presentation is a bit dense, which makes it tedious to follow the paper. It seems like the introduction is written in a somewhat flashy manner. It is not clear how some points in the Introduction are related to technical contents. (See suggestions and questions below.)

* The claims seem to be correct, even though I did not go through the detailed proofs.

* The contributions are fairly novel, but the presentation can be improved.

**Suggestions and questions:**

1. The paper says that “We emphasize that we are not advocating for doing data deletion through differentially-private mechanisms that cap the information content of all records equally, deleted or otherwise, which is known to be inefficient (Sekhari et al., 2021). Instead, data-deletion mechanisms should provide two differing information reattainment bounds; one for records currently in the database in the form of a differential privacy guarantee and the other for records previously deleted in the form of a data-deletion guarantee.”

* The method proposed in the paper relies on retraining with DP-GD. So, the above statement is a bit confusing. It would be good to provide more details.

2. It would be important to give more details about how computational savings over retraining are calculated.

3. Notation $\mathcal{O}$ is used several times before it is defined (on page 4).

4. It is not clear why Theorems 1 and 2 need to be included with details in the main text. That space may be better used to give some high level details about the proofs of Theorems 3 and 4.

5. Edit request is considered as a replacement operation, and it is argued that adversary can duplicate a target record before its deletion. Considering edit requests in machine unlearning seems a bit stronger adversary model. Also, the service provider can employ simple defenses such as examine the edit request to check for duplication with any existing training sample, and discard any request that attempts to insert a duplicate item. It would be helpful to discuss these points.


**Strength And Weaknesses:**

**Strengths:**
1. The problem of machine unlearning (data deletion) is practically relevant and technically challenging. It is important to analyze current methods and develop new definitions of data deletion in ML.

**Weaknesses:**
1. The paper does not provide sufficient details on how the compute savings are calculated. Data deletion method proposed in Def. 5.1 is essentially retraining the model using noisy-GD (DP-GD). Since it also performs retraining (but with noisy-GD, it is not clear how the compute savings in Table 1 are calculated. It is important to add more details on this.

2. The paper does not discuss how the reduction result in Theorem 5 relates to the reduction result in [Gupta et al. 2021]. It would be helpful to give more details.

3. The paper does not present any empirical results to support the theory. In contrast, many prior works including [Gupta et al. 2021] present some empirical results to corroborate the theory.


**Summary Of The Paper:**

This paper investigates whether the existing definitions for machine unlearning are ‘complete’. Prior works on machine unlearning defined ‘certified unlearning’ (at a high level) as indistinguishability from retrained models. This paper argues that this is incomplete, and demonstrate two vulnerabilities (Theorems 3 and 4) such  that a majority of existing unlearning guarantees suffer from one of them. Then, the paper proposes a stronger definition of unlearning, and uses it to propose objectives for data deletion. Finally, the paper analyzes convex and non-convex loss functions to that DP-GD (noisy GD) satisfies the proposed objectives.

**Summary Of The Review:**

The paper asks important questions about machine unlearning, and shows some interesting results. However, the presentation can be significantly improved, and some details need to be added.

---

> ### Author Response · Authors · 2022-11-15
> **Response 1 to reviewer 4yeU**
>
> > The paper does not provide sufficient details on how the compute savings are calculated. Data deletion method proposed in Def. 5.1 is essentially retraining the model using noisy-GD (DP-GD). Since it also performs retraining (but with noisy-GD, it is not clear how the compute savings in Table 1 are calculated. It is important to add more details on this.
>
> We clarify that our data-deletion mechanism proposed in Definition 5.1 is *not retraining a model from scratch using Noisy-GD*. Our data-deletion mechanism $\mathcal R_{\text{Noisy-GD}}$ in Definition 5.1 falls under the Descent-to-Delete approach of Neel et al., 2021, wherein, after each deletion request $u_i$ received at step $i$, we run Noisy-GD to fine-tune *the previous model* $\hat\Theta_{i-1}$ by performing a small number, $K_{\mathcal R}$, of gradient descent steps over records in the modified database $\mathcal D_i \stackrel{\text{def}}{=} \mathcal{D_{i-1}} \circ u_i$. That is, our data-deletion mechanism is defined as
> $$
> \mathcal R_{\text{Noisy-GD}}(\mathcal D_{i-1}, u_i, \hat\Theta_{i-1}) \stackrel{\text{def}}{=} \text{Noisy-GD}(\mathcal D_{i-1} \circ u_i, \hat\Theta_{i-1}, K_{\mathcal R}) = \hat\Theta_{i} \quad \text{for all}\ i\geq 1.
> $$
> Our learning or fresh retraining algorithm $\mathcal A_{\text{Noisy-GD}}$ is also Noisy-GD, but it starts with a random model $\Theta_0$ taken from a Gaussian weight initialization distribution $\rho$ and runs a significantly larger number of gradient update steps $K_{\mathcal A} > K_{\mathcal R}$. In our proposed data-deletion pipeline, the learning algorithm is executed only once on the initial database $\mathcal D_0$ to get the first model
> $$
> \hat\Theta_0 = \mathcal A_{\text{Noisy-GD}}(\mathcal D_0) \stackrel{\text{def}}{=} \text{Noisy-GD}(\mathcal D_0, \Theta_0, K_{\mathcal A}),
> $$
> after which only the data-deletion algorithm $\mathcal R_{\text{Noisy-GD}}$ is used to process edit requests cheaply.
>
> The compute saving listed in the third column of Table 1 describes how fewer gradient computations a data-deletion/unlearning algorithm $\mathcal R$ requires for processing the $i$th edit request in comparison to the fresh retraining algorithm $\mathcal A$, under the constraint that the models produced by both $\mathcal R$ and $\mathcal A$ must satisfy the same utility bounds. That is to say, if we want to generate models with near-optimal excess empirical risk guarantees after every edit request, how much faster would it be if we fine-tuned the current model with $\mathcal R$ rather than training a fresh model with $\mathcal A$. For a fair comparison across different unlearning approaches in the literature, we additionally require that each algorithm pair $(\mathcal A, \mathcal R)$ listed in Table 1 satisfy an $(\epsilon, \delta)$-data-deletion guarantee for all non-adaptive requesters.
>
> > 1. The paper says that “We emphasize that we are not advocating for doing data deletion through differentially-private mechanisms that cap the information content of all records equally, deleted or otherwise, which is known to be inefficient (Sekhari et al., 2021). Instead, data-deletion mechanisms should provide two differing information reattainment bounds; one for records currently in the database in the form of a differential privacy guarantee and the other for records previously deleted in the form of a data-deletion guarantee.”
> > - The method proposed in the paper relies on retraining with DP-GD. So, the above statement is a bit confusing. It would be good to provide more details.
>
> Our proposed data-deletion algorithm $\mathcal R_{\text{Noisy-GD}}$ is not a retraining algorithm but a fine-tuning algorithm, as clarified earlier. Furthermore, we do not argue that our proposed algorithm $\mathcal R_{\text{Noisy-GD}}$ is a reliable data-deletion algorithm simply because Noisy-GD is differentially private. Instead, in Section 5, we provide information bounds explicitly for the deleted records in the form of our data-deletion guarantees for $(\mathcal A_{\text{Noisy-GD}}, \mathcal R_{\text{Noisy-GD}})$. Our data-deletion guarantees are much smaller than Noisy-GD's DP guarantees presented in Theorems 1 and 2. An interesting distinction between a DP guarantee and a data-deletion guarantee for Noisy-GD is that while the DP guarantee worsens on doing more iterations, the data-deletion guarantee constantly improves. This is because performing a noisy gradient step transfers information from the records in the current database to the model, but it only removes information about the deleted records from the model due to the injection of noise.

---

> > ### Comment · Reviewer_4yeU · 2022-12-01
> > **Thank you for clarifications**
> >
> > I thank the authors for providing detailed answers to my questions.

---

> ### Author Response · Authors · 2022-11-15
> **Response 2 to reviewer 4yeU**
>
> > The paper does not discuss how the reduction result in Theorem 5 relates to the reduction result in [Gupta et al. 2021]. It would be helpful to give more details.
>
> We refer the reviewer to our following [comment](https://openreview.net/forum?id=goLFJ0ZNwl&noteId=dvw6JdqPe-) where we highlight the fundamental differences with Gupta et al., 2021. We will add this discussion as an appendix to our paper.
>
> > Edit request is considered as a replacement operation, and it is argued that adversary can duplicate a target record before its deletion. Considering edit requests in machine unlearning seems a bit stronger adversary model. Also, the service provider can employ simple defenses such as examine the edit request to check for duplication with any existing training sample, and discard any request that attempts to insert a duplicate item. It would be helpful to discuss these points.
>
> We mean duplicate in the broader sense of information and not the explicit record (database value) alone. In our Theorem 4 of Section 3, we present a construction where an interactive requester can alter other records in the database to encode the information of a target record being deleted. This encoding need not be in form of the same explicit database value of the target record, but can be a general pattern induced over the combination of other records that may not even be trivially detectable. Consider a naive scenario, where a celebrity posts an image but later wants it deleted. Other fan accounts could have extracted his image from a published ML model by running reconstruction attacks, edited the image in multiple different ways (crop, resize, add text, etc.) and reposted it so the information about the target datapoint to be deleted is present in the database irrespective of carrying out the deletion operation on just the celebrity’s photo. Our proposed Noisy-GD based data-deletion algorithm prevents such attacks as no fan account can accurately reconstruct the celebrity's posts accurately in the first place by virtue of differential privacy.
>
> > The paper does not present any empirical results to support the theory. In contrast, many prior works including [Gupta et al. 2021] present some empirical results to corroborate the theory.
>
> We acknowledge that we do not provide any empirical results. Instead, in Theorems 6 and 7, we provide utility guarantees (in the form of excess empirical risk) for our proposed data-deletion algorithm that match the known lower bound for convex losses and is near-optimal for non-convex losses.
>
> > Notation $\mathcal O$ is used several times before it is defined (on page 4).
>
> Thank you for pointing this out. We will fix it.
>
> > It is not clear why Theorems 1 and 2 need to be included with details in the main text. That space may be better used to give some high level details about the proofs of Theorems 3 and 4.
>
> Thank you for this suggestion. We will try to address it for the sake of clarity in some key sections of the paper.

---

### Decision · Program_Chairs · 2023-01-20

**Decision:**

Reject

**Justification For Why Not Higher Score:**

Significant changes from the original submitted version

**Justification For Why Not Lower Score:**

N/A

**Metareview: Summary, Strengths And Weaknesses:**

This paper saw enormous amounts of reviewer comments and responses from the authors. It appears that the original submitted paper had a number of lacking aspects, including, e.g., a limited perspective on previous works, claiming DP is necessary for data deletion, etc. The authors did a heroic effort in revising the paper, addressing many of these issues, and I believe the paper has improved dramatically since the original submission. However, I, and the reviewers, feel uncomfortable accepting the paper in its current state, without careful scrutiny afforded to the large number of new claims and proofs. While reviewers should be expected to investigate small, few, and well-scoped changes, multiple reviewers requested significant changes, which modifies the content of the paper from the original submission in a stark fashion. This is not a journal submission, so I do not expect reviewers to scrutinize such a changed manuscript for ICLR. The manuscript, however, appears at a glance to be far improved from the original submission and will likely see acceptance at a future conference.